# Multilinear Mixture of Experts:
# Scalable Expert Specialization through Factorization

**James Oldfield**[1]* **Markos Georgopoulos** **Grigorios G. Chrysos**[2] **Christos Tzelepis**[3]
**Yannis Panagakis**[4,5] **Mihalis A. Nicolaou**[6] **Jiankang Deng**[7] **Ioannis Patras**[1]

[1]Queen Mary University of London [2]University of Wisconsin-Madison [3]City University of London [4]National and Kapodistrian University of Athens [5]Archimedes AI, Athena RC [6]The Cyprus Institute [7]Imperial College London

## Abstract

The Mixture of Experts (MoE) paradigm provides a powerful way to decompose dense layers into smaller, modular computations often more amenable to human interpretation, debugging, and editability. However, a major challenge lies in the computational cost of scaling the number of experts high enough to achieve fine-grained specialization. In this paper, we propose the **Mu**ltilinear **M**ixture **o**f **E**xperts ($\boldsymbol{\mu}$**MoE**) layer to address this, focusing on vision models. $\mu$MoE layers enable scalable expert specialization by performing an implicit computation on prohibitively large weight tensors *entirely in factorized form*. Consequently, $\mu$MoEs (1) avoid the restrictively high inference-time costs of dense MoEs, yet (2) do not inherit the training issues of the popular sparse MoEs' discrete (non-differentiable) expert routing. We present both qualitative and quantitative evidence that scaling $\mu$MoE layers when fine-tuning foundation models for vision tasks leads to more specialized experts at the class-level, further enabling manual bias correction in CelebA attribute classification. Finally, we show qualitative results demonstrating the expert specialism achieved when pre-training large GPT2 and MLP-Mixer models with parameter-matched $\mu$MoE blocks at every layer, maintaining comparable accuracy. Our code is available at: https://github.com/james-oldfield/muMoE.

## 1 Introduction

The Mixture of Experts (MoE) architecture [1] has reemerged as a powerful class of conditional computation, playing the pivotal role in scaling up recent large language [2, 3, 4, 5], vision [6], and multi-modal models [7]. MoEs apply different subsets of layers (referred to as 'experts') for each input, in contrast to the traditional approach of using the same single layer for all inputs. This provides a form of input-conditional computation [8, 9, 10, 11] that is expressive yet efficient. However, through their substantial performance gains, an important emergent property of MoEs is frequently underutilized: the innate tendency of experts to specialize in distinct subtasks. Indeed, the foundational work of Jacobs et al. [12] on MoEs describes this property, highlighting how implementing a particular function with modular building blocks (experts) often leads to subcomputations that are easier to understand individually than their dense layer counterparts–with larger expert counts allowing for more fine-grained specialization.

Independent of model performance, a successful decomposition of the layer's functionality into human-comprehensible subtasks offers many significant benefits. Firstly, the mechanisms through which a network produces an output are more *interpretable*: the output is a sum of modular components, each contributing individual functionality. Yet, the value of interpretable computation

---

*Corresponding author: j.a.oldfield@qmul.ac.uk

38th Conference on Neural Information Processing Systems (NeurIPS 2024).

extends beyond just transparency [13] and explainability [14]. An important corollary of successful task decomposition amongst experts is that layers are easier to debug and edit. Biased or unsafe behaviors can be better localized to specific experts' subcomputation, facilitating manual correction or surgery in a way that minimally affects the other functionality of the network. Addressing such behaviors is particularly crucial in the context of foundation models; being often fine-tuned as black boxes pre-trained on unknown, potentially imbalanced data distributions. Furthermore, there is evidence that traditional fairness techniques are less effective in large-scale models [15, 16]. However, to achieve fine-grained expert specialism at the class level (or more granular still), one needs the ability to significantly scale up the number of experts. When using only a small expert count, each expert is forced to process and generalize across *multiple* distinct semantic concepts, hindering specialization. Conversely, a large expert count means each can specialize to a more specific set of semantically similar inputs. Alas, the dominating 'sparse' MoE paradigm of selecting only the top-$K$ experts [17] is not only parameter-inefficient for large expert counts, but also has several well-known issues due to its discrete expert routing–often leading to training instability and difficulties in scaling the total expert count, amongst other challenges [18, 19].

In this paper, we propose the *Multilinear Mixture of Experts* ($\mu$MoE) layer to address these issues. $\mu$MoEs are designed to scale gracefully to dense operations involving *tens of thousands* of experts at once through implicit computations on a factorized form of the experts' weights. Furthermore, in contrast to the dominant sparse MoEs' [17] non-differentiable nature, $\mu$MoEs are differentiable by design, and thus do not inherit the

Table 1: Benefits of the proposed $\mu$MoEs' model form over existing MoEs.

|  | Differentiable | Parameter-efficient | FLOPs-efficient |
|---|---|---|---|
| Dense MoE [1] | ☺ | ☹ | ☹ |
| Sparse MoE [17] | ☹ | ☹ | ☺ |
| **$\mu$MoE (ours)** | ☺ | ☺ | ☺ |

associated training issues. We summarize the benefits of $\mu$MoEs' model form over existing MoEs in Table 1. Crucially, we show evidence that scaling up the number of $\mu$MoE experts leads to increased expert specialism when fine-tuning foundation models for vision tasks. Our evidence is provided in three forms: (1) firstly, through the usual qualitative evaluation of inspecting inputs by their expert coefficients. Secondly (2), we further explore the *causal* role of each expert through counterfactual interventions [20]. Lastly, (3) we show how final-layer $\mu$MoE expert specialism facilitates the practical task of model editing–how subcomputation in specific combinations of experts biased towards demographic subpopulations can be manually corrected through straightforward guided edits.

Building on these findings, we demonstrate that $\mu$MoEs offer a compelling alternative to MLPs for pre-training both vision and language models with up to 100M parameters–enabling large numbers of specialized experts while maintaining comparable performance and parameter counts to the original networks' *single* dense MLPs.

Our contributions and core claims can be summarized as follows:

- We introduce $\mu$MoE layers–a mechanism for computing vast numbers of subcomputations and efficiently fusing them conditionally on the input.
- We show both qualitatively (through visualization) and quantitatively (through counterfactual intervention) that *increasing the number of $\mu$MoE experts increases task modularity*–learning to specialize in processing just specific input classes when fine-tuning large foundation models for vision tasks. Further, we show manual editing of $\mu$MoE expert combinations can straightforwardly mitigate demographic bias in CelebA attribute classification.
- We pre-train both language (GPT2) and vision (MLP-mixer) $\mu$MoE networks, establishing experimentally that models with parameter-matched $\mu$MoE blocks are competitive with existing MLP blocks whilst facilitating expert specialism (qualitatively) throughout.

## 2   Related Work

**Mixture of Experts**   Recent years have seen a resurgence of interest in the Mixture of Experts (MoE) architecture for input-conditional computation [17, 12, 21, 2]. One primary motivation for MoEs is their increased model capacity through large parameter count [17, 4, 2]. In contrast to a single dense layer, the outputs of multiple experts performing separate computations are combined (sometimes with multiple levels of hierarchy [22, 23]). A simple approach to fusing the outputs is by taking either a convex [23] or linear [24] combination of the output of each expert. The

seminal work of Shazeer et al. [17] however proposes to take a *sparse* combination of only the top-$K$ most relevant experts, greatly reducing the computational costs of evaluating them all. More recent works employ a similar sparse gating function to apply just a subset of experts [2, 25], scaling to billions [3] and trillions of parameters [4]. The discrete expert selection choice of sparse MoEs is not without its problems, however–often leading to several issues including training stability and expert under-utilization [18, 19].

Particularly relevant to this paper are works focusing on designing MoE models to give rise to more interpretable subcomputation [26, 27, 28]–hearkening back to one of the original works of Jacobs et al. [12], where experts learned subtasks of discriminating between different lower/uppercase vowels. Indeed a common observation is that MoE experts appear to specialize in processing inputs with similar high-level features. Researchers have observed MoE experts specializing in processing specific syntax [17] and parts-of-speech [29] for language models, and foreground/background [30] and image categories (e.g. 'wheeled vehicles') [24] in vision. Evidence of shared vision-language specialism is even found in the multi-modal MoEs of Mustafa et al. [7].

Several works instead target how to make conditional computation more efficient: by sharing expert parameters across layers [31], factorizing gating network parameters [32], or dynamic convolution operations [33]. Relatedly, Gao et al. [34] jointly parameterize the experts' weight matrices with a Tensor-Train decomposition [35]. However, such approach still suffers from the Sparse MoE's instability and expert under-utilization issues, and stochastic masking of gradients must be performed to lead to balanced experts. Furthermore, whilst Gao et al. [34] share parameters across expert matrices, efficient implicit computation of thousands of experts simultaneously is not facilitated, in contrast to the $\mu$MoE layer.

**Factorized layers**    in the context of deep neural networks provide several important benefits. Replacing traditional operations with low-rank counterparts allows efficient fine-tuning [36] / training [37, 38], and modeling of higher-order interactions [39, 40, 41, 42, 43], and convolutions [44]. In addition to reducing computational costs, tensor factorization has also proven beneficial in the context of multi-task/domain learning [45, 46] through the sharing of parameters/low-rank factors across tasks. Furthermore, parameter efficiency through weight factorization often facilitates the design and efficient implementation of novel architectures such as polynomial networks [47, 48, 49] or tensor contraction layers [50]. The recent DFC layer in Babiloni et al. [51] also performs dynamic computation using the CP decomposition [52] like $\mu$MoEs. Nevertheless, the two works have very different goals and model properties due to how the weight matrices are generated. $\mu$MoEs take a sparse, convex combination of $N$ explicit experts' latent factors. This consequently leads to specialized subcomputations in a way that facilitates the interpretability and editability presented in this paper. DFCs can be seen to apply an MLP to input vectors at this step in analogy, which does not provide the necessary model properties of interest here.

## 3   Methodology

We first formulate the proposed $\mu$MoE layer in Section 3.1, introducing 2 unique resource-efficient models and forward passes in Section 3.1.1. Finally, we show in Section 3.1.2 how $\mu$MoEs recover linear MoEs as a special case.

**Notation**    We denote scalars $x \in \mathbb{R}$ with lower-case letters, and vectors $\mathbf{x} \in \mathbb{R}^{I_1}$ and matrices $\mathbf{X} \in \mathbb{R}^{I_1 \times I_2}$ in lower- and upper-case boldface latin letters respectively. Tensors $\mathcal{X} \in \mathbb{R}^{I_1 \times I_2 \times \ldots \times I_d}$ of order $d$ are denoted with calligraphic letters. We refer to the $(i_1, i_2, \ldots, i_d)$-th element of this tensor with both $\mathcal{X}(i_1, i_2, \ldots, i_d) \in \mathbb{R}$ and $x_{i_1 i_2 \ldots i_d} \in \mathbb{R}$. Finally, we use a colon to index into all elements along a particular mode: given $\mathcal{X} \in \mathbb{R}^{I_1 \times I_2 \times I_3}$ for example, $\mathbf{X}_{::i_3} \in \mathbb{R}^{I_1 \times I_2}$ or equivalently $\mathcal{X}(:, :, i_3) \in \mathbb{R}^{I_1 \times I_2}$ is the matrix at index $i_3$ of the final mode of the tensor. We use $\mathcal{X} \times_n \mathbf{u}$ to denote the **mode-$n$ (vector) product** [53] of a tensor $\mathcal{X} \in \mathbb{R}^{I_1 \times I_2 \times \ldots \times I_N}$ and vector $\mathbf{u} \in \mathbb{R}^{I_n}$ whose resulting elements are given by $(\mathcal{X} \times_n \mathbf{u})_{i_1 \ldots i_{n-1} i_{n+1} \ldots i_N} = \sum_{i_n=1}^{I_n} x_{i_1 i_2 \ldots i_N} u_{i_n}$.

### 3.1   The $\mu$MoE layer

$\mu$MoEs provide a scalable way to execute and fuse large numbers of operations on an input vector by formalizing conditional computation through resource-efficient multilinear oper-

ations. A $\mu$MoE layer comprised of $N$ many experts (and a single level of expert hierarchy) is parameterized by weight tensor $\mathcal{W} \in \mathbb{R}^{N \times I \times O}$ and expert gating parameter $\mathbf{G} \in \mathbb{R}^{I \times N}$. Given an input vector $\mathbf{z} \in \mathbb{R}^I$ (denoting the hidden representation of an individual token, for example), its forward pass can be expressed through the series of tensor contractions:

$$\mathbf{a} = \phi(\mathbf{G}^\top \mathbf{z}) \in \mathbb{R}^N,$$
$$\mathbf{y} = \mathcal{W} \times_1 \mathbf{a} \times_2 \mathbf{z}$$
$$= \sum_{n=1}^{N} \sum_{i=1}^{I} \mathbf{w}_{ni:} z_i a_n \in \mathbb{R}^O, \qquad (1)$$

where $\mathbf{a}$ is the vector of expert coefficients and $\phi$ is the entmax activation [54, 55]. The $\mu$MoE layer can be understood as taking a sparse, convex combination of $N$ many affine transformations[2] of input vector $\mathbf{z}$, weighted by the coefficients in $\mathbf{a}$. The first tensor contraction in the forward pass ($\sum_i \mathbf{W}_{:i:} z_i \in \mathbb{R}^{N \times O}$) matrix-multiplies the input vector with *every* expert's weight matrix. The following tensor contraction with expert coefficients $\mathbf{a}$ takes a linear combination of the results, yielding the output vector. The forward pass can be visualized intuitively as

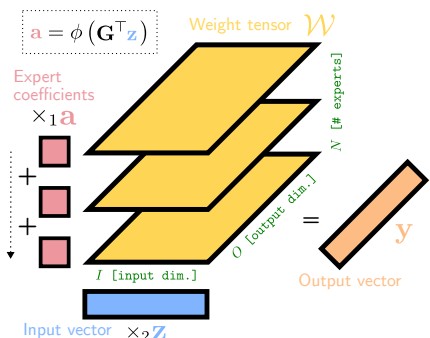

Figure 1: The forward pass of an (unfactorized) $\mu$MoE layer as a series of tensor contractions: the experts' weight matrices (yellow 2D slices) are matrix-multiplied with the input vector and summed (weighted by the red expert coefficients).

multiplying and summing over the modes in a 3D tensor, which we illustrate in Figure 1. Furthermore, $\mu$MoEs readily generalize to hierarchical conditional computations by introducing additional modes to the weight tensor and corresponding vectors of expert coefficients (see Appendix E).

### 3.1.1 Computation in factorized form

Our key insight is that the dense $\mu$MoE forward pass over all $N$ experts simultaneously can be **computed entirely in factorized form, needing never materialize prohibitively large weight tensors**. This allows $\mu$MoEs' computations to scale gracefully to many thousands of experts simultaneously, without the problematic top-$K$ gating [17]. To achieve this, we (1) first parameterize the experts' weights $\mathcal{W} \in \mathbb{R}^{N \times I \times O}$ with a tensor factorization and (2) re-derive fast forward passes of Equation (1) to operate solely in factorized form.

In the context of a $\mu$MoE layer, the various choices of tensor factorizations make different trade-offs regarding parameter/FLOP counts and rank constraints. We derive two unique resource-efficient $\mu$MoE variants to suit different computational budgets and choices of expert counts. We now present the derivations of the forward passes of the factorized $\mu$MoE models (with `einsum` pseudocode implementations in Appendix B):

**CP$\mu$MoE** Imposing CP structure [52, 56] of rank $R$ on the weight tensor, we can write $\mathcal{W} = \sum_{r=1}^{R} \mathbf{u}_r^{(1)} \circ \mathbf{u}_r^{(2)} \circ \mathbf{u}_r^{(3)} \in \mathbb{R}^{N \times I \times O}$ as a sum of $R$ outer products, with factor matrices $\mathbf{U}^{(1)} \in \mathbb{R}^{R \times N}, \mathbf{U}^{(2)} \in \mathbb{R}^{R \times I}, \mathbf{U}^{(3)} \in \mathbb{R}^{R \times O}$. This reduces the parameter count from $NIO$ (such as with sparse/dense MoEs and regular $\mu$MoEs) to just $R(N + I + O)$. Crucially, we can further rewrite the CP$\mu$MoE layer's forward pass entirely in factorized form without ever materializing the full tensor (plugging the CP-composed tensor into Equation (1)) as:

$$\mathbf{y} = \sum_{n=1}^{N} \sum_{i=1}^{I} \left( \sum_{r=1}^{R} \mathbf{u}_r^{(1)} \circ \mathbf{u}_r^{(2)} \circ \mathbf{u}_r^{(3)} \right)_{ni:} z_i a_n = \sum_{r=1}^{R} \left( \mathbf{U}^{(2)} \mathbf{z} \right)_r \left( \mathbf{U}^{(1)} \mathbf{a} \right)_r \mathbf{u}_r^{(3)} \in \mathbb{R}^O, \qquad (2)$$

with Equation (2) being analogous to the fast computation in Babiloni et al. [51], only here the operations of combining the weights and producing the outputs can be expressed in a single step. Whilst the original naive CP$\mu$MoE forward pass has a FLOP count[3] of $NIO$, the fast computation

---

[2]Incrementing the dimension of the second 'input' mode of the weight tensor $\mathcal{W} \in \mathbb{R}^{N \times (I+1) \times O}$ and appending a 1 to the input vector $\mathbf{z} \in \mathbb{R}^{I+1}$ folds a per-expert bias term into the computation.

[3]We adopt the convention of counting fused multiply-adds as one operation [57]. Note that the small additional expert coefficients cost is constant across models and thus ignored in comparisons.

above has just $R(N + I + O)$ (the same number of factorized layer parameters). With moderate values of both $R$ and $N$, the layer becomes significantly more resource-efficient than vanilla $\mu$MoEs.

**TR$\mu$MoE**    We propose a second $\mu$MoE variant based on the Tensor Ring [58] (TR) factorization that can offer even better efficiency for large values of $N$. In TR format, $\mathcal{W} \in \mathbb{R}^{N \times I \times O}$ has three factor tensors: $\mathcal{U}^{(1)} \in \mathbb{R}^{R_1 \times N \times R_2}, \mathcal{U}^{(2)} \in \mathbb{R}^{R_2 \times I \times R_3}, \mathcal{U}^{(3)} \in \mathbb{R}^{R_3 \times O \times R_1}$, where $R_i$ are the manually chosen ranks[4]. The weight tensor's elements in TR format are given by: $w_{nio} = \text{tr}\big(\mathbf{U}^{(1)}_{:n:}\mathbf{U}^{(2)}_{:i:}\mathbf{U}^{(3)}_{:o:}\big)$ [58]. TR$\mu$MoE's forward passes can be computed efficiently by contracting the first two factor tensors with the input/expert coefficients vectors and then combining the results:

$$\mathbf{y} = \sum_{n=1}^{N} \sum_{i=1}^{I} \mathbf{w}_{ni:} z_i a_n = \sum_{r_1=1}^{R_1} \sum_{r_3=1}^{R_3} \big(\underbrace{(\mathcal{U}^{(1)} \times_2 \mathbf{a})(\mathcal{U}^{(2)} \times_2 \mathbf{z})}_{[R_1 \times R_3]}\big)_{r_1 r_3} \mathbf{u}^{(3)}_{r_3:r_1} \in \mathbb{R}^O, \qquad (3)$$

yielding a modified FLOP count of $(R_1 N R_2 + R_2 I R_3 + R_1 R_2 R_3 + R_1 O R_3)$ with just $(R_1 N R_2 + R_2 I R_3 + R_3 O R_1)$ parameters. With large $N$ contributing to the computational cost only through $R_1 N R_2$, the TR$\mu$MoE can prove even more resource-efficient than CP$\mu$MoEs by choosing small values of $R_1, R_2$. We refer readers to Appendix D for a further discussion of decomposition choice, derivations of how tensor rank translates to expert matrix rank, and FLOPs comparisons.

### 3.1.2 $\mu$MoEs recover dense MoEs as a special case

Finally, we note how unfactorized $\mu$MoE layers with a single level of expert hierarchy recover dense MoE layers [17, 11] as a special case. When computing Equation (1) over the full materialized weight tensor, one can alternatively write the output element-wise as $y_o = \mathbf{a}^\top \mathbf{W}_{::o}\mathbf{z}$. This highlights an interesting technical connection between neural network layers: dense MoE layers in this tensor formulation can be seen to share a similar functional form to bilinear layers, which have also found applications in interpretability [59, 60].

## 4    Experiments

We start in Section 4.1 by presenting both qualitative and quantitative experiments validating that the experts learn to specialize in processing different semantic clusters of the input data. In Section 4.2 we demonstrate one practical benefit of the learned specialism–showing how expert-conditional re-writing can correct for specific demographic bias in CelebA attribute classification. Finally, in Section 4.3 we train both large language and large vision models with $\mu$MoE layers throughout–providing qualitative evidence of expert specialism and model performance competitive with networks using MLP blocks. Please see Appendix H for detailed ablation studies, and Appendix I for experiments with hierarchical $\mu$MoEs.

**Implementation details**    Before applying the activation function to the expert coefficients we apply batch- and layer-normalization to $\mu$MoE layers in vision and language models respectively (see Appendix H.3 for an ablation). Interestingly, we do not find the need for any load-balancing losses. We fix the TR$\mu$MoE ranks to be $R_1 = R_2 = 4$ throughout (see Appendix D.1.2).

### 4.1    Expert specialism: visualization & intervention

Our first objective is to show that **scaling $\mu$MoE's expert count leads to more specialized experts**. We provide evidence of this effect both qualitatively (through *visualization*) and quantitatively (through *intervention*).

To isolate the impact of $\mu$MoE layers and varying expert counts, we first explore the controlled setting of fine-tuning large foundation models CLIP [61] `ViT-B-32` and DINO [62] on ImageNET1k (following the fine-tuning protocol in Ilharco et al. [63, 64]). Whilst fine-tuning large foundation models is an important application of $\mu$MoE layers in its own right (e.g. as explored later in Section 4.2 for fairer models), the ability to cheaply train many models with different $\mu$MoE layer configurations forms an ideal setting in which to study their properties.

---

[4]Setting $R_1 = 1$ recovers a Tensor Train [35] $\mu$MoE.

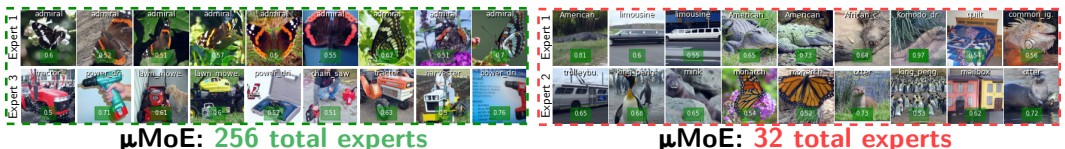

**µMoE: 256 total experts**          **µMoE: 32 total experts**

Figure 2: Specialization in 256 vs 32 total expert CPµMoE layers (fine-tuned on CLIP ViT-B-32). Each row displays *randomly* selected images processed (with coefficient ≥ 0.5) by the first few experts for the two models. The more we scale the expert count, the greater the apparent expert specialism (to single visual themes or image categories).

### 4.1.1 Qualitative results

We first show *random* examples in Figure 2 of images processed (with expert coefficient ≥ 0.5) by the experts by each CPµMoE layer (the class labels and expert coefficients are overlaid in white and green text respectively). Using only a modest number of experts (e.g. 32) appears to lead to some 'polysemanticity' [65] in experts–with some processing unrelated classes of images (e.g. 'gators', 'limos', and a 'quilt' for Expert 1 on the right). On the other hand, using a much larger number of total experts appears to yield more specialization, with many experts contributing their computation to only images of the same single class label or broader semantic category. Please see Figure 16 in the Appendix for many more random images for the first 10 experts per model to observe this same trend more generally, and Figure 17 for even finer-grained specialism with 2048-expert µMoE layers.

### 4.1.2 Quantitative results: expert monosemanticity

The qualitative evidence above hints at the potential of a prominent benefit to scaling up the number of experts with µMoEs. Such subjective interpretations alone about expect specialism are *hypotheses*, rather than conclusions however [66]. Similarities in images processed by the same expert give us an intuitive explanation of its function but do not show the expert's computation contributes *causally* [20, 67, 68] to the subtask of processing specific human-understandable patterns of input features [69, 70]. However, the absence of ground-truth labels for interpretable features of the input one may be interested in (e.g. specific types of textures in images, or words related to 'Harry Potter') makes this difficult to quantify in any objective or systematic manner.

Despite the absence of fine-grained labels, we *can* quantify and compare the class-level specialism a µMoE expert exhibits on the ImageNET1k dataset as an (imperfect) proxy [71].

Following the causal intervention protocol of Elazar et al. [20], we ask the specific counterfactual question about solely each expert $n$ in a µMoE layer in turn: *"had expert $n$'s weight matrix $\mathbf{W}_n$ not contributed its computation, would the network's test-set accuracy for class $c$ have dropped?"* Practically speaking, given a network fine-tuned with an µMoE layer, we achieve this by intervening in the forward pass by zeroing the $n^{\text{th}}$ expert's weight matrix $\mathbf{W}_n := \mathbf{0}$, leaving every other aspect of the forward pass completely untouched. Let the elements of $\mathbf{y}, \hat{\mathbf{y}}^{(n)} \in \mathbb{R}^C$ denote the test set accuracy for the $C = 1000$ ImageNET1k classes, pre- and post-intervention of expert $n$ respectively. We collect the normalized difference to per-class accuracy in the vector $\mathbf{d}^{(n)}$, whose elements are given by $d_c^{(n)} = (y_c - \hat{y}_c^{(n)})/y_c$. At the two extremes, when the full network's accuracy for

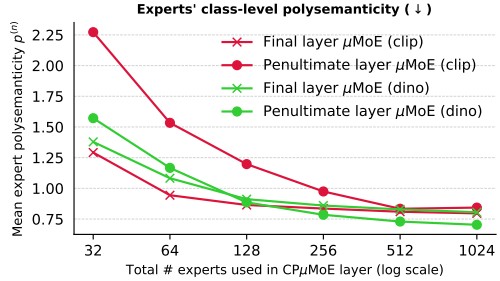

Figure 3: **Higher expert counts lead to more monosemantic experts**: mean expert class-level polysemanticity of Equation (4) (↓) as a function of the total number of experts. Results are shown for both CLIP ViT-B-32 and DINO models fine-tuned on ImageNET1k with CPµMoE layers.

class $c$ drops completely from $y_c$ to 0 upon manually excluding expert $n$'s computation we get $d_c^{(n)} = 1$, whilst $d_c^{(n)} = 0$ means the absence of the subcomputation did not change class $c$'s test set accuracy at all. We thus estimate the 'class-level polysemanticity' of expert $n$ as the distance between

Table 2: Fairness metrics for baseline models and after applying standard fairness techniques, for the two experiments on CelebA. A `CPμMoE-r512-e128` model is used as the final layer.

| | (a) Bias towards 'Old females' for 'Age' prediction head | | | | | (b) Bias towards 'Blond males' for 'Blond Hair' prediction head | | | | | |
|---|---|---|---|---|---|---|---|---|---|---|---|
| | Target subpop. acc. (↑) | Equality of opp. [76] (↓) | STD bias [77] (↓) | Subpop. Max-Min [78] (↑) | Test set acc. (↑) | Target subpop. acc. (↑) | Equality of opp. [76] (↓) | STD bias [77] (↓) | Subpop. Max-Min [78] (↑) | Test set acc. (↑) | # Params |
| Linear | 0.516 | 0.226 | 0.185 | 0.516 | 88.944 | 0.346 | 0.534 | 0.263 | 0.346 | 95.833 | 30.7K |
| HighRankLinear | 0.513 | 0.228 | 0.186 | 0.513 | 88.920 | 0.353 | 0.529 | 0.260 | 0.353 | 95.831 | 827K |
| **CPμMoE** | **0.555** | **0.197** | **0.167** | **0.555** | **89.048** | **0.409** | **0.476** | **0.236** | **0.409** | **95.893** | 578K |
| + oversample | 0.669 | 0.086 | 0.120 | 0.669 | **89.009** | 0.655 | 0.226 | 0.131 | 0.655 | **95.750** | 578K |
| + adv. debias [79] | 0.424 | 0.274 | 0.226 | 0.424 | 87.785 | 0.193 | 0.630 | 0.325 | 0.193 | 95.031 | 579K |
| + blind thresh. [76] | 0.843 | **0.082** | 0.084 | 0.700 | 83.369 | 0.843 | 0.139 | 0.063 | 0.841 | 92.447 | 578K |
| + expert thresh. **(ours)** | **0.866** | 0.097 | **0.066** | **0.756** | 84.650 | **0.847** | **0.051** | **0.048** | **0.846** | 94.895 | 578K |

the difference vector and the one-hot vector:

$$p^{(n)} = ||\mathbf{d}^{(n)} - \mathbb{1}^{(n)}||_2, \tag{4}$$

where index $\arg\max_c(d_c^{(n)})$ of $\mathbb{1}^{(n)}$ has a value of 1 (and values of 0 everywhere else). This encodes the signature of a perfectly class-level monosemantic expert, for which *all* accuracy for a single class alone is lost in the counterfactual scenario in which the expert $n$ did not contribute. We plot in Figure 3 the average expert polysemanticity $p^{(n)}$ for all experts with non-zero difference vectors[5], observing a steady drop in its value as $N$ increases from 32 to 1024 total experts. In other words, **increasing $N$ leads to individual experts increasingly responsible for a single subtask**: classifying all inputs of just one class. As shown in Figure 3 we observe this trend both when $\mu$MoEs are used as final classification layers and as penultimate layers (followed by a ReLU activation and linear classification layer), and for multiple pre-trained foundation models. We further refer readers to the bar plots of the values of $\mathbf{d}^{(n)}$ (the per-class accuracy changes) in Figures 18 and 19, where this trend is observable through mass concentrated on increasingly fewer class labels as the number of experts increases.

## 4.2 Expert re-writing: conditional bias correction

We further validate the modular expert hypothesis of $\mu$MoEs and simultaneously provide a concrete example of its usefulness by correcting demographic bias in attribute classification. Classifiers trained to minimize the standard binary cross-entropy loss often exhibit poor performance for demographic subpopulations with low support [72, 73]. By identifying which combination of experts is responsible for processing target subpopulations, we show how one can straightforwardly manually correct mispredictions in a targeted way–without *any* re-training.

We focus on mitigating bias towards two low-support subpopulations in models with $\mu$MoE final layers fine-tuned on CelebA [74]: (a) bias towards images labeled as 'old females' for age prediction [75], and (b) bias towards images labeled as 'blond males' for blond hair prediction [15]. Concretely, we train $N = 128$ multi-label $\mu$MoE final layer models for the 40 binary attributes in CelebA, jointly optimizing a pre-trained CLIP ViT-B-32 model [61] backbone, again following the fine-tuning setup in Ilharco et al. [63, 64]. All results presented in this section are the average of 10 runs with different random seeds.

**Experimental setup** Let $C$ be a set collecting the expert coefficients $\mathbf{a} \in \mathbb{R}^N$ from forward passes of the training images belonging to the target subpopulation. We evaluate the subpopulation's mean expert coefficients $\bar{\mathbf{a}} = 1/|C| \sum_{\mathbf{a} \in C} \mathbf{a} \in \mathbb{R}^N$, proposing to manually re-write the output of this expert combination. We modify the layer's forward pass for the $o^{\text{th}}$ output head for attribute of interest (e.g. 'blond hair') as:

$$y_o = \mathbf{a}^\top \mathbf{W}_{::o}\mathbf{z} + \lambda \bar{\mathbf{a}}^\top \mathbf{a}. \tag{5}$$

Here, the term $\lambda \bar{\mathbf{a}} \in \mathbb{R}^N$ specifies, for each expert, how much to increase/decrease the logits for attribute $o$, with $\lambda$ being a scaling hyperparameter[6]. Taking the dot product with an input image's expert coefficients $\mathbf{a}$ applies the relevant experts' correction terms (in the same way it selects a subset of the most relevant experts' weight matrices). We report a range of standard fairness metrics for both the model rewriting and networks trained with existing techniques (that aim to mitigate demographic

---

[5]I.e. we include only experts that, when ablated in isolation, alter the class accuracy; please see the Appendix for discussion on expert load.

[6]We set $\lambda := N$ for all experiments for simplicity, but we note that its value could require tuning in different experimental setups. The sign of $\lambda$ is chosen to correct the bias in the target direction (whether to move the logits positively/negatively towards CelebA's e.g. young/old binary age labels respectively).

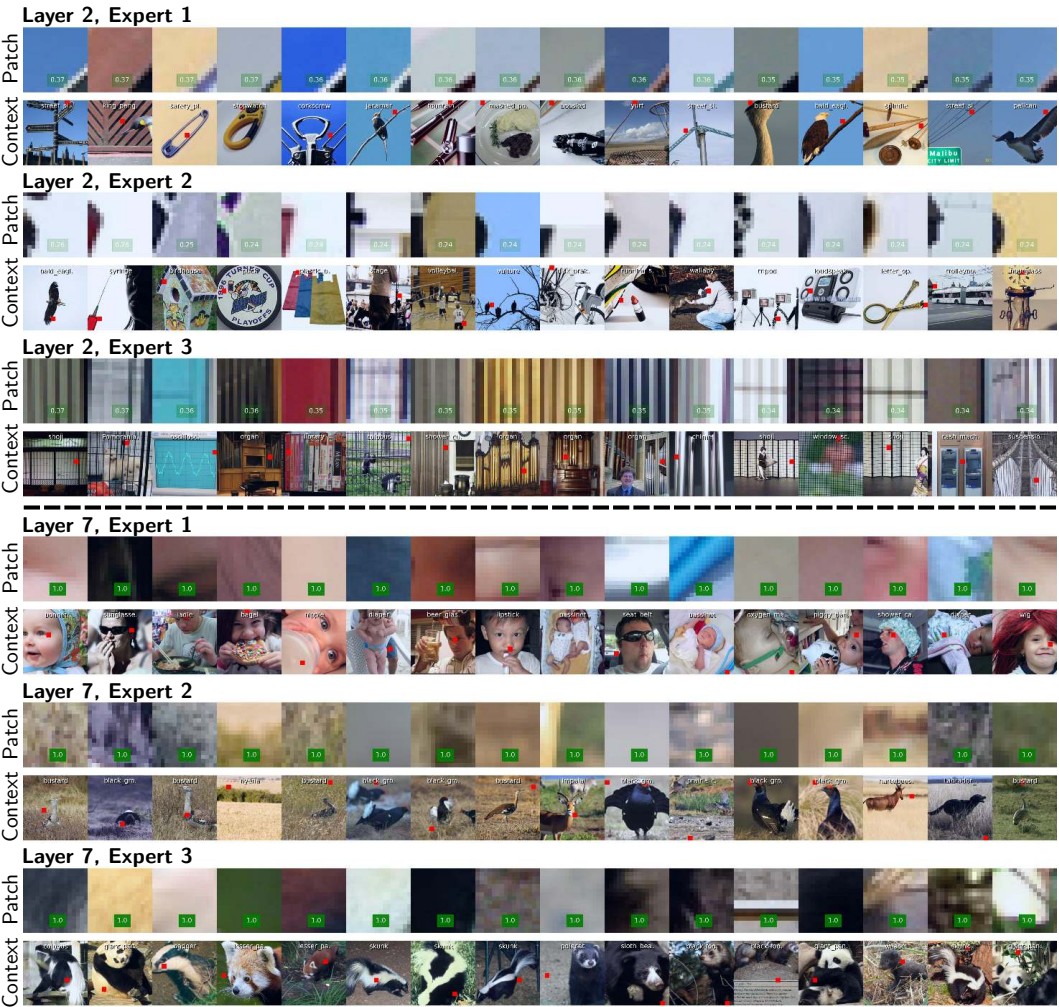

Figure 4: Top-activating patches (top rows) and their full images (second rows) for the first 3 experts across 2 CP$\mu$MoE-e64 layers in $\mu$MoE MLP-mixer [80] models–$\mu$MoE blocks exhibit coarse-grained specialism (e.g. texture) earlier and more fine-grained specialism (e.g. objects) deeper in the network.

bias without requiring images' sensitive attribute value at test time). These are shown in Table 2 for the two different experiments on CelebA, where the proposed intervention outperforms baseline alternative methods in the majority of settings. Please see Appendix J for details about the baseline methods and fairness metrics used, and further discussion of results.

### 4.3 Large language/vision $\mu$MoE networks

Finally, we train from scratch 12 layer 124M-parameter GPT-2 [81] LLMs on OpenWebText [82] for the language domain and 8 layer S-16 variant[7] MLP-Mixers [80] on ImageNET1k [83] for vision. We replace *every* MLP block's 2 linear layers with 2 $\mu$MoE layers. Each token $t$'s input vector $\mathbf{z}_t \in \mathbb{R}^I$ is therefore transformed with $\mu$MoE blocks of the form:

$$\mathbf{y}_t = \sum_{n_2=1}^{N} \sum_{h=1}^{H} \mathbf{w}_{n_2h:}^{(2)} \text{GELU}\bigg( \sum_{n_1=1}^{N} \sum_{i=1}^{I} \mathbf{w}_{n_1i:}^{(1)} z_{ti} a_{tn_1} \bigg)_h a_{tn_2}, \quad \mathbf{a}_t = \phi(\mathbf{G}^\top \mathbf{z}_t),$$

where $\mathbf{a}_t \in \mathbb{R}^N$ are the expert coefficients for each specific token and block, $H$ is the dimension of the block's hidden layer, and $\mathcal{W}^{(1)} \in \mathbb{R}^{N \times I \times H}, \mathcal{W}^{(2)} \in \mathbb{R}^{N \times H \times O}$ are the (implicit) $\mu$MoE weight

---

[7]The S-16 model is the largest configuration that fits into 4x80GB A100 GPUs using the original paper's batch size of 4096.

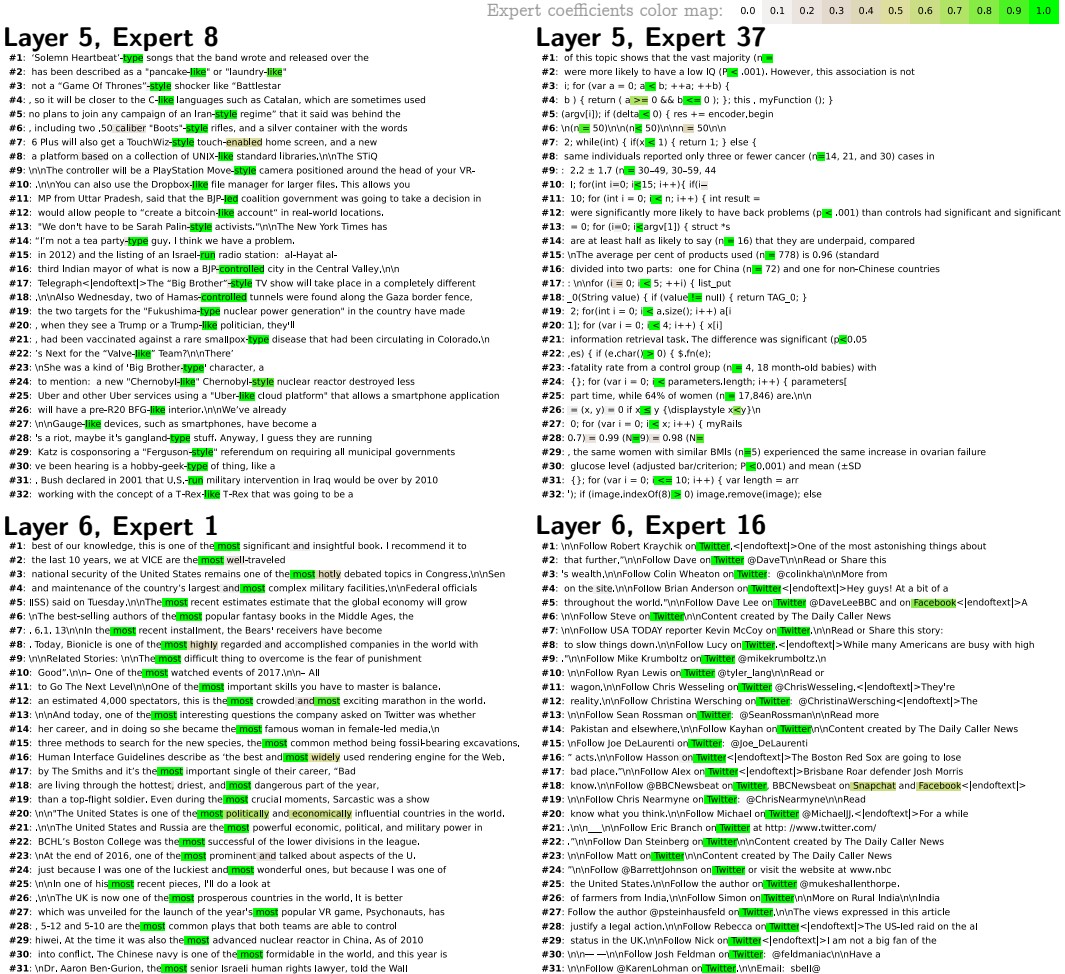

Figure 5: Top-activating generated tokens for 4 manually selected experts for GPT-2 trained with CPµMoE blocks at every layer (each token is highlighted by the coefficient of the expert in question), exhibiting specializations to concepts including compound adjectives and equality operators.

tensors for each of the two layers. We manually set the µMoE ranks to parameter-match each original network and set the number of experts (per block) to $N = 64$ for vision models and $N = 256$ for LLMs. Consequently, with this configuration, **each layer's µMoE block performs computations with $N$ experts yet has the same parameter counts and FLOPs as a single, dense MLP block**.

**µMoE-Mixer**  For vision, our key findings are that earlier µMoE channel-mixing blocks' experts appear (qualitatively) to exhibit specialisms to colors, shapes, and textures, whilst later layers exhibit more object-specific specialization. We plot the patches from the training set for which each expert most contributes its computation in Figure 4 for both a shallow and deep layer to illustrate this–earlier layers' experts contribute strongly to the processing of similar *patches* (top rows, e.g. specific edges) whilst later layers' experts process tokens based more on the similarity of their surrounding semantic context (bottom rows, e.g. images of animals). We further show in Figure 12 results for the first 2 experts across all 8 blocks where such scale-specific specialism is apparent across the entire network.

**µMoE-GPT2**  For LLMs, we see promising qualitative evidence of experts specializing throughout a corpus of 1M generated 100-token sequences. At layer 5, for example, the generated tokens that use expert 8 with the highest coefficient are compound adjectives (Figure 5), whilst expert 37 most highly activates for equality and comparison operators in code and scientific text (please see examples of

Table 3: Comparison of $\mu$MoEs and dense MLPs across different models and tasks. We use $N = 64$ $\mu$MoE experts for the two vision tasks and $N = 256$ for GPT2. MLP mixers and GPT2s are pre-trained for 300 epochs and 100k iterations respectively, whilst CLIP is fine-tuned for 10 epochs.

| | **MLP-mixer** S-16 (ImageNET1k) | | **GPT-2 NanoGPT** (OWT) | | **CLIP** B-32 (ImageNET1k) | |
| | Val. acc. ($\uparrow$) | #params | Val. loss ($\downarrow$) | #params | Val. acc. ($\uparrow$) | #params |
|---|---|---|---|---|---|---|
| MLPs | 70.31 | 18.5M | **2.876** | 124M | 77.99 | 769K |
| **TR$\mu$MoEs** | 71.26 | 18.3M | 2.886 | 124M | **78.71** | 771K |
| **CP$\mu$MoEs** | **71.29** | 18.6M | 2.893 | 124M | 78.07 | 769K |

many unfiltered experts in Figures 13 and 14). Whilst monosemanticity is not always attained, $\mu$MoE layers nonetheless facilitate a level of specialism not facilitated by dense MLP layers.

One important result here is that $\mu$MoE networks in this setup are significantly more parameter-efficient than both dense and sparse MoEs with the same expert count, as shown in Table 4. For example, GPT-2 models with 256 sparse/dense MoE experts require a prohibitive 14.5B MLP parameters alone, relative to just 57M MLP parameters with $\mu$MoEs of the same expert counts.

**$\mu$MoE performance** Finally, we substantiate our claim that networks pre-trained and fine-tuned with parameter-matched $\mu$MoE layers are competitive with their existing linear layer alternatives across multiple domains/machine learning tasks. We present in Table 3 the performance results for MLP-Mixer S-16 [80], NanoGPT GPT-2 [81], and (fine-tuned) CLIP ViT-B-32 [61] models on the OWT and ImageNET1k datasets. Following Section 4.1.1, we replace all linear

Table 4: MLP parameters required for networks with the same expert counts.

| Model | NanoGPT (gpt2) $N = 256$ | MLP-Mixer (S-16) $N = 64$ |
|---|---|---|
| Dense/Sparse MoE | 14.5B | 1.13B |
| **CP$\mu$MoE** | **57.0M** | **17.7M** |
| **TR$\mu$MoE** | **57.4M** | **17.4M** |

layers with $\mu$MoE blocks (and a single $\mu$MoE final layer for fine-tuning CLIP). We initialize all linear layers following the default PyTorch $U[-k, k]$ initialization for a fair comparison. Please see Appendix F for experimental details and learning curves, and Appendix I for experiments with varying expert count and hierarchical $\mu$MoEs. Crucially, whilst $\mu$MoE layers provide additional interpretability benefits through scalable expert specialization, they do not sacrifice accuracy when parameter-matched to MLP blocks, as seen from the comparable performance.

## 5 Conclusion

In this paper, we introduced the Multilinear Mixture of Experts layer ($\mu$MoE). We demonstrated that larger expert counts lead to increased specialization, and how $\mu$MoE layers make this computationally tractable through factorized forward passes. $\mu$MoEs scale to large expert counts much more gracefully than existing MoEs, yet avoid the issues from popular gating mechanisms. As a further practical example of $\mu$MoE's task decomposition, we illustrated how manual guided edits can be made to correct bias towards demographic subpopulations in fine-tuned foundation models. Having also shown matching performance in addition to expert specialism in both large vision and language models, we believe $\mu$MoE layers constitute an important step towards facilitating increasingly performant models that do not trade off fairness/interpretability for accuracy.

**Limitations** Firstly, it is important to state again that our quantitative evaluation only captures expert behavior on the test set, not out-of-distribution data [70, 84]. Furthermore, expert specialism in large models is only demonstrated qualitatively (through the expert coefficients) due to the absence of fine-grained labels. Developing ways of quantifying fine-grained expert specialism is an important direction for future research. Finally, our experimental results demonstrated comparable accuracies of $\mu$MoE networks only for models with parameter counts on the order of 100 million. Where resources permit, future work should explore the scalability of expert specialization and performance of $\mu$MoEs in even larger-scale LLMs.

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

# Appendix

## Table of Contents

## A  Broader impact

This paper presents work whose goal is to advance the field of *interpretable* machine learning. Our goal is not to improve model capabilities but rather an orthogonal one of designing architectures

more interpretable and controllable. As with many work with an interpretability focus, however, the $\mu$MoE layer could nonetheless facilitate the further development of SOTA models through its more expressive computation. We thus encourage the development of further guardrails against potentially harmful dual-uses of such technology. We release our code upon acceptance to facilitate further research along such lines.

# B    Fast $\mu$MoE implementations

We here detail how to implement the fast forward passes of the $\mu$MoE models in a batch-wise manner, where each mini-batch element is a 2D matrix of shape $\mathbf{Z} \in \mathbb{R}^{T \times C}$ (with 'token' and 'channel' dimensions) with PyTorch and einops' [85] einsum:

## B.1    CP$\mu$MoE einsum implementation

The CP$\mu$MoE forward pass can be implemented with:

```
# CPmuMoE (r=CP rank, b=batch_dim, t=tokens,
# i=input_dim, o=output_dim, a[e]=expert_coefs, n*=expert_dims)
y = einsum(G3, a[0]@G1.T, z@G2.T, 'r o, b t r, b t r -> b t o')
```

And a two-level hierarchical CP$\mu$MoE with an additional factor matrix as:

```
# CPmuMoE (r=CP rank, b=batch_dim, t=tokens,
# i=input_dim, o=output_dim, a[e]=expert_coefs, n*=expert_dims)
#################
# A 2-level hierarchical CPmuMoE, assuming Gi's of appropriate shape
y = einsum(G4, a[0]@G1.T, a[1]@G2.T, z@G3.T,
           'r o, b t r, b t r, b t r -> b t o')
```

## B.2    TR$\mu$MoE einsum implementation

TR$\mu$MoEs can be implemented with:

```
# TRmuMoE (r*=TR ranks, b=batch_dim, t=tokens,
# i=input_dim, o=output_dim, a[e]=expert_coefs, n*=expert_dims)

# batched mode-2 tensor-vector products
f1 = einsum(a[0], G1, 'b t n1, r1 n1 r2 -> b t r1 r2')
f2 = einsum(z, G2, 'b t i, r2 i r3 -> b t r2 r3')

# batch-multiply f1@f2
fout = einsum(f1, f2, 'b t r1 r2, b t r2 r3 -> b t r1 r3')

# contract with final TR core
y = einsum(G3, fout, 'r3 o r1, b t r1 r3  -> b t o')
```

And a two-level hierarchical version with an additional TR-core as:

```
# TRmuMoE (r*=TR ranks, b=batch_dim, t=tokens,
# i=input_dim, o=output_dim, a[e]=expert_coefs, n*=expert_dims)
#################
# A 2-level hierarchical TRmuMoE, assuming additional TR cores Gi
f1 = einsum(a[0], G1, 'b t n1, r1 n1 r2 -> b t r1 r2')
f2 = einsum(a[1], G2, 'b t n2, r2 n2 r3 -> b t r2 r3')
f3 = einsum(z, G3, 'b t i, r3 i r4 -> b t r3 r4')

# batch-multiply f1@f2@f3
fout = einsum(f1, f2, 'b t r1 r2, b t r2 r3 -> b t r1 r3')
fout = einsum(fout, f3, 'b t r1 r3, b t r3 r4 -> b t r1 r4')

# contract with final TR core
y = einsum(G4, fout, 'r4 o r1, b t r1 r4  -> b t o')
```

## C $\mu$MoE forward pass visualization

For intuition, we provide a visualization in Figure 6 of the step-by-step series of tensor contractions $\mathcal{W} \times_1 \mathbf{a} \times_2 \mathbf{z} \in \mathbb{R}^O$ that the $\mu$MoE computes (in non-factorized form).

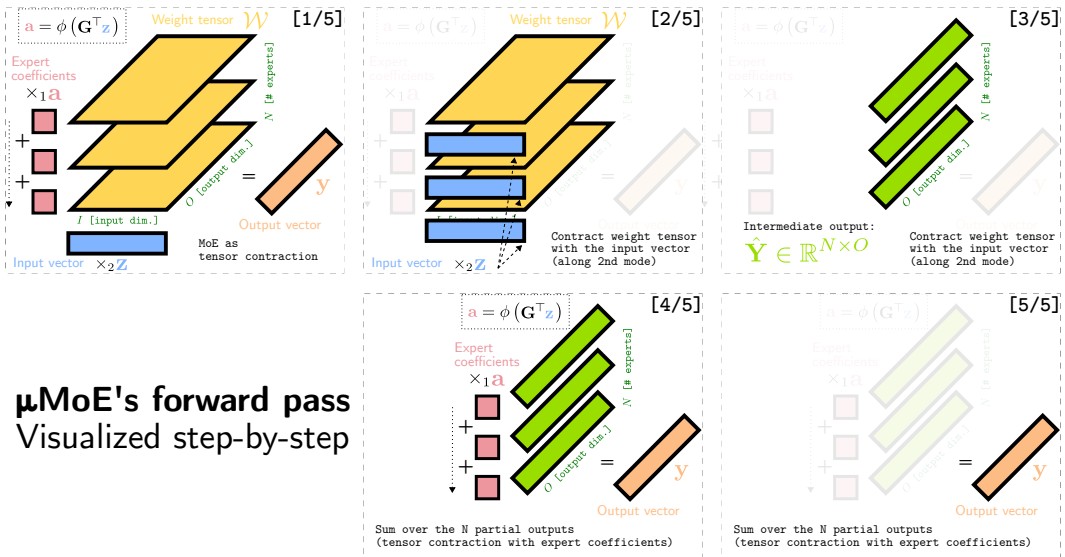

Figure 6: An intuitive visualization of the $\mu$MoE (unfactorized) forward pass, as visualized (as a series of tensor contractions) in 5 steps. Each step contributes to producing the output vector $\mathbf{y} \in \mathbb{R}^O$ either by contracting with the expert coefficients $\mathbf{a} \in \mathbb{R}^N$, or with the input vector $\mathbf{z} \in \mathbb{R}^I$, along the appropriate mode of the collective weight tensor $\mathcal{W} \in \mathbb{R}^{N \times I \times O}$.

## D Decomposition choice, matrix rank, and computational cost

In this section we present a further detailed discussion of decomposition choice, validating our choices and comparing alternative options. The computational costs of each fast $\mu$MoE forward pass and tensor-matrix rank relationships implications derived in this section are summarized in Table 5.

Table 5: A computational comparison of decomposition choice for $\mu$MoE layers and existing MoEs.

| | Param-efficient (medium $N$) | Param-efficient (large $N$) | # Parameters | Estimated # FLOPs | Max. expert matrix rank |
|---|---|---|---|---|---|
| Dense MoE | ☹ | ☹ | $NIO$ | $NIO$ | $\min\{I, O\}$ |
| Sparse MoE | ☹ | ☹ | $NIO$ | $KIO$ | $\min\{I, O\}$ |
| **CP$\mu$MoE** | ☺ | ☺ | $R(N + I + O)$ | $R(N + I + O)$ | $\min\{I, O, R\}$ |
| **TR$\mu$MoE** | ☺ | ☺ | $R_1 N R_2 + R_2 I R_3 + R_3 O R_1$ | $R_2 I R_3 + R_1 N R_2 + R_1 R_2 R_3 + R_1 O R_3$ | $\min\left\{R_3 \cdot \min\{R_1, R_2\}, I, O\right\}$ |

### D.1 Tensor ranks to matrix rank

One important consideration is how the chosen tensor ranks bound the resulting experts' matrix rank in $\mu$MoE layers. Here, we derive the matrix ranks as a function of tensor ranks for each model in turn.

### D.1.1 CP$\mu$MoEs: rank analysis

CP$\mu$MoEs are parameterized by factor matrices $\mathbf{U}^{(1)} \in \mathbb{R}^{R \times N}, \mathbf{U}^{(2)} \in \mathbb{R}^{R \times I}, \mathbf{U}^{(3)} \in \mathbb{R}^{R \times O}$ for chosen CP-rank $R$. Following *Section 3* of Kolda and Bader [53] which provides the matricization/unfolding of CP tensors, we can write expert $n$'s weight matrix as

$$\mathbf{W}_n = \mathbf{U}^{(2)^\top} \left( \mathbf{U}^{(1)^\top}_{:n} \odot \mathbf{U}^{(3)^\top} \right)^\top \in \mathbb{R}^{I \times O}, \tag{6}$$

where $\odot$ is the Khatri-Rao product [53], and $\mathbf{U}_{:n}^{(1)} \in \mathbb{R}^{R \times 1}$ is the column of the factor matrix associated with expert $n$ (including a singleton dimension for the Khatri-Rao product to be well-defined). Through the linear algebra rank inequality for matrix products, we have

$$\text{rank}(\mathbf{W}_n) = \text{rank}\left(\mathbf{U}^{(2)\top}\left(\mathbf{U}_{:n}^{(1)\top} \odot \mathbf{U}^{(3)\top}\right)^\top\right) \leq \min\left\{\text{rank}(\underbrace{\mathbf{U}^{(2)}}_{R \times I}), \text{rank}(\underbrace{\mathbf{U}_{:n}^{(1)\top} \odot \mathbf{U}^{(3)\top}}_{O \times R})\right\}. \tag{7}$$

Therefore a single CP$\mu$MoE's $n$th expert's matrix rank is bounded by $\min\{I, O, R\}$.

### D.1.2 TR$\mu$MoEs: rank analysis

We now turn our attention to TR$\mu$MoEs, where we will see that the TR ranks $R_1, R_2, R_3$ translate very favorably into matrix rank at smaller computational cost than with CP$\mu$MoEs. First recall that TR$\mu$MoEs are parameterized instead by core tensors $\mathcal{U}^{(1)} \in \mathbb{R}^{R_1 \times N \times R_2}, \mathcal{U}^{(2)} \in \mathbb{R}^{R_2 \times I \times R_3}$, $\mathcal{U}^{(3)} \in \mathbb{R}^{R_3 \times O \times R_1}$, with chosen ranks $R_1, R_2, R_3$. We can derive an expression to materialize expert $n$'s matrix through the sum of matrix products of the TR cores as:

$$\mathbf{W}_n = \sum_{r_3=1}^{R_3}\left(\underbrace{\mathbf{U}_{r_3::}^{(3)}}_{O \times R_1}\underbrace{\mathbf{U}_{:n:}^{(1)}}_{R_1 \times R_2}\underbrace{\mathbf{U}_{::r_3}^{(2)}}_{R_2 \times I}\right)^\top \in \mathbb{R}^{I \times O}. \tag{8}$$

The matrix product rank inequality applies to each $I \times O$ matrix summand, whilst the matrix sum rank inequality applies to the outer matrix sum:

$$\text{rank}(\mathbf{W}_n) = \text{rank}\left(\sum_{r_3=1}^{R_3}\left(\mathbf{U}_{r_3::}^{(3)}\mathbf{U}_{:n:}^{(1)}\mathbf{U}_{::r_3}^{(2)}\right)^\top\right) \tag{9}$$

$$\leq \sum_{r_3=1}^{R_3}\text{rank}\left(\left(\mathbf{U}_{r_3::}^{(3)}\mathbf{U}_{:n:}^{(1)}\mathbf{U}_{::r_3}^{(2)}\right)^\top\right) \tag{10}$$

$$\leq \sum_{r_3=1}^{R_3}\min\left\{\text{rank}\left(\mathbf{U}_{r_3::}^{(3)}\right), \text{rank}\left(\mathbf{U}_{:n:}^{(1)}\right), \text{rank}\left(\mathbf{U}_{::r_3}^{(2)}\right),\right\}. \tag{11}$$

Consequently, expert $n$'s materialized weight matrix in TR$\mu$MoEs has a more generous upper bound of $\min\left\{R_3 \cdot \min\{R_1, R_2\}, I, O\right\}$[8].

Through this analysis, we observe that one can choose large values of $R_3$ yet small $R_1, R_2$ to yield a high expert matrix rank with few parameters, justifying the choice of $R_1 = R_2 = 4$ in the main paper.

### D.1.3 Tucker$\mu$MoEs: rank analysis

One popular alternative decomposition is the Tucker decomposition [86]. Here we derive the resulting matrix rank of this alternative $\mu$MoE variant and detail why it's not as desirable as the proposed $\mu$MoE variants.

A Tucker$\mu$MoE composes an $\mu$MoE weight tensor through the series of mode-$n$ products [53]: $\mathcal{W} = \mathcal{Z} \times_1 \mathbf{U}^{(1)} \times_2 \mathbf{U}^{(2)} \times_3 \mathbf{U}^{(3)}$, where $\mathcal{Z} \in \mathbb{R}^{R_N \times R_I \times R_O}$ is the so-called 'core tensor' and $\mathbf{U}_1 \in \mathbb{R}^{N \times R_N}, \mathbf{U}_2 \in \mathbb{R}^{I \times R_I}, \mathbf{U}_3 \in \mathbb{R}^{O \times R_O}$ are the 'factor matrices' for the tensor's three modes.

Again following Kolda and Bader [53] a single expert $n$'s weight matrix can be rewritten through the matricization involving the Kronecker product $\otimes$ as:

$$\mathbf{W}_n = \mathbf{U}^{(2)}\mathbf{Z}_{(2)}\left(\mathbf{U}_n^{(1)} \otimes \mathbf{U}^{(3)}\right)^\top \in \mathbb{R}^{I \times O}, \tag{12}$$

---

[8]Regardless of how large $R_3$ is, the rank of the matrix cannot exceed $\min\{I, O\}$.

where $\mathbf{Z}_{(2)} \in \mathbb{R}^{R_I \times (R_O \cdot R_N)}$ is the so-called mode-2 (matrix) unfolding of the core tensor [53]. Consequently, the same rank inequality applies:

$$\text{rank}(\mathbf{W}_n) = \text{rank}\left( \mathbf{U}^{(2)}\mathbf{Z}_{(2)} \left( \mathbf{U}_n^{(1)} \otimes \mathbf{U}^{(3)} \right)^\top \right) \tag{13}$$

$$\leq \min\left\{ \text{rank}(\underbrace{\mathbf{U}^{(2)}}_{I \times R_I}), \text{rank}(\underbrace{\mathbf{Z}_{(2)}}_{R_I \times (R_O \cdot R_N)}), \text{rank}(\underbrace{\mathbf{U}_n^{(1)} \otimes \mathbf{U}^{(3)}}_{O \times (R_O \cdot R_N)}) \right\}, \tag{14}$$

Where we see the much more restrictive matrix rank upper bound applies: $\min\{\min(I, R_I), \min(R_I, R_O \cdot R_N), \min(O, R_O)\}$. Thus in practice, *both* $R_I, R_O$ need to be large to yield a large matrix rank, which is in conflict with the goal of maintaining a moderate number of parameters.

### D.2  Why is low-rankness a reasonable assumption?

Given we've seen that parameter-efficient $\mu$MoE layers lead to low-rank expert weight matrices, a natural question is whether or not low-rankness in MLP linear layers' weight matrices is a reasonable assumption or constraint.

Our strongest piece of evidence supporting the claim is experimental in nature: we've seen from the results in Section 4.3 that using all parameter-matched $\mu$MoE layers for both MLP mixers and GPT-2 models leads to no significant drop in accuracy from their linear layer counterparts (see also Appendix I for many more results).

To investigate this further we perform a rank ablation on our trained MLP-Mixer model with the original linear layers' weights. Concretely, we compute the truncated SVD of each MLP block's 2 linear layer weight matrices. We explore the impact on the model's ImageNET1k validation set accuracy when using only the top-$k$ singular vectors/values (the best rank-$k$ approximation [87]). The validation set accuracy using truncated SVD weights in every mixer block is plotted in Figure 7–we see here that discarding as many as *half* the total number of (bottom) singular vectors/values to approximate the original weights

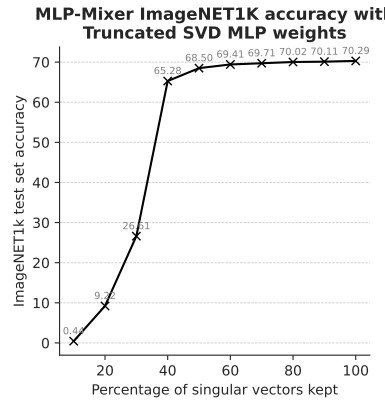

Figure 7: Val. accuracy for an S-16 MLP-mixer when performing truncated SVD on all MLP's linear layers' weight; model accuracy is closely retained even with half the singular vectors.

leads to negligible difference to the validation set accuracy. In other words, low-rank approximations of MLP Mixers' weights retain their representational power sufficiently well to produce nearly the same validation set accuracy as the original model. Such findings are consistent with results in recent work in the language domain [88], where low-rank approximations of MLP layers can even sometimes boost original performance. The accuracy retained by MLP Mixers here even after such aggressive rank reduction constitutes further evidence that full-rank weights are not always necessary.

### D.3  MoE/$\mu$MoE parameter count comparisons

We plot in Figure 8 the parameter counts for $\mu$MoE layers as a function of the expert counts (sweeping from $N = 2$ experts through to $N = 16,384$), relative to dense/sparse MoEs (with rank $R_1 = R_2 = 4$ TR$\mu$MoEs), for the first layer in a MLP-mixer channel-mixing block [80]. As can be seen, both $\mu$MoE variants are vastly more parameter-efficient than dense/sparse MoEs.

Given TR$\mu$MoEs offer even better parameter efficiency for larger numbers of experts, we suggest opting for CP$\mu$MoEs when using expert counts less than $\sim 128$, and considering TR$\mu$MoEs for higher values.

**Latency and memory usage**  comparisons between the $\mu$MoE, linear layers, and alternative MoEs are shown in Table 6, where the $\mu$MoEs perform favorably.

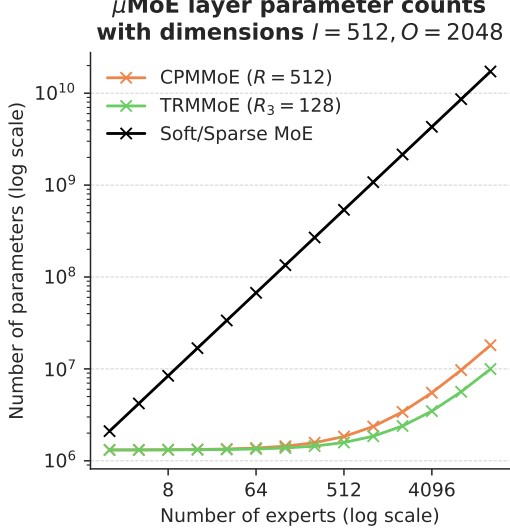

Figure 8: $\mu$MoE layer parameter count as a function of expert count.

Table 6: Comparison of different layers' peak memory usage and latency (per single input). We use 128 experts in each MoE layer, and set the rank of the $\mu$MoEs to parameter-match that of the linear layer.

| Layer type | Peak memory usage (MB) | Latency per single input (ms) |
| --- | --- | --- |
| Linear layer | 12.07 | 0.01 |
| Dense MoE ($N = 128$) | 390.17 | 1.17 |
| Sparse MoE ($N = 128$) | 765.19 | 0.80 |
| TR$\mu$MoE ($N = 128$) | 15.87 | 0.94 |
| CP$\mu$MoE ($N = 128$) | 14.02 | 1.05 |

# E   Hierarchical $\mu$MoE model derivations

In the main paper, the fast forward passes are derived for a single level of expert hierarchy. One additional attractive property of $\mu$MoEs is their straightforward extension to multiple levels of expert hierarchy–one simply increments the number of modes of the weight tensor and includes another tensor contraction with new expert coefficients. Hierarchical $\mu$MoEs intuitively implement "and" operators in expert selection at each level, and further provide a mechanism through which to increase the total expert count at a small parameter cost. Here, we derive the fast forward passes for $\mu$MoE layers in their most general form with $E$ levels of expert hierarchy. For intuition, we first further visualize $\mu$MoE layers with 2 levels of hierarchy in Figure 9–note how we have an extra mode to the weight tensor, and an extra contraction over the new expert mode to combine its outputs.

Given that hierarchical $\mu$MoEs involve very high-order tensors, we adopt the popular mode-$n$ product [53] to express the forward passes in as readable a way as possible. The **mode-$n$ (vector) product** of a tensor $\mathcal{X} \in \mathbb{R}^{I_1 \times I_2 \times \ldots \times I_N}$ and vector $\mathbf{u} \in \mathbb{R}^{I_n}$ is denoted by $\mathcal{X} \times_n \mathbf{u}$ [53], with its elements given by:

$$(\mathcal{X} \times_n \mathbf{u})_{i_1 \ldots i_{n-1} i_{n+1} \ldots i_N} = \sum_{i_n=1}^{I_n} x_{i_1 i_2 \ldots i_N} u_{i_n}.$$

We first introduce the formulation of an $E$-level hierarchical $\mu$MoE layer from Equation (1) in the main paper: given input $\mathbf{z} \in \mathbb{R}^I$, the most general form of $\mu$MoE layer is parameterized by weight tensor $\mathcal{W} \in \mathbb{R}^{N_1 \times \ldots \times N_E \times I \times O}$ and $E$ many expert gating parameters $\{\mathbf{G}_e \in \mathbb{R}^{I \times N_e}\}_{e=1}^{E}$. The

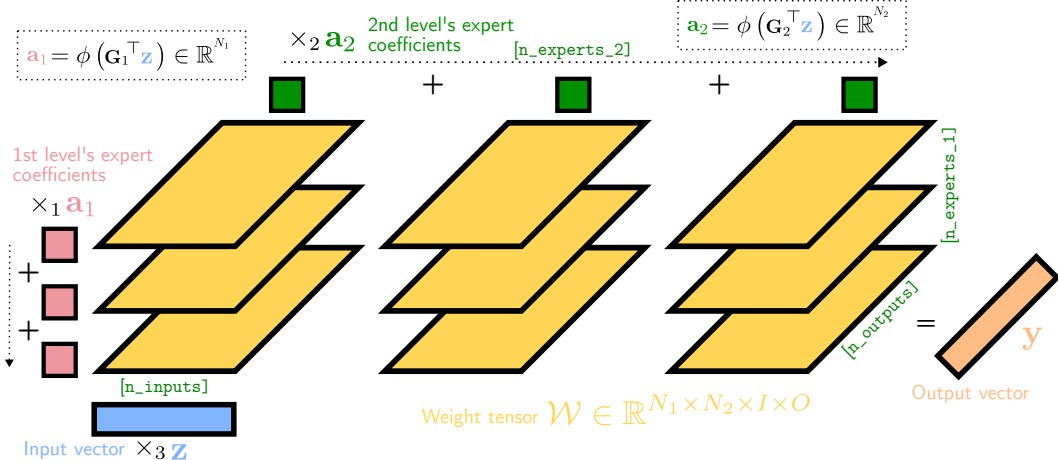

Figure 9: Illustration of a **two-hierarchy** $\mu$MoE layer's (unfactorized) forward pass as a series of tensor contractions. The $N_1 \cdot N_2$ many experts' weight matrices are visualized as 2D horizontal slices in yellow, which are (1) matrix-multiplied with the input vector, (2) summed over the first expert mode (weighted by the first expert coefficients $\mathbf{a}_1$ in red), and (3) summed over the second expert mode (weighted by the second expert mode's coefficients $\mathbf{a}_2$ in dark green).

explicit, unfactorized forward pass is given by:

$$\mathbf{a}_e = \phi(\mathbf{G}_e^\top \mathbf{z}) \in \mathbb{R}^{N_e}, \quad \forall e \in \{1, \dots, E\},$$

$$\mathbf{y} = \mathcal{W} \times_1 \mathbf{a}_1 \times_2 \dots \times_E \mathbf{a}_E \times_{E+1} \mathbf{z}$$

$$= \sum_{n_1=1}^{N_1} a_{1n_1} \cdots \sum_{n_E=1}^{N_E} a_{E n_E} \big( \underbrace{\mathbf{W}_{n_1 \dots n_E ::}^\top}_{O \times I} \mathbf{z} \big) \in \mathbb{R}^O, \quad (15)$$

where Equation (15) is expressed as sums over the $E$-many expert modes to make it clear that hierarchical $\mu$MoEs take convex combinations of $\prod_{e=1}^{E} N_e$ many experts' outputs (given there are $N_e$ experts at each level of hierarchy). With expert coefficients $\{\mathbf{a}_e \in \mathbb{R}^{N_e}\}_{e=1}^{E}$, the factorized forward passes of the most general hierarchical $\mu$MoE layers are given for the two variants below.

### E.1 Hierarchical CP$\mu$MoE

The full CP$\mu$MoE model of rank $R$ has an implicit weight tensor $\mathcal{W} = \sum_{r=1}^{R} \mathbf{u}_r^{(1)} \circ \mathbf{u}_r^{(2)} \circ \mathbf{u}_r^{(3)} \circ \cdots \circ \mathbf{u}_r^{(E+2)} \in \mathbb{R}^{N_1 \times \cdots \times N_E \times I \times O}$, with factor matrices $\mathbf{U}^{(1)} \in \mathbb{R}^{R \times N_1}, \dots, \mathbf{U}^{(E)} \in \mathbb{R}^{R \times N_E}, \mathbf{U}^{(E+1)} \in \mathbb{R}^{R \times I}, \mathbf{U}^{(E+2)} \in \mathbb{R}^{R \times O}$. The implicit, factorized forward pass is given by:

$$\mathbf{y} = \left( \sum_{r=1}^{R} \mathbf{u}_r^{(1)} \circ \mathbf{u}_r^{(2)} \circ \mathbf{u}_r^{(3)} \circ \cdots \circ \mathbf{u}_r^{(E+2)} \right) \times_1 \mathbf{a}_1 \times_2 \dots \times_E \mathbf{a}_E \times_{E+1} \mathbf{z}$$

$$= \sum_{r=1}^{R} \mathbf{u}_r^{(E+2)} \big( \sum_{n_1, \dots, n_E, i} u_{rn_1}^{(1)} a_{1_{n_1}} \cdots u_{rn_E}^{(E)} a_{E_{n_E}} u_{ri}^{(E+1)} z_i \big)$$

$$= \sum_{r=1}^{R} \mathbf{u}_r^{(E+2)} \big( \mathbf{U}^{(1)} \mathbf{a}_1 \big)_r \cdots \big( \mathbf{U}^{(E)} \mathbf{a}_E \big)_r \cdot \big( \mathbf{U}^{(E+1)} \mathbf{z} \big)_r \in \mathbb{R}^O. \quad (16)$$

### E.2 Hierarchical TR$\mu$MoE

In TR format, $\mathcal{W} \in \mathbb{R}^{N_1 \times \cdots \times N_E \times I \times O}$ has $E+2$ factor tensors: $\mathcal{U}^{(1)} \in \mathbb{R}^{R_1 \times N_1 \times R_2}, \dots, \mathcal{U}^{(E)} \in \mathbb{R}^{R_E \times N_E \times R_{E+1}}, \mathcal{U}^{(E+1)} \in \mathbb{R}^{R_{E+1} \times I \times R_{E+2}}, \mathcal{U}^{(E+2)} \in \mathbb{R}^{R_{E+2} \times O \times R_1}$, where $R_i$ are the manually chosen ranks. The weight tensor's elements are given by:

$$w_{n_1 \dots n_E i o} = \text{tr}\big( \mathbf{U}_{:n_1:}^{(1)} \cdots \mathbf{U}_{:n_E:}^{(E)} \mathbf{U}_{:i:}^{(E+1)} \mathbf{U}_{:o:}^{(E+2)} \big).$$

We derive the fast factorized forward pass in terms of a series of mode-2 products:

$$\mathbf{y} = \sum_{i} \sum_{n_1,\ldots n_E} \mathcal{W}(n_1, \cdots, n_E, i, :) \mathbf{a}_1(n_1) \cdots \mathbf{a}_E(n_E) \mathbf{z}(i) \tag{17}$$

$$= \sum_{r_1, r_{E+2}} \mathbf{u}^{(E+2)}_{r_{E+2}:r_1} \big( \underbrace{(\mathcal{U}^{(1)} \times_2 \mathbf{a}_1) \cdots (\mathcal{U}^{(E)} \times_2 \mathbf{a}_E)(\mathcal{U}^{(E+1)} \times_2 \mathbf{z})}_{R_1 \times R_{E+2}} \big)_{r_1 r_{E+2}} \in \mathbb{R}^O. \tag{18}$$

# F  Experimental details

## F.1  Network configurations and hyperparamters

Here we provide the full experimental details and setups to reproduce the performance results in the paper for each of the networks. We further include the per-epoch accuracy plots for additional transparency into the training processes.

The experimental configurations used to reproduce the performance results in the main paper follow as closely as possible those specified in the main paper of MLP-mixer [80] and open-source code (`https://github.com/lucidrains/mlp-mixer-pytorch`), the open-source code for NanoGPT (`https://github.com/karpathy/nanoGPT`) for GPT2 [81], and the robust fine-tuning protocol of [89] for CLIP [61]. These values are summarized in Table 7. We plot the learning curves for the training of both models in Figures 10 and 11.

Table 7: Experimental configuration and settings for the results reported in the main paper in Section 4.3.

|  | Learning rate | Batch size | Weight decay | Warmup steps | Training duration | Stochastic depth | RandAugment strength | Dropout | Mixup strength | Mixed precision | Random seed | Hardware |
|---|---|---|---|---|---|---|---|---|---|---|---|---|
| MLP Mixer | 1e-3 | 4096 | 1e-4 | 10k | 300 epochs | True | 15 | 0 | 0.5 | bf16 | 0 | 4xA100 80GB |
| NanoGPT | 6e-4 | 24 | 1e-1 | 2k | 100k iter. | False | 0 | 0 | 0 | fp16 | 0 | 4xA100 80GB |
| CLIP | 3e-5 | 4096 | 1e-1 | 500 | 10 epochs | False | 0 | 0 | 0 | fp16 | 0 | 1xA100 80GB |

**Rank choices**   Throughout all experiments in the main paper, we fix the TR$\mu$MoE ranks for the first two modes to be $R_1 = R_2 = 4$. This way, we can maximize the effective expert matrix ranks at a low parameter cost, as shown in Appendix D.1.2. The final TR rank $R_3$ is varied to parameter-match the networks in question. For CP$\mu$MoEs, we set the single CP rank $R$ to parameter-match the baselines.

**Training times**   Each MLP mixer model takes just under 3 days to train on 4xA100 80GB GPUs. The NanoGPT models take 2-3 days to train for $100k$ iterations, with the same resources.

## F.2  Weight initialization

We initialize each element of the factor matrices/tensors for the input and output modes from a $U[-\sqrt{k}, \sqrt{k}]$ distribution (following PyTorch's linear layers' initialization strategy), for $k = 1/\text{in\_features}$, where in\_features is the dimension of the input to each factor matrix/tensor during the factorized forward passes.

Factor matrices for the expert modes are initialized to replicate the weight matrices along the expert mode (plus optional noise). For CP$\mu$MoEs, this corresponds to sampling the factor matrices' elements from a $\mathcal{N}(1, \sigma)$ distribution. For TR$\mu$MoEs, the weight matrices can instead be replicated along the expert mode by initializing each slice (e.g. $\mathcal{G}_1(:, i, :)$) as a diagonal matrix with its elements sampled from $\mathcal{N}(1, \sigma)$. In all our experiments we set $\sigma := 1$ to introduce noise along the first expert mode, and $\sigma := 0$ for additional expert modes.

# G  Expert specialism: additional results

## G.1  Large scale models

We first show in Figure 12 the top-activating examples for MLP-mixers trained with both CP$\mu$MoE and TR$\mu$MoE blocks. Examples are shown for the first two experts as they appear numerically for

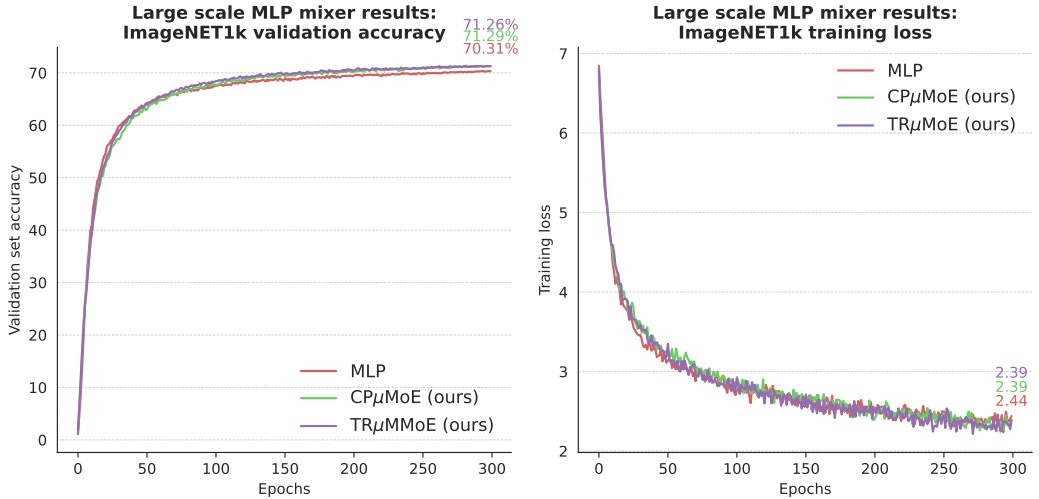

Figure 10: Training loss and validation accuracy for the MLP-mixers models for 300 epochs.

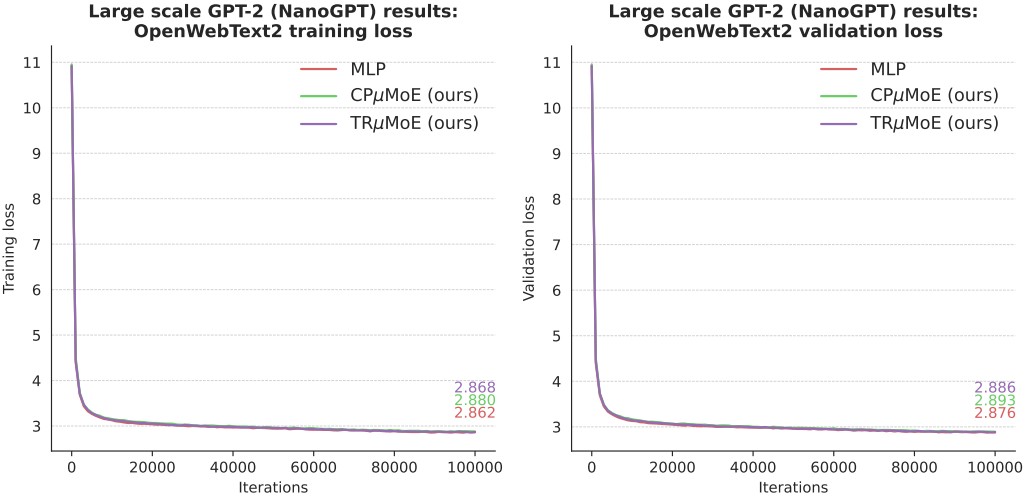

Figure 11: Training and validation loss for the GPT-2 models for 100k iterations.

each of the 8 layers, where we observe the same phenomenon of earlier blocks specializing to textures, and later blocks to higher-level abstract concepts/objects.

Secondly, in Figure 13 we show the top 32 activating tokens for the first 6 experts (as they appear numerically) for layer 5 in GPT2 models trained with CP$\mu$MoEs replacing every MLP block. Whilst there are clear coherent themes amongst the top-activating tokens, we do see some examples of multiple themes being processed with high coefficients by the same experts (e.g. example #20 in expert 2's top-activating examples appears unrelated to the context of the other top-activating tokens) indicating a certain degree of expert polysemanticity (as expected in the large open domain of web text).

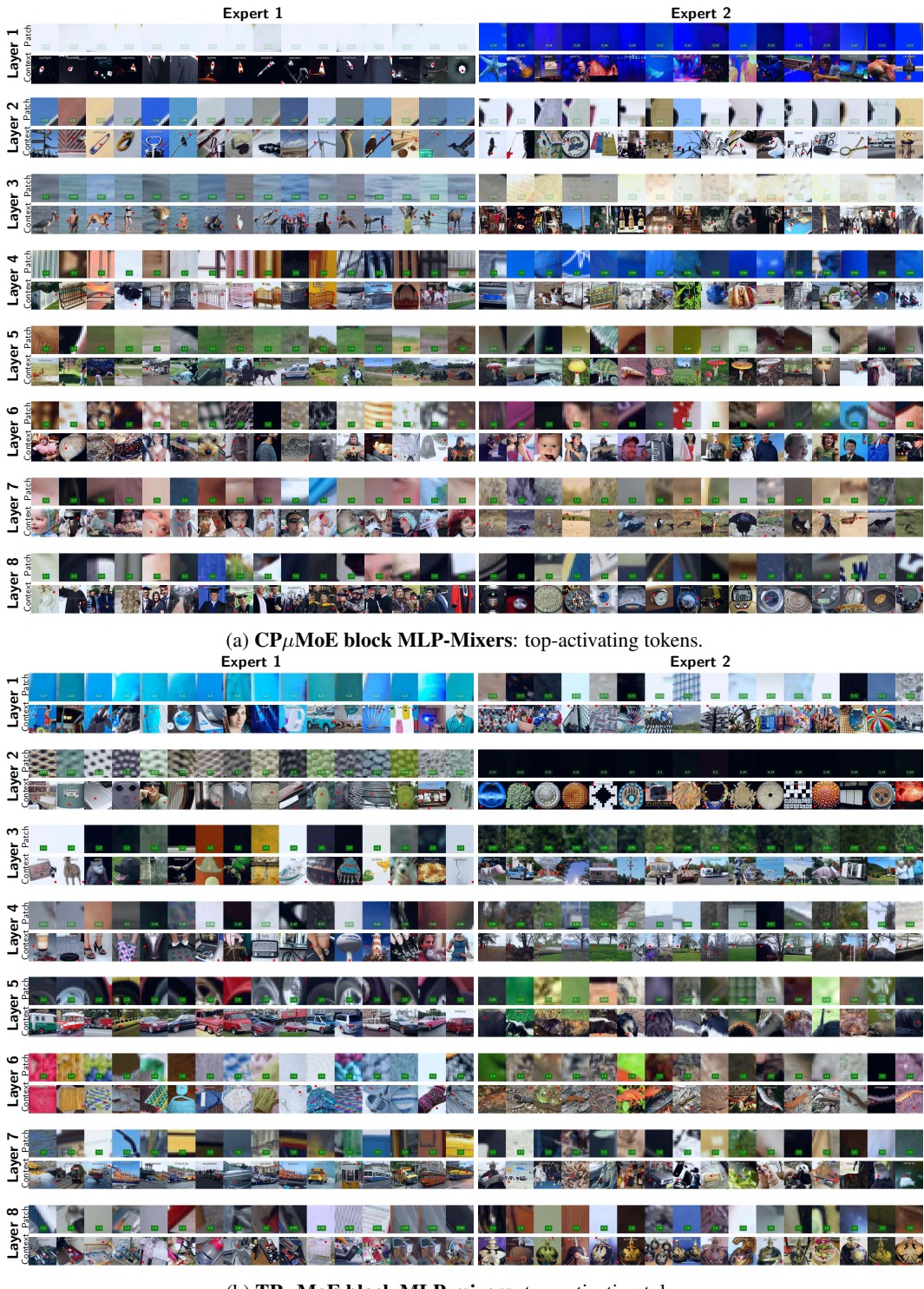

(a) **CP$\mu$MoE block MLP-Mixers**: top-activating tokens.

(b) **TR$\mu$MoE block MLP-mixers**: top-activating tokens.

Figure 12: Top-activating patches (and their surrounding image context) for the first experts at two blocks in MLP-mixer models. $\mu$MoE blocks (with $N = 64$) exhibit coarse-grained specialism (e.g., texture) earlier and more fine-grained specialism (e.g., object category) deeper in the network.

**Expert coefficients color map:**

0.0 0.1 0.2 0.3 0.4 0.5 0.6 0.7 0.8 0.9 1.0

## Layer 5, Expert 1

**#1**: best of our knowledge, this is one of the most significant and insightful book. I recommend it to
**#2**: the last 10 years, we at VICE are the most well-traveled
**#3**: national security of the United States remains one of the most hotly debated topics in Congress,\n\nSen
**#4**: and maintenance of the country's largest and most complex military facilities.\n\nFederal officials
**#5**: IISS) said on Tuesday.\n\nThe most recent estimates estimate that the global economy will grow
**#6**: \nThe best-selling authors of the most popular fantasy books in the Middle Ages, the
**#7**: . 6.1. 13\n\nIn the most recent installment, the Bears' receivers have become
**#8**: . Today, Bionicle is one of the most highly regarded and accomplished companies in the world with
**#9**: \n\nRelated Stories: \n\nThe most difficult thing to overcome is the fear of punishment
**#10**: Good".\n\n- One of the most watched events of 2017.\n\n- All
**#11**: to Go The Next Level\n\nOne of the most important skills you have to master is balance,
**#12**: an estimated 4,000 spectators, this is the most crowded and most exciting marathon in the world.
**#13**: \n\nAnd today, one of the most interesting questions the company asked on Twitter was whether
**#14**: her career, and in doing so she became the most famous woman in female-led media.\n
**#15**: three methods to search for the new species, the most common method being fossil-bearing excavations.
**#16**: Human Interface Guidelines describe as 'the best and most widely used rendering engine for the Web.
**#17**: by The Smiths and it's the most important single of their career, "Bad
**#18**: are living through the hottest, driest, and most dangerous part of the year,
**#19**: than a top-flight soldier. Even during the most crucial moments, Sarcastic was a show
**#20**: \n\n"The United States is one of the most politically and economically influential countries in the world,
**#21**: .\n\nThe United States and Russia are the most powerful economic, political, and military power in
**#22**: BCHL's Boston College was the most successful of the lower divisions in the league.
**#23**: \nAt the end of 2016, one of the most prominent and talked about aspects of the U.
**#24**: just because I was one of the luckiest and most wonderful ones, but because I was one of
**#25**: \n\nIn one of his most recent pieces, I'll do a look at
**#26**: .\n\nThe UK is now one of the most prosperous countries in the world. It is better
**#27**: which was unveiled for the launch of the year's most popular VR game, Psychonauts, has
**#28**: , 5-12 and 5-10 are the most common plays that both teams are able to control
**#29**: hiwei. At the time it was also the most advanced nuclear reactor in China. As of 2010
**#30**: into conflict, The Chinese navy is one of the most formidable in the world, and this year is
**#31**: \nDr, Aaron Ben-Gurion, the most senior Israeli human rights lawyer, told the Wall
**#32**: case or his parents as it was one of the most significant topics of the day, with the school

## Layer 5, Expert 2

**#1**: Afghan soldiers and Afghan police, but with the fall of NATO in Afghanistan, the U
**#2**: warned that there would be no peace after the fall of the Soviet Union, as the Soviet Union had
**#3**: \n10/22/16\n\nRise of the Sub-Prime\n\nBryan Hitch
**#4**: The Empire Strikes Back' and 'Return of the Jedi'.\n\nThe game
**#5**: tripled its number of airstrikes in Iraq since the fall of Mosul.\n\nThe United States has two
**#6**: Nighter, The Singing Star, The Return of the Jedi, The Force Awakens, The Force
**#7**: to 52,400.\n\nSince the fall of the Soviet Union, the number of American workers
**#8**: Kremlin has seen a change in attitude following the fall of the military in Georgia. Vladimir Putin is now
**#9**: Machine." In: "The Fall of Jim Crow." In: "
**#10**: very stable in terms of its military since the fall of communism and even though it's a
**#11**: land has had a moment, or since the fall of the Berlin Wall, It is a memory of
**#12**: 's activities are alleged to have continued after the fall of the Soviet Union, and have been monitored by
**#13**: %\n\nStar Wars: Episode II - Return of the Jedi: 0%
**#14**: \nIn the late 1980s, after the fall of the Soviet Union, the government began to issue
**#15**: be confronted with a conventional military threat since the fall of the Soviet Union in 1991," the
**#16**: \n\nAll soldiers in the army after the fall of the Berlin wall were Germans. The Germans came
**#17**: the-scenes in Star Wars Episode VI: Return of the Jedi.
**#18**: Oz" (2013)\n\n"The Return of Oz" (1999)\n\n(2015
**#19**: \n"With the fall of the Soviet Union, the United States'
**#20**: \nThe three-year-old game was up next, with the Bulldogs emerging as winners on the
**#21**: Army, which was formed in 2013 after the fall of the Soviet Union, The Free Syrian Army is
**#22**: \n\n"After the fall of the Taliban, it's very important
**#23**: The family had stored items there since before the fall of Aleppo.\n\nThe family issued a statement
**#24**: to a huge crowd of protesters, chanting "Down with the Palestinians" as they watch. Amongst
**#25**: through a string of severe economic crises since the fall of the Berlin Wall, and the World War II
**#26**: in the country's biggest urban conflict since the fall of the Soviet Union.\n\nThey include the
**#27**: rebuilt during the Tamil Munra period following the fall of the BritishLA and after its destruction in the
**#28**: \nSince the fall of the Berlin Wall in 1989, Germany has seen
**#29**: \nSince the fall of the Berlin Wall, the Berlin elite, along
**#30**: ird: It's the first time since the fall of 2002 that an "old friend" of mine
**#31**: " from "Hitch House 2: Rise of the Elders," starring Tom Hardy
**#32**: of the Syrian government, in 2012 after the fall of Assad's regime.\n\nHe said that

## Layer 5, Expert 3

**#1**: on how to operate in the marketplace but at the same time I'm still waiting for them
**#2**: very depressed at first,"\n\nAt first, but eventually, he was able to self
**#3**: \n\nOn its face, the campaign to close that loophole should be
**#4**: at face value. The only real difference, at first sight, is that the "collo
**#5**: t know what to do anymore.\n\nAt first we just wanted to buy stocks, but then
**#6**: for my friends (who, let's face it, love PB).\n\nSo I
**#7**: s just so exciting,"\n\nAt first, the protests appear to be peaceful, But
**#8**: the event show the stars of the future, at first appearing as a dragon shape or nigh-
**#9**: \n1. Pick up the pencil\n\nAt first, you should strike the pencil with your fingers
**#10**: getting out of the car are intense and unexpected at best even for a man who is normally a casual
**#11**: of the two studies, the study is misleading at best," the article said. "The conclusion is
**#12**: Republican voters will vote for the Republican nominee. At first, this will not be the case. It
**#13**: \n\nOn the surface, there are no significant differences between the two
**#14**: It's the kind of idea that at first might seem a little odd to a fan of
**#15**: is currently being built in the California desert. At first, the telescope's equipment was set up as
**#16**: \nThe initial response from the community was mixed at first, The group of students that had been invited
**#17**: \n\nAt first glance it seems that the first half of the
**#18**: archived and updated to reflect the changes.<[endoftext]>At first, it looks like the American government is cracking
**#19**: blog, it's a book. At first it seems like a really good book, but
**#20**: really aware of what this album is about. At first, it's an album about death
**#21**: reference to the method in question.\n\nAt first glance, this looks reasonable until you realize that
**#22**: was for being with him.\n\nOn the surface, it seems like she wanted to be with
**#23**: on the problems will go down, but at the same time, it's going to look good.
**#24**: , they are working for one company. But at first, they are working for the same company,
**#25**: loading it at the same time,\n\nAt first I only thought that my friend was dead,
**#26**: want to play a chess game,\n\nAt first I don't know what I'm losing.
**#27**: kind of hard to get a meeting with somebody at first just to get clarification, but it'
**#28**: , tells BBC News: "I'd say at first I didn't know what was happening with the
**#29**: that.'"\n\nOn the surface, there was nothing wrong with a new player
**#30**: will be given in the post: \n\nAt first, our project is to deliver this experience to
**#31**: for men's clothing.\n\nAt first, they asked me if I thought they had
**#32**: No player was suspended more than he was. At first, he got his suspension for a month and

## Layer 5, Expert 4

**#1**: lot about it," Reiser said. "It's very important to know
**#2**: ra Karam. He also accused Israeli forces of "perpetuating an apartheid system that harms
**#3**: s "misogyny" remark was "factually false."\n\n
**#4**: for the United States," he said. "It basically gives the State Department and the
**#5**: public debate about the right to privacy and what constitutes "privacy." Since the 2000
**#6**: remain in the lineup," Gibson says. "The players around me are trying to help
**#7**: " Deason said of the new project. "It's not a gimmick that
**#8**: move to do it," he said. "Hopefully, I will be able to play
**#9**: 're doing," he said. "They have the right to make allegations against
**#10**: \n\nAs if the United States is not being "disappointed" by the people,
**#11**: element to the game," he said. "If you don't have the
**#12**: system like no other," he said. "But this is the United States of America
**#13**: neaky" story about a successful entrepreneur who "has no trouble getting promoted" is
**#14**: for the right reason," he said. "We're doing it for the
**#15**: the world," Klinsmann said. "We know that. Our goal is to
**#16**: release for the book," he said. "We have a publisher now who sees it
**#17**: lahoma, a member of the House Intelligence Committee. "We have to get that hard,
**#18**: jury found that her work as a prison specialist was "excessive and that she should not be
**#19**: of a long week," he said. "The first team doesn't have
**#20**: good piece of work," he said. "I don't really like the
**#21**: ("a liar!") and "sucking the Earth".\n
**#22**: Sept. 8 testimony before the Senate Judiciary Committee. "We are not going to be as cautious
**#23**: evtek said the committee's decision was "a great mistake," adding:
**#24**: said that men are more likely to call their friends "girlfriend," but only 10 percent
**#25**: " he said, adding that the North has "proved that China is a threat to
**#26**: in the United States," he said. "I do like it as a matter of
**#27**: is described as a "radical" and "curious" writer, who has
**#28**: 's for sure," he said. "But we need to know what happened before
**#29**: , who said that the Occupy Wall Street movement was "fundamentally about the power of the
**#30**: " of the initiative and allow the scheme to "provide a platform for experimentation,
**#31**: to show the potential of the technology. Sooner or later, we would see what the future would
**#32**: the police and the justice system, which she called "a system of intimidation against the public

## Layer 5, Expert 5

**#1**: make its trademarks more exclusive, and therefore more valuable, by making them more scarce and worthless.\n
**#2**: working alongside the European Commission to develop, and implement, the regulations.\n\nThe Commission will also
**#3**: a general outline of, and also a reference on, the subject material. The subject division is then
**#4**: on the brink of a new, and very dangerous, world. It's about to change.\n
**#5**: be receiving from others is not, and is not, true in all cases. The things you are
**#6**: confers a protective, and sometimes even a fatal, protective effect on the brain.\n\nThe
**#7**: new law will give me, and other gun owners, as I watch this debate unfold, and I
**#8**: night a week with, and have a relationship with, a man named Tony," he added
**#9**: Department of State have provided, and are still developing, proposals for a response to the court's request
**#10**: be noted that the first, but not the last, is a statement of fact by the Irish-
**#11**: federal government to monitor the work on, and for, the Arctic.\n\nOther states have developed
**#12**: of the most important, if not the most important, things that have happened in the past 100 years
**#13**: person shooter mode similar to, but not identical to, Elite. And it's not far behind Elite
**#14**: blood that is analogous to, and specifically correlated with, increased stress-related cortisol levels in young healthy
**#15**: alment of your first, but hopefully not final, interview for a future job in an engineering department
**#16**: .\n\nMany of the smaller, but powerful, battery-powered and mobile phone screens are built
**#17**: have to accept an increasingly authoritarian, and increasingly hostile, government in place in order to survive,\n
**#18**: Obama administration has then turned the first, and only, rule on Iran. The Obama administration has refused
**#19**: have been able to deliver our first, and only, launch of the software," says Mass
**#20**: also shows the central government has undertaken, and taken, unprecedented steps to tackle the flow of energy,
**#21**: caused by a combination of, and the symptoms of, psychosis. It is associated with major mental injury
**#22**: After this, the best, if not the best, of the last 30 years is exactly what we
**#23**: as the central enforcer of, and advocate for, the Obama administration's policies on the
**#24**: er and for having committed, and only admitted to, a fraud of more than $1 million,"
**#25**: is to test other more common, but less common, varieties of Bumblebee, such as Pr
**#26**: final day being held in the small, but beautiful, town of Wroclaw, in the south
**#27**: e-mails from, and could have obtained from, "a group of industry insiders that have been
**#28**: says the country can still hear, and even see, the voices of the poor and the homeless.
**#29**: past few months to help revitalize, and improve, the laws of the Obama administration. Its
**#30**: contrast, the Liberals have retained, and may retain, their incumbency advantage from the late-term
**#31**: ." It did not indicate when, or even if, such a coordinated response would be contemplated.\n
**#32**: school system has been under, and is still under, the burden of the County of Oakland and Oakland

## Layer 5, Expert 6

**#1**: the court order to review their own data, Ufford said.\n\nA second public hearing
**#2**: pilots to return safely to their jobs, Mr. Jaffe added.\n\n"It
**#3**: long time to catch up."\n\nNewdow said he hopes to see the Legislature
**#4**: , but it has yet to be enforced, Chiappetta said.\n\nA spokesman for
**#5**: to learn how to be involved."\n\nRains said he worked hard for his student-ath
**#6**: that has characterized much of their political careers, Tien said. "
**#7**: not being contemplated at this stage,"\n\nStathopoulis, a native of Hain
**#8**: for the Democrat-controlled Senate,\n\nMiringoff warned the GOP could
**#9**: pass — has been estimated as $40 million, St. John said.\n\nBut the city
**#10**: what we ended up with."\n\nPollack performed the studies and, in 2009,
**#11**: great," she says.\n\nPugh agrees. "I'm working
**#12**: of a strategy," Mr. Riegle said.\n\n"The fact that
**#13**: should use one."\n\nMr. J. Wilson said that while some economists have said
**#14**: out what the right thing is."\n\nBhattarai has been doing real estate work at
**#15**: good for workers or bad for workers, Gagas says. But with the recent
**#16**: the actions of a terrorist."\n\nBut Camps said it's not just the military that is
**#17**: , would prefer that unions pay their dues, DeMarco said, but if public-sector unions don
**#18**: affected by the past month,"\n\nAl-Khelo
**#19**: the video of the police shooting."\n\nFerrero said some of the details in the video are
**#20**: at what's happening in the Senate," Boucaudelle said. "They're going to have
**#21**: to include the most outrageous provisions," Toner said. "That makes me very nervous
**#22**: to replace the aging guardrail.\n\nKnecht said the Guardico case has also left
**#23**: Geoffrey Rains on December 4.\n\nRains, who had been in the country for two
**#24**: would be running for president."\n\nPileggi said Trump's comments about people with disabilities
**#25**: public benefit to being the mayor."\n\nFretwell says the report is a warning to other
**#26**: perspective, and it's not good"\n\nHeymann said that even as the climate is changing
**#27**: be represented in the justice system,"\n\nStauffer said she is concerned about police policies
**#28**: the polls and solve our nation's problems," Boddicker said. "We need to be fighting
**#29**: true to ourselves."\n\nMs Rios was working at an American embassy in Havana when
**#30**: who specializes in immigration.\n\nRepublicans, Daugaard said
**#31**: get everything turned around."\n\nPugh said it was "impossible"
**#32**: \n\nIn his comments, Pugh said: "It's not just me that

Figure 13: Top-activating 32 tokens for the first unfiltered experts 1-6 (as ordered numerically) at layer 5 in the CP$\mu$MoE GPT2 model (**Please find the next 6 experts in Figure 14**).

## Layer 5, Expert 7

#1: /T, J, M, for the BBC\n\nWASHINGTON (AP
#2: \n\n(AP Photo/Jim Cole] (AP)\n\nWASHINGTON — The Obama
#3: May. (Melina Mara/The Washington Post)\n\nThe number of people employed in manufacturing
#4: , Fla, (David Campion/Getty Images)\n\nNEW YORK (AP) — A
#5: photo, (AP Photo/Jacquelyn Martin)\n\nWASHINGTON (AP) — The U
#6: AP Photo/Manuel Balce Ceneta]\n\nThe political left and the ideological right
#7: , (AP Photo/J. Scott Applewhite)\n\nSen. Ted Cruz (R-
#8: (Reuters / Jeffrey Sinis for the Washington Post\n\n
#9: Thursday. (Matt McClain/The Washington Post)\n\nIn the wake of the shooting of
#10: \n\n(AP Photo/Matt McClain] A man walks into the bathroom of a bank
#11: Vote," (AP Photo/Carolyn Kaster)\n\nThe Clintons' re-election have
#12: (Noah Gararaff/The Washington Post)\n\nYou can almost hear the conversation,
#13: (Gillian Brockell/The Washington Post)\n\nPresident Trump and Russian President Vladimir Putin
#14: Donald Trump, (Evan Vucci/AP)\n\nThe president of the University of Wisconsin
#15: AP Photo/Manuel Balce Ceneta]\n\n(CNSNews.com)
#16: (AP Photo/Timothy D. Easley]\n\nThe Trump campaign has abandoned the endorsement
#17: AP Photo/Saul Pastrana, File]\n\nBy Alison Rehm, The Associated
#18: 2016. (AP Photo/Charlie Neibergall)\n\nRepublican presidential candidates have stonewalled
#19: \n\n(AP Photo/Richard Drew]\n\nToday's news update is
#20: (AP Photo/John Bazemore, File]\n\nI was told this because I
#21: Photo/Pablo Martinez Monsivais, File]\n\nWhite House chief of staff Josh Earn
#22: and Government Reform Committee, (Alex Brandon/AP)\n\n(Or take a look back at
#23: , (AP Photo/Eduardo Munoz]\n\nU.
#24: \n\n(AP Photo/Gerald Herbert]\n\nA man is seen inside the Central
#25: \n\n(AP Photo/David J, Phillip]\n\nPolice said that a man attacked a
#26: Photo by Jabin Botsford/The Washington Post]\n\nUpdate 10: 15
#27: (Courtesy of the U.S. Air Force]\n\nA federal judge has set a hearing
#28: \n\n(AP Photo/Eric Gay]\n\n(CNSNews.com)
#29: <[endoftext]>(AP Photo/Cheryl Gaynes]\n\nThe
#30: 14, (AP Photo/John Bazemore]\n\nOn Wednesday, the University of Michigan
#31: , (AP Photo/J, Scott Applewhite]\n\nThis post has been updated,\n
#32: members of the Russian government. (The Washington Post]\n\nThe Washington Post reported Sunday after a

## Layer 5, Expert 8

#1: , but all the signs suggest it will be fair game to leave. Corporate giants have long been reluctant
#2: .\n\nOr, if you're feeling adventurous, try making
#3: 't see how that can be 'fair game,' So how can you pick a
#4: little more friendly than me, but they're fair game, I do know pretty much what that means
#5: been posed a few hours earlier, "Woe is me and I am, I am the
#6: sharing and as always,\n\nKevin\n\nPS: I would love for you to comment and
#7: started to turn sharply.\n\n"Woe is me and I am, I am the
#8: Let us know in the comments below.\n\nPS: If you've got a question
#9: U.S. Army Special Forces soldiers (sans the American flag on the back, of course
#10: .\n\nSpecial thanks to Paddy\n\nPS & MFK will be at the game,
#11: funky yeast and things like that,\n\nPS] I know you're a big fan of
#12: handled this crisis."<[endoftext]>We're long past the point of looking at a person'
#13: \n\nThanks,\n\nJai Thomas\n\nPS – Also, I would like to point out
#14: dance and dancing and singing and all that. Woe is me, I know this is hard,
#15: reading!\n\n-Eddie\n\nPS: You can also mail your feedback to
#16: needs to be reformed,"\n\nSans Eames, Socialist Party's former
#17: uruShark (MySpace)\n\nGood News – Tango\n\nBad News – T
#18: Public Education". However, it is well past time to sit back and listen to the voices
#19: a T-shirt with the phrase 'Woe is me' on the front.\n
#20: Associated Press, "It's long past time for the group to return to a time
#21: a greater champion of your cause,\n\nWoe to the others if you quit your job immediately
#22: Il have to answer that for myself.<[endoftext]>Sans–based developer Maxis Games has today announced
#23: accountant.\n\n* *\n\nWoe to anyone, but the public\n\nConsider
#24: \n\nHe said: "It's entirely fair game for anybody to be involved. For us,
#25: \n\nNot all cats are fair game, Some are big cats, Some are small
#26: ," he said,\n\n"It's fair game"\n\n"What we're seeing in
#27: .\n\nIn other words: It is well past time that the artist is permanently unemployed and given
#28: "I think in this case, it is fair game," said Dr. William M.
#29: this rock band like this?", it's a good sport and I'll get up and go "Fuck
#30: society as a whole,\n\nIt is well past time for the working poor (those who are
#31: what you can see on an official page (sans the game), it's for the
#32: in the sugar and spice when I'm feeling really lazy,\n\nIf anybody comes to

## Layer 5, Expert 9

#1: contacts, including the authenticity of any previous conversations,\n\nThe Obama administration did not elaborate on the
#2: Francisco, with another 1,200 in Portland,\n\nIn San Francisco, they've had four
#3: of those charges have been tested against the man,\n\nProsecutors said officers shot the man as he
#4: about the lawsuit, but declined to comment further,\n\n"That is not something we are going
#5: said, though no criminal wrongdoing has been alleged,\n\n"I think that there'
#6: been carried out as the target had been located,\n\nThe African Union Mission in Somalia (AM
#7: to take place there, Fyfe said,\n\nThe work is being done by the U
#8: did not immediately respond to a request for comment,\n\n
#9: "an existing public facility" as an option,\n\n"The bill also calls for a public
#10: not Swanson will accept an extension with the Braves,\n\nThe Braves could be considering trading Swanson as
#11: Those funds totaled $1,900 a month,\n\nTo help pay for the grant, the
#12: her staff she had a contract at the zoo,\n\nThe pair would go on to wed in
#13: to the source, who declined to comment further,\n\n"The complaint is the most serious of a
#14: said the company had no comment on the matter,\n\nThe company also said it considers the project
#15: if the case was involved with a police officer,\n\n
#16: the company had already been contacted about the plan,\n\nA spokesman for
#17: \n\nHe died later Saturday, police said,\n\nThe New York Police Department said Mank
#18: and Pakistan earlier this month, the BBC reported,\n\nYesterday, the BBC's Tim Wallace reported
#19: separately and had not been involved in any violence,\n\nMr Corbyn said he had "full confidence
#20: \n\nHe could not immediately be reached for comment,\n\n"I don't know
#21: been waiting for him to arrive at the hospital,\n
#22: , but died at the hospital the next day,\n\nThe organ donor was said to have been
#23: clear that they're being forced to sell out,"\n\nIn a release, the Tribune said that
#24: statement that it had no comment on the lawsuit,\n
#25: the report, which it was unable to cite,\n\nFord was visiting the city of Sirte
#26: of Administrative Hearings said it had no comment,\n
#27: caches and hit JSIL positions, the officials added,\n\nReporting by Tom Perry; Additional reporting
#28: then leaves the main camp to a police station,\n\nThe police have searched the house, and
#29: has not heard of any prosecutions of the officers,\n\nThe strikes occurred in the Syrian capital that
#30: open the bags, but no drugs were found,\n\n
#31: woman could not be reached for comment Thursday evening,\n\nThe mother, who has also been identified
#32: unable to comply with all subpoenas it receives,\n\nThe California Department of Justice declined to comment

## Layer 5, Expert 10

#1: I'm still waiting for them to catch up, I'm hopeful they will not
#2: location-based advertising is designed to help users keep up with their friends and family on the road,
#3: product by following us on Twitter.\n\nSign up to our Newsletter by\n\nEnter your email
#4: create a users name * for more information, Sign up here>>\n\nEmail<[endoftext]>
#5: akhchivan, I find that you cannot keep up with the number of students. I see that
#6: our weekly newsletter, get on,\n\nSign up here>>\n\nEmail<[endoftext]>They're working
#7: mail newsletter, every Wednesday from your inbox. Sign up here,<[endoftext]>When you talk of the evolution
#8: to see. Check it out,\n\nSign up for Meet the Press Daily email alerts and get
#9: \n\nThe most reliable politics newsletter. Sign up for POLITICO Play, a daily email update,
#10: app, is a free service that lets you keep up with the latest Google news, photos and more
#11: you can join the official Discord channel, and keep up with the development on the official Steam and Indie
#12: Petersen, Editor\n\nLike this story? Sign up for our Daily Digest to get Tablet Magazine
#13: feel safe, it makes you think, it makes up for it in ways you don't think of
#14: .\n\nIf voters have any chance of making up their
#15: The best way to watch The Simpsons is to catch up on the first season of the FX TV series
#16: \n\n"If an elected congresswoman is holding up the best evidence when she comes out and declares
#17: ley Field Inn is one place where you can catch up and pick up the game. The game is
#18: season, the Chargers were the only team to catch up to the Rams last season on the offensive side
#19: who are not in line at the grocery store makes up about 25 percent of those who buy food on
#20: for Disease Control and Prevention statistics,\n\nSign up for the Power on Trial newsletter Get our trial
#21: \n\nPhiladelphia Eagles: With the Eagles defensive line holding up well throughout the game, the Eagles won a
#22: \n\nOf course, keeping up with the world of data, and especially of
#23: because their parents don't bother to keep up the\n\nKids are often
#24: years now, and I think it's really catching up, It's really noticeable, It's easier
#25: to 8,800 jobs in a town that makes up about 2 percent of the city's employment,
#26: s the time when the franchise is starting to catch up to the original series,\n\nOver
#27: ? I'm too bogged down keeping up with,\n\nSo they're
#28: content in this month's month to keep up with your work, or maybe use some of
#29: You're not going to have to keep up with the world, you'll never
#30: it was so fast, that I couldn't keep up with it. After an hour or so I
#31: .\n\nThe most reliable politics newsletter. Sign up for POLITICO Playbook and get the latest news
#32: The Final Score: 5-0\n\nStory continues below\n\nIf you're going to take

## Layer 5, Expert 11

#1: 5 more than 5-6. Because I know I'm sitting on a flat sheet of
#2: \n\nI said I was going to stick with it, and I
#3: something positive and that is my passion, I know I
#4: as opposed to DVD: "I know that I'm in England, I think I
#5: feel very insecure as a student, I know what I would do if I had to do it again
#6: year, he will get a chance to show that he belongs in the NFL,\n\n"
#7: As for the $5-million, I realize I'm looking to put a little doubt
#8: profiles.\n\nBenji told detectives he knew he didn't belong to
#9: " he said,\n\n"They think they can be very vocal, and it'
#10: German newspaper,\n\n" I told her I think I have a lot to offer her,
#11: see you all here,\n\nBut I think I've put a lot of the comments
#12: the "first" game that I thought I would try and collect with DOTA 2,
#13: a good pass rusher\n\nEven if you think you can get away with it, you should keep
#14: \n\n2: 00: You think you can win this thing, but it'
#15: m very aware of this,\n\nI know I'm probably not in a position to
#16: 't,\n\n"We know what we have to do," says Mr S
#17: want to go back to summer 2013, I thought I was just doing something else, I mean,
#18: as a bad question, I'm certain I'm here to answer it without ever
#19: talk to you in a safe place, I know I did,\n\nTake them to the local
#20: X as a good piece of work and I know I'll be getting the next one too
#21: I just called the secretary and I told her that I'd
#22: railway station,\n\nHe then told the man he had killed the woman because she had not paid
#23: know?\n\nA: I told him that I didn't know, and that I
#24: I know," he said, "I know what I need to do, I know what I need
#25: task to tackle this project, and I knew that I had to try to overcome the resistance to say
#26: In the past, I've said that I wouldn't be bothered by the presence
#27: t sure what it was about, I knew that I had gone to complete my own version
#28: ?" she asked, I told her that I was too embarrassed
#29: .\n\nI can't really say I'm surprised. The odds of re
#30: \n\nI know I'm having a hard time explaining myself
#31: \n" I told him that I didn't want to go,
#32: \n\nI cannot say I'm sorry to friends, family and

## Layer 5, Expert 12

#1: -based approach to sustainable fisheries management, including the establishment of sustainable fishery management plans, especially for
#2: \n"The US has openly called for the creation of
#3: illegal under international law. The resolution calls for the establishment of peace based on the Oslo Accords,
#4: more than a dozen cities in India calling for the release of Modi's alleged "political
#5: "an international community that is committed to the peaceful establishment of the entire region," and they
#6: development.\n\nThe steps announced Monday include the establishment and\n\nA governmental plan for North
#7: in September 2016 will, by 2018, require the establishment of a national, voluntary service delivery network,
#8: development of the study of human intelligence and the development of research into theories of human nature and human
#9: a dramatic crackdown on undocumented immigrants, calling for the creation
#10: The city council recently passed a resolution calling for the establishment of a "Ministry of Minorities
#11: suffering from health issues in recent months, announced the establishment of an armistice with the U.
#12: comes just weeks after Gov, Chris Christie ordered the release of records from the State Police and the New
#13: UK joined a formal agreement on a mechanism for the establishment of trade in energy, which is intended to
#14: abetting the assassination and possessing a firearm during the commission of a crime, is serious,\n\n
#15: instances where the threat has been shown to justify the commission of a violent act," he said,\n
#16: priest of El Salvador have all publicly called for the establishment of a national inquiry into the new sex-
#17: for an end to the occupation of Jerusalem and the establishment of a sovereign Palestinian state. The resolution was
#18: The resolution calls for international cooperation and 'the establishment of a new international framework for the global education
#19: 's national security adviser has repeatedly called for the creation of a political panel to deal with Iran.
#20: speech in 2011, in which he called for the creation of a new Islamic state in Syria, Sheikh
#21: ONG KONG — The Chinese military has announced the establishment
#22: September 18, 2013, the Iranian government announced the establishment
#23: , providing for the establishment of a mechanism for the establishment of military operations in Libya, which was rejected
#24: the 1973 Arab-Israeli war that led to the creation of the Middle East peace group,\n\n
#25: \n\nThe U,N, charter calls for the establishment of a "united and peaceful forum
#26: 's Constitution, which in turn allowed for the establishment of the Constitution Party
#27: the face of a global effort aimed at preventing the spread of WMD. But critics of the ban
#28: The Government of the United Kingdom is committed to the establishment of a North Atlantic Free Trade Association, or
#29: violence, two counts of having a weapon during the commission of a felony, and one count of reckless
#30: the means of achieving our aims, This includes the establishment of the world food system, the eradication
#31: use firearms, knives, or other weapons in the commission of a felony or a serious felony conviction,
#32: with "allegedly inciting or assisting in the commission or instigation of the murder of someone."

Figure 14: Top-activating 32 tokens for the unfiltered experts 7-12 (as ordered numerically) at layer 5 in the CP$\mu$MoE GPT2 model.

## G.2 LLM steering

Here we provide additional evidence that the experts' specialization is mechanistically relevant to the functionality of the network, in the sense that we use them to steer the LLM's output.

In particular, we use a larger GPT-2 model trained from scratch with $\mu$MoE layers at each MLP layer, **using 2048 experts at every layer**, following the setup in Section 4.3. By modifying the forward pass of the trained model—specifically, adding selected expert cluster center vectors to each token's input latent activation vector before applying the $\mu$MoE layer—we can consistently control the model to generate outputs aligned with specific themes. Illustrations of this approach, using 4 different manually chosen experts (with their first 8 generated samples) are shown in Figure 15. The selected experts guide the language model's outputs toward discussing topics such as climate change, police brutality, or foreign politics. We suggest that these findings further demonstrate the effectiveness of the $\mu$MoE layer in facilitating controllable generation of language model outputs.

However, we note that these initial results are hand-selected examples of some of the experts which do exhibit sensible specialization. We find many experts, when activated, do not steer the generations in such an interpretable high-level manner.

## G.3 CLIP ViT-B-32

**Qualitative visualization**   Additional results to further substantiate the claims in the main paper about expert class-modularity are presented here. Firstly in Figure 16 are many more random images (of those with expert coefficient $\geq 0.5$) of the first few experts as they are ordered numerically. Furthermore, when we use an even larger number of experts (i.e. 2048) we observe a select few experts developing what appear to be very fine-grained specialisms, as shown in Figure 17. For example, images with large coefficients for #203 are often animals on top of laptops, whilst images with high coefficients for #1203 are animals eating corn.

**Counterfactual intervention barplots**   Next, we show barplots of the class labels whose test set accuracies are most changed under the counterfactual question in the main paper: "had (expert $n$) not contributed its weight, how would the class predictions have changed?". These are shown in Figure 18 and Figure 19 when using a CP$\mu$MoE as a final and penultimate layer respectively. As can be seen, we often observe that a higher number of experts (the final rows in brown color) lead to experts that, upon ablation, cause the model to lose almost all its accuracy for fewer classes. Experts here are chosen in numerical order and only those yielding $\geq 0.5$ total accuracy change to any class upon counterfactual ablation.

# H   Ablation studies

## H.1   Entmax vs softmax

We find the use of the entmax activation function [54, 55] to produce more monosemantic experts, as quantified by the measure of polysemanticity used in the main paper. We show in Figure 20 the mean expert polysemanticity (of those experts that affect the class accuracy upon ablation) for CP$\mu$MoE-r512 final layer models fine-tuned with various numbers of experts. As can be seen, the entmax function consistently produces more monosemantic experts for larger total expert counts. We attribute this to the sparsity in entmax's post-activation distribution (whereas the softmax function can just as readily output a uniform distribution over all expert coefficients).

# Raw model outputs

**Sample 1**
The biggest issue of today's world is the pressure on the planet's fragile biosphere to become more carbon-rich. People have been making alternative energy sources since the dawn of civilisation, but there is no evidence for them, says the UNEP report. "The climate crisis is

**Sample 2**
The biggest issue of today's world is the lack of unity. One of the biggest problems is that as the vast majority of the media don't want to say so, it is increasingly difficult to communicate what is called "the truth" with the public.      In fact, in some

**...**
The biggest issue of today's world is the internet, which not only protects the most important aspects of our lives, but also has important benefits that greatly benefit other people. We can bypass the police, governments, banks and everyone else. Nobody wants to be charged with a crime, and nobody

The biggest issue of today's world is not climate change, but the failure to make the right choices about who to send to war," he said. "The danger is that our leaders are unwilling to do the right thing and avoid a massive catastrophe."      Watt noted that the United

The biggest issue of today's world is this:      As the world continues to get more and more food into the hands of the poor, we are also finding out that food is becoming harder to find. And the pace of demand for food is quickly in decline due to some key reasons

The biggest issue of today's world is not how to manage a growing population (although we must do that), but how to manage a growing food production system. Growing food in urban areas is a challenge, but it also is a strategic opportunity. That's because city dwell

The biggest issue of today's world is the continuing need for mass-human immigration. Of the 3 million new visas issued in fiscal year 2010, 2.4 million have been approved, and the rest (1.6 million) will be, at best, temporary. That's because of

**Sample 8**
The biggest issue of today's world is not the problems of the last century; it's the problems of today.      The world of today is experiencing a profound change. We are less and less rational, more and more embittered, and the world will inevitably evolve to a situation

# Steering LLM outputs ↓

## Intervening to increase:
"weather"/"climate" expert

**Sample 1**
The biggest issue of today's world is that of climate change, and many people are already building their homes on land from Mount Meru in the month-long dry monsoon that freezes into March. But that could be the biggest issue facing India; even in February, a low-pressure

**Sample 2**
The biggest issue of today's world is the risk of giant rain, especially in the British Isles, with the high temperature expected to reach an highest maximum of 2C. It's likely to become a drier pattern again today with winds of up to 10 miles per hour.

**...**
The biggest issue of today's world is the warm temperatures that you can expect to see today, with the heat centering around the equator. But, it is not too cold, so we have to approach this problem cautiously. So far so good, anyway!      So, here

The biggest issue of today's world is not to burn down the planet. But it is to freeze it. We need a lot more water ice, and we need a lot more sunshine. But the climate has warmed, and now there is an area of relatively low temperature that is getting warmer

The biggest issue of today's world is this:      As summer approaches, many farmers will grow one-to-two tons of wine every year.      But, most will have some water. There's nothing to worry about, especially in the Southern California mountains. If

The biggest issue of today's world is a lack of sunshine, which makes the heat evaporate away. The heat in the day is a much-remarsened mist. We're in a perfect storm of sunshine here, at high pressure, and it's going to be cloudy for a

The biggest issue of today's world is a lack of rainfall and extreme temperatures, so the coastal area will be dry to some extent. That means the rest of the area will be prone to the high temperature, but those regions should be dry, and the temperature could be as high as 100

**Sample 8**
The biggest issue of today's world is that, on average, the atmosphere is too hot to be able to cool, although the warmer air is melting to some parts of the Arctic continent.[3] The area is covered with a haze of wind-shear (twice), with a

## Intervening to increase:
"programming" expert

The biggest issue of today's world is that of the 'new' value: 'a' with `result', from which all operations are overloaded.      The other class that lists a type is `f`, which is a lambda, will be a type that is a new Value.

The biggest issue of today's world is the lack of, as a constant:      ()<A> -> A . And the get() type only makes the function return value, but, from type<A> implements A<T<T<T>] , it

The biggest issue of today's world is the accumulation of non-ditional , or ___init___ , in every object.      Here, we have an instance of , and an are, with type F , and .      The first type is , and the second,

The biggest issue of today's world is not to solve, but to be able to create. And it is: Any new construct cannot be created anywhere in the world (except at most one). These types of construct is a member of the set , where .      If we

The biggest issue of today's world is this:      As a result, the world will grow one world after another, after all.      We will see millions, billions, and billions of things. And the way we think of the world is in the identity of some kind of

The biggest issue of today's world is that it's a problem of type (higher type) in an overloaded type, like a function. My type is a result, in a type like a, you get a result, after run(T), and the type t is a result is

The biggest issue of today's world is the absence of a class.      "A big number is always a number, and the sum of all the other values is always a number. But the (biggest) type is always a, and the same is always a. But the

The biggest issue of today's world is that, on every stream, no matter of size, exception is not a superclass, and a static class cannot be. It is a single construct and void is a new int. But a non-class is not any of its complement.

## Intervening to increase:
"police violence" expert

**Sample 1**
The biggest issue of today's world is that we don't have enough police officers, and we have too many immigrants from the United States who aren't helping with the deportation of illegal immigrants.      "We have a drug problem, we've got a problem with drugs. We've got

**Sample 2**
The biggest issue of today's world is the militarization of police, especially in the US. It also has to do with the media's ability to get to the bottom of what's happening in Ferguson, and the killing of multiple civilians.      The US military's lack of accountability puts

**...**
The biggest issue of today's world is the accumulation of resources that are being moved, and the effects of that. In 2016, Donald Trump, Texas police officers are killed in the street.      "He has insulted the Mexican flag, the flag of the United States,"

The biggest issue of today's world is not immigration, but the war on drugs. The war on drugs is a racist, violent criminal regime that is in the process of dismantling our country's efforts to keep us safe.      The War on Drugs has been an unjust and unw

The biggest issue of today's world is this:      As the United States says, the #8 target in Charlottesville #Charlottesville police death is "driving" violence, and other #2's getting killed in the US. https://t.w/r #2's killed are

The biggest issue of today's world is that we are constantly on the same side and we are constantly on the side of the Palestinians, however much we are there as well," Clinton said in July. "And we also have to be here before the people that are so violent, so many

The biggest issue of today's world is a lack of respect for the police, and the way law enverses that, police can be getting killed in a lot of other ways, including the use of drones. That's where I think, as a police officer, it's a greater

**Sample 8**
The biggest issue of today's world is that we have a broken immigration system. We have a broken economic system. We have a police officer or a policeman who is unarmed and is being continually killed by a person who owns a vehicle. They have killed five officers this past week.

## Intervening to increase:
"foreign politics" expert

The biggest issue of today's world is that the Americans must implement a plan to end the war with Iraq and to improve postwar relations, American leaders said. However, the Soviet Union was always wary of the American policies.      Korea, the North Koreans, the US and Great Britain

The biggest issue of today's world is coming to the realization that since the cold war era, the United States has turned its back on the Soviet Union. They secretly pushed through the historic deal in 1983, and the Soviets were all too eager for a resolution to stop the Berlin and Tehran.

The biggest issue of today's world is the United States' difficulties with the East, and the US, in every sense of the world. They see no other use for the region is the obvious obvious of a new problem: the problem of Western imperialism. Also, the US sees no trouble

The biggest issue of today's world is how to solve the issue of the Black Sea when it is. On a modern scale, the US Embassy in Moscow had changed the status quo through Washington's representative.      From the time of Napoleon until the Russian Empire's recognition of Russia in 18

The biggest issue of today's world is this, both sides are also pursuing a plan to avoid a permanent strategic alliance but they cannot reach for this is now calling its long-term friendship could be achieved with the current.      The US government, especially in Washington, does not want to

The biggest issue of today's world is a country's strategic response to Iraq's invasion and the rest of Iraq ruled in the 1950s.      A week after the start of the war, it launched a massive operation to find Baghdad's exiled neighbour and to capture or occupy the city.

The biggest issue of today's world is a fight between the two leaders over their mutual aspirations. Those were long-lasting issues over the Berlin-Ottoman-Rabid government have now tried to resolve. And those difficulties have left the country with a war-weary Russian President and

The biggest issue of today's world is surely, on a broad level, some of the most consequential economic positions are being kept for half a year.      Donald Trump and Vladimir Putin have been at every level to try to end a crisis over Russia - but the last days were also a

Figure 15: **Steering LLM outputs by forcefully activating experts:** adding specific manually chosen expert's cluster centers to GPT-2's activation vectors at particular layers reliably steer the LLM generations towards specific themes, based on the learned expert specialism. For example, we see an expert that steers discussion towards police violence, or about the climate. The initial prompt in every instance is the text: ''The biggest issue of today's world is''.

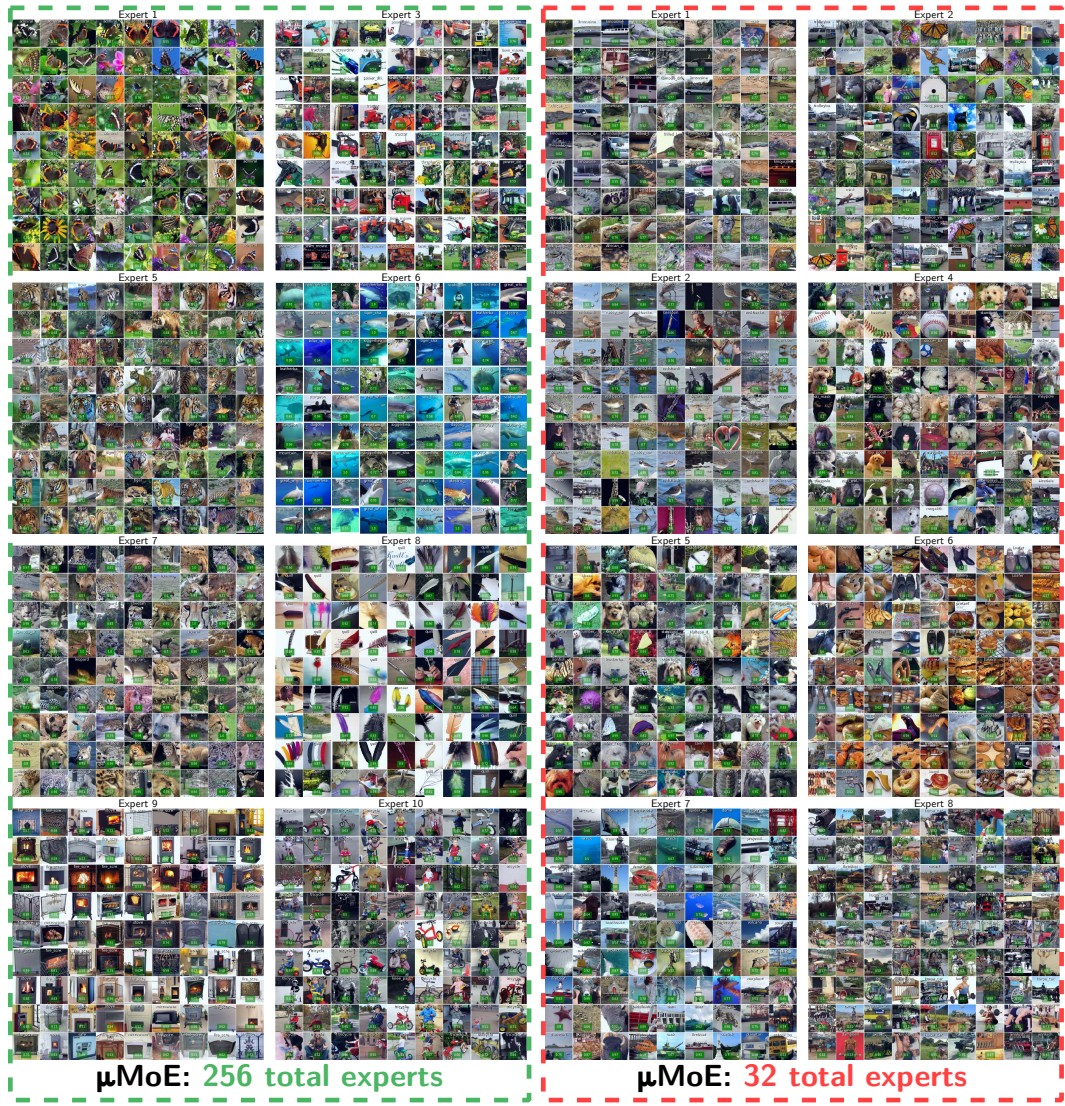

Figure 16: **High** vs **low** **total expert count**: *Randomly* selected training set images with expert coefficient $\geq 0.5$ for the first 10 numerical experts (of those processing any images with coefficient $\geq 0.5$). Results are with CP-r512 $\mu$MoE layers with 256 (left) and 32 (right) total experts respectively. We highlight the apparent specialism of the experts when a higher total number is used. **(Please zoom for detail)**

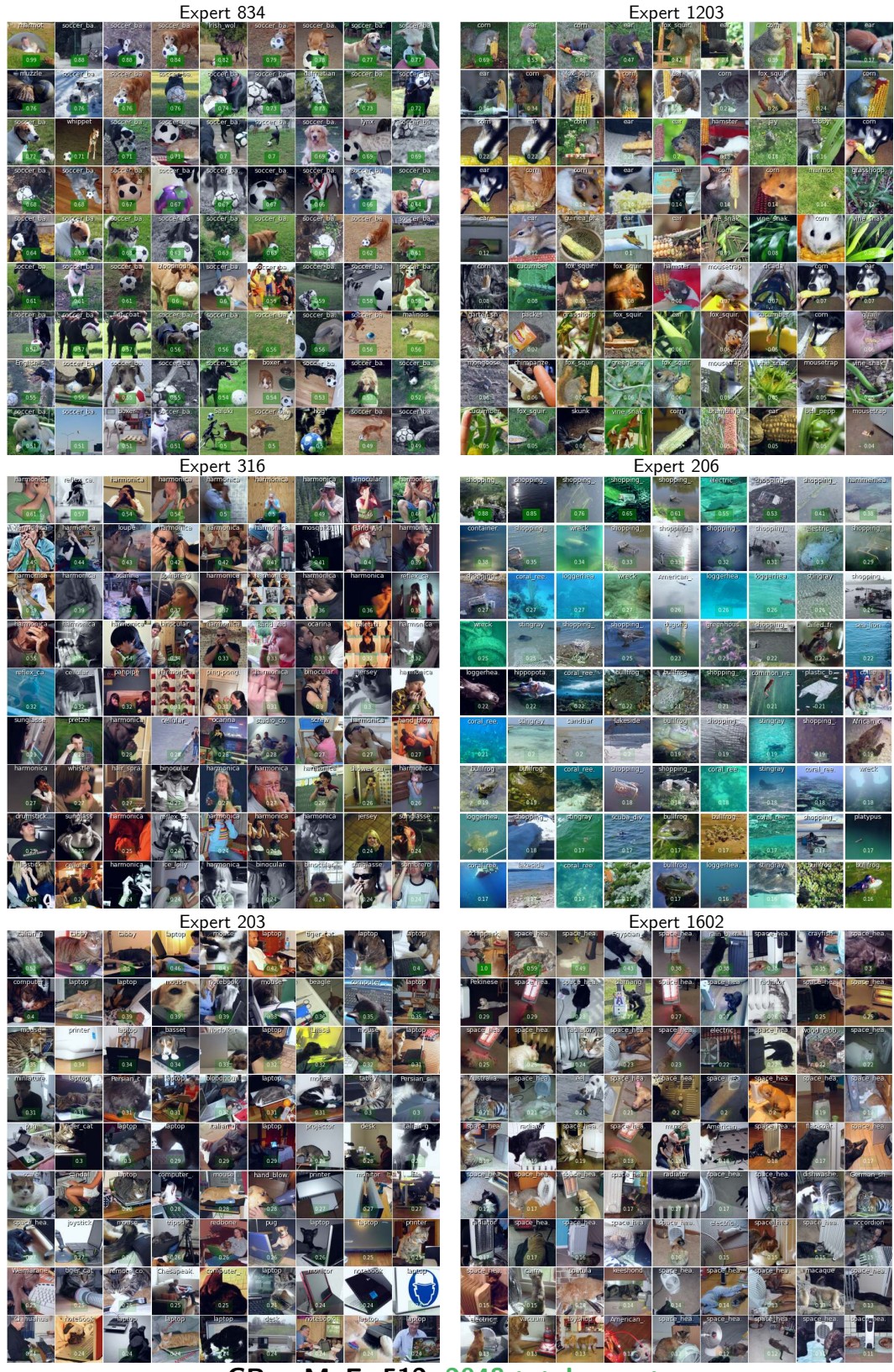

**CPmuMoE-r512:** 2048 total experts

Figure 17: **Fine-grained expert specialisms**: *Manually* selected experts (and images ranked by *highest* expert coefficients) processing what appears to be very fine-grained categories (e.g. animals with footballs, trolleys in water, etc.). Model fine-tuned on ImageNET1k with a high number of 2048 experts and a CP-r512 $\mu$MoE final CLIP layer. **(Please zoom for detail)**

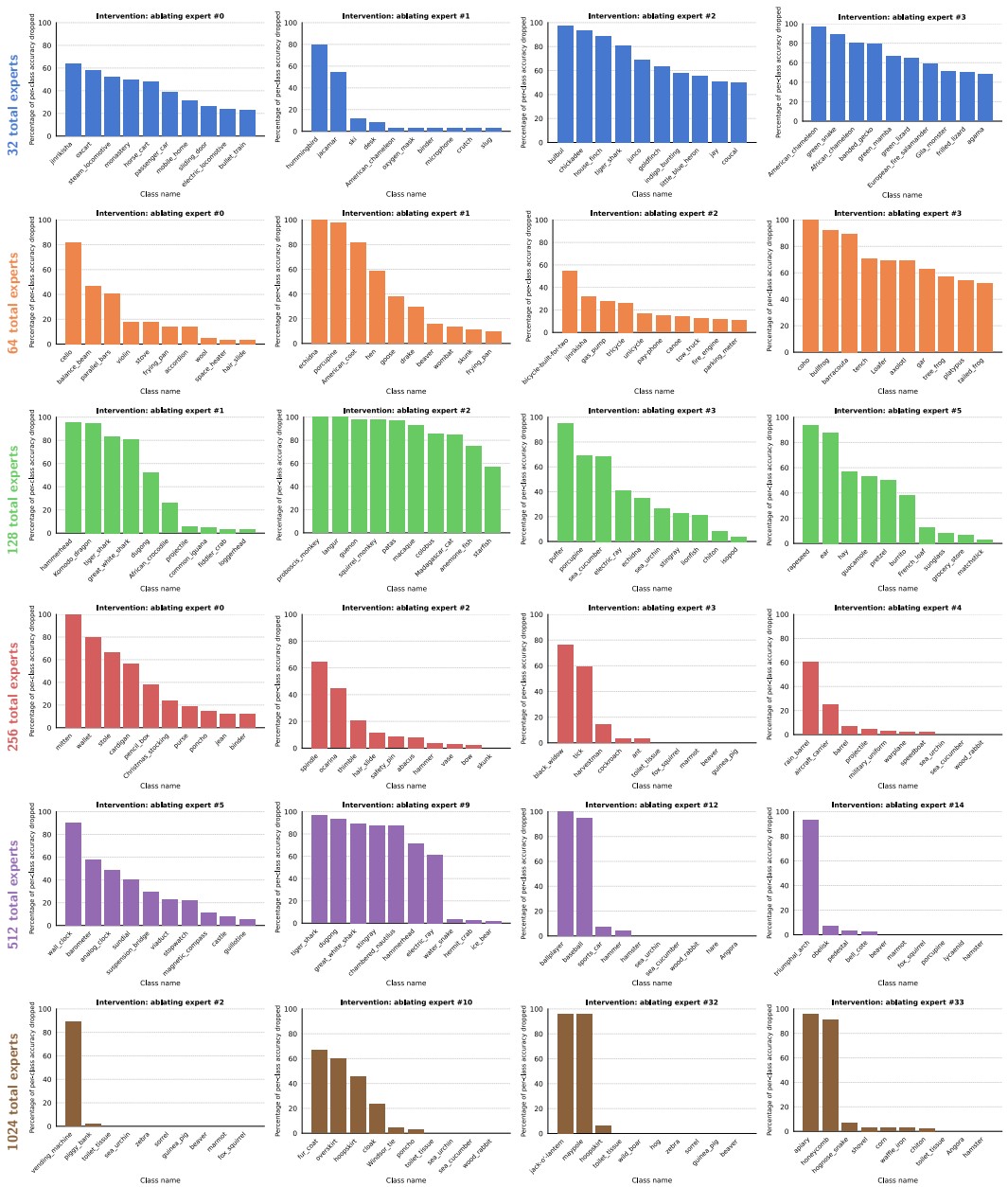

Figure 18: **Penultimate layer CPμMoE**: Percentage of per-class test set accuracy lost when intervening and ablating particular experts (along the columns). In general, the more total experts (rows), the more class-level monosemantic the experts are as indicated by the mass centred on fewer classes, and with higher magnitude. Shown are the first 4 experts in each model (row) to change $\geq 0.5$ of any class' accuracy when counterfactually ablated.

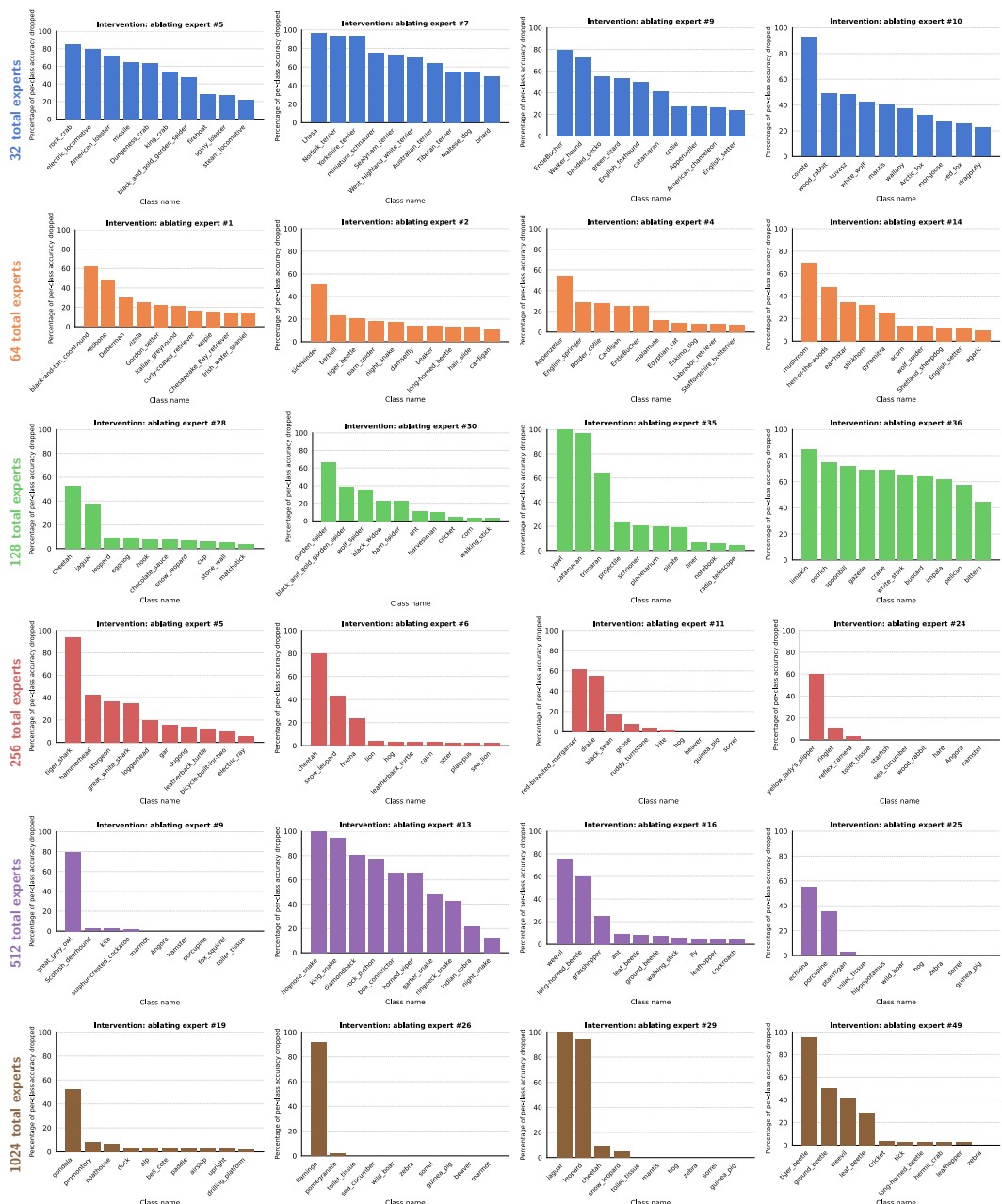

Figure 19: **Final layer CPμMoE**: Percentage of per-class test set accuracy lost when intervening and ablating particular experts (along the columns). In general, the more total experts (rows), the more class-level monosemantic the experts are as indicated by the mass centred on fewer classes, and with higher magnitude. Shown are the first 4 experts in each model (row) to change $\geq 0.5$ of any class' accuracy when counterfactually ablated.

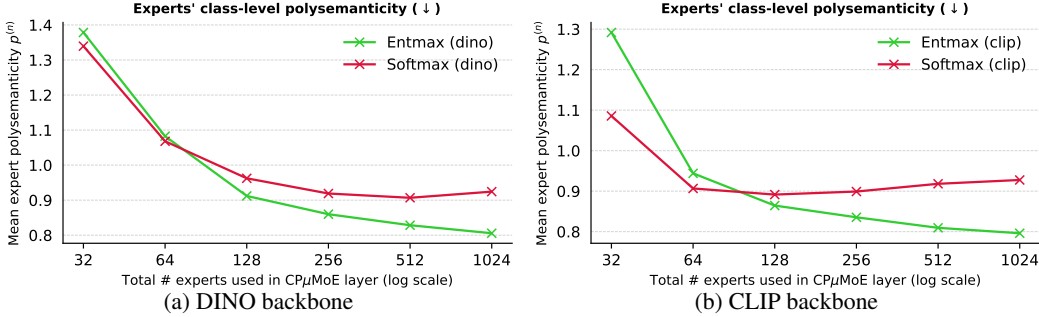

|  |  |
|---|---|
| (a) DINO backbone | (b) CLIP backbone |

Figure 20: **Softmax vs Entmax ablation** `CPμMoE-r512` final layers trained on ImageNET, and the resulting class-level polysemanticity. For large values of experts, the entmax activation produces more specialized experts.

## H.2 Fast forward pass computation speedups

We next report in Table 8 the actual number of FLOPs (as reported by https://detectron2.readthedocs.io/en/latest/_modules/fvcore/nn/flop_count.html) when executing PyTorch $\mu$MoE layers using the naive forward pass relative to the cost when using the fast `einsum` computation derived in Appendix B–the fast computation is many orders of magnitude less expensive (using one A100 GPU).

Table 8: Original $\mu$MoE layers' FLOPs vs the fast `einsum` forward passes in Appendix B (for $N = 512$ experts with 768-dimensional input and output dimensions).

|  | **CPμMoE** | **TRμMoE** |
|---|---|---|
| Original FLOPs | 155.1B | 622.8B |
| **Fast model FLOPs** | **1.4M** | **3.5M** |

## H.3 Batch normalization

We next perform an ablation study for the use of batch normalization (BN) before the activation function for the expert coefficients. We study CP$\mu$MoE final layer layers with CLIP ViT-B-32, quantifying BN's effect on expert class-monosemanticity as a function of the expert count. Concretely, we perform the same class-level polysemanticity experiments as in the main paper, with and without batch normalization in Figure 21. As can be seen clearly, the batch normalization models lead to individual experts that are increasingly class-monosemantic as desired (as a function of the total expert count).

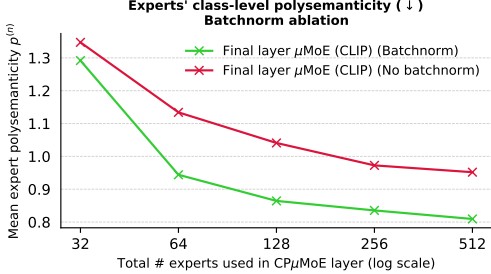

Figure 21: Ablation study: batch normalization leads to more class-level monosemantic experts.

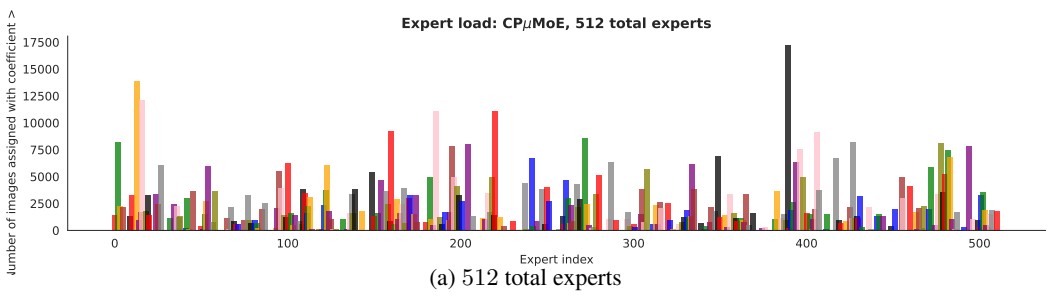

(a) 512 total experts

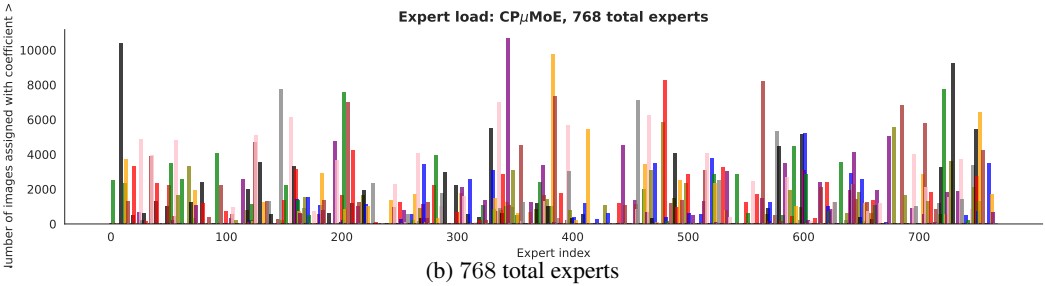

(b) 768 total experts

Figure 22: Expert load: Number of training set images with expert coefficient $a_n \geq 0.5$ for CP$\mu$MoE models fine-tuned on ImageNET1k. Bars are drawn with 3x width and colored sequentially in a repeating order of distinct colors to help visually distinguish between neighbors.

## H.4 Expert load

Here, we plot the expert load in Figure 22 to give a visual indication of how many images are processed by each expert with $a_e \geq 0.5$ for CP$\mu$MoE final layers fine-tuned on ImageNET1k with a CLIP backbone. Whilst clearly, not all experts have images with a coefficient of at least $0.5$, we see a relatively uniform spread over all experts. Furthermore, we note the cost from 'dead' experts is not particularly troublesome in an $\mu$MoE given its factorized form–speaking informally, we would rather have too many experts than too few, so long as there exist select individual experts conducting the subcomputations of interest.

# I Additional performance results

## I.1 CLIP ViT-B-32 ImageNET1k ablations

Here, we compare the performance of parameter-matched $\mu$MoE final layers (for varying expert counts $N$) to linear layers for fine-tuning large vision-language models (CLIP ViT-B-32) on ImageNET1k. Following the robust fine-tuning protocol of [89], we use the largest possible batch size (to fit on one A100 GPU) of $4096$, and the same learning rate of $3e - 05$.

For $\mu$MoE layers, we reduce the layer ranks to parameter match *single* linear layers for each value of total expert count. We plot in Figure 23a the ImageNET1k validation loss after 10 epochs of training, where all expert counts out-perform the linear layers initialized the same default way with elements from $U[-k, k]$. However, to parameter-match single dense linear layers, we must decrease the $\mu$MoE layer rank upon increasing the expert count. This is a concrete example of where the extra parameter efficiency of TR$\mu$MoEs can come in useful (as discussed in Appendix D.1.2). Consequently, TR$\mu$MoEs' resulting expert matrix ranks are increasingly larger than that of CP$\mu$MoEs in the parameter-matched setting. For example, the parameter-matched layers with 512 experts in Figure 23a have a max expert matrix rank of 165 for the CP$\mu$MoE compared to a much larger 208 for the TR$\mu$MoE.

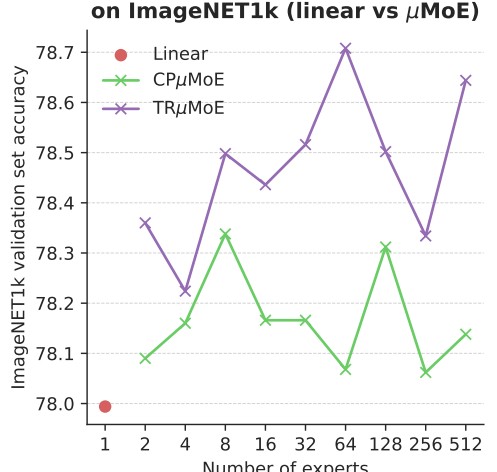

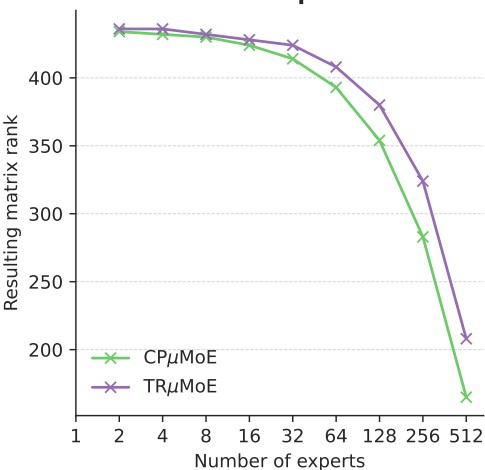

(a) Accuracy comparison ($\mu$MoE vs Linear)   (b) Rank comparison (CP$\mu$MoE vs TR$\mu$MoE)

Figure 23: Comparative analysis of fine-tuning CLIP ViT-B-32 with $\mu$MoE layers using different configurations. **All experiments have the same number of parameters**.

Table 9: **Hierarchical** `S-16` **TR$\mu$MoE-mixers and CP$\mu$MoE-mixers**: ImageNET1k val. accuracy at 300 epochs pre-training; $N_1 = 64$, $N_2 = 2$ experts).

| Model | Val. acc. ($\uparrow$) | # Experts per block | # Params |
|---|---|---|---|
| MLP | 70.31 | n/a | 18.5M |
| **CP$\mu$MoE** (hierarchy=1) | 71.29 | 64 | 18.6M |
| **TR$\mu$MoE** (hierarchy=1) | 71.26 | 64 | 18.3M |
| **CP$\mu$MoE** (hierarchy=2) | 71.24 | $64 \cdot 2$ | 19.5M |
| **TR$\mu$MoE** (hierarchy=2) | **71.56** | $64 \cdot 2$ | 18.7M |

We attribute TR$\mu$MoE's even greater performance gains over CP$\mu$MoEs here to the more favorable relationship between tensor rank and expert matrix rank (a larger weight matrix rank meaning the resulting layers' activations live in a larger dimensional subspace) (see Figure 23b).

## I.2 Hierarchical $\mu$MoEs

**Hierarchical $\mu$MoE Mixers** We train from scratch two hierarchical $\mu$MoE MLP-mixer `S-16` models for 300 epochs on ImageNET following the same configuration as in Section 4.3 of the main paper. Concretely, we use a **two-level** hierarchical $\mu$MoE with $N_1 = 64$ experts for the first level and $N_2 = 2$ experts for the second layer (128 total effective experts). As shown through the results in Table 9, the hierarchical $\mu$MoE's also perform well against the MLP alternatives, whilst providing even better parameter-efficiency.

**Hierarchical $\mu$MoE fine-tuning layers** We also perform additional experiments with hierarchical $\mu$MoEs used to fine-tune CLIP `ViT-B-32` models on ImageNet1k. Here we use the experimental setup in [63, 64], training each model for a single epoch with the specified learning rate of $1e-05$. We fine-tune hierarchical $\mu$MoE CLIP models with up to 4 levels of hierarchy as shown in Table 10, where the best-performing models (averaged over 5 runs) are found with 2 levels of hierarchy.

## I.3 Comparisons to dense/sparse MoEs

The goal of the $\mu$MoE layer is to facilitate more interpretable subcomputations with a similar number of parameters and FLOPs to regular dense layers. Whilst the layer does not aim to improve on the *capabilities* of existing MoE layers, we nonetheless provide an initial comparison study here in Figure 24 for completeness. As can be seen, in addition to the scalable expert specialization provided,

Table 10: **Hierarchical $\mu$MoEs**: Mean validation-set accuracy with a CLIP ViT-B-32 fine-tuned with hierarchical $\mu$MoE final layers on ImageNET1k. Shown are the number of parameters as the number of total experts increases to 8192 with 4 levels of hierarchy, and the corresponding number of parameters needed for each expert total using a hierarchy 1 $\mu$MoE, and regular MoE. Results are the average over 5 runs with different seeds. Additional expert modes for TR$\mu$MoEs have the additional ranks set equal to the corresponding number of experts at the new mode(s) (e.g. 2 and 4).

(a) Hierarchical `CP`$\mu$MoEs ($R = 512$) fine-tuning CLIP `ViT-B-32` on ImageNET1k.

| Hierarchy | Val acc | Weight tensor shape | Total # experts | # Params | # Params needed (w/ 1 hierarchy $\mu$MoE) | # Params needed (w/ regular MoE) |
|---|---|---|---|---|---|---|
| 1 | $73.78 \pm 0.07$ | $\mathcal{W} \in \mathbb{R}^{128 \times I \times O}$ | 128 | 1,069,568 | 1,069,568 | 98,432,000 |
| 2 | $73.84 \pm 0.11$ | $\mathcal{W} \in \mathbb{R}^{128 \times 2 \times I \times O}$ | 256 | 1,072,128 | 1,233,408 | 196,864,000 |
| 3 | $73.80 \pm 0.14$ | $\mathcal{W} \in \mathbb{R}^{128 \times 2 \times 2 \times I \times O}$ | 512 | 1,074,688 | 1,561,088 | 393,728,000 |
| 4 | $73.82 \pm 0.06$ | $\mathcal{W} \in \mathbb{R}^{128 \times 2 \times 2 \times 2 \times I \times O}$ | 1024 | 1,077,248 | 2,216,448 | 787,456,000 |
| 2 | $\mathbf{73.89 \pm 0.10}$ | $\mathcal{W} \in \mathbb{R}^{128 \times 4 \times I \times O}$ | 512 | 1,074,688 | 1,561,088 | 393,728,000 |
| 3 | $73.85 \pm 0.08$ | $\mathcal{W} \in \mathbb{R}^{128 \times 4 \times 4 \times I \times O}$ | 2048 | 1,079,808 | 3,527,168 | 1,574,912,000 |
| 4 | $73.82 \pm 0.09$ | $\mathcal{W} \in \mathbb{R}^{128 \times 4 \times 4 \times 4 \times I \times O}$ | 8192 | 1,084,928 | 11,391,488 | 6,299,648,000 |

(b) Hierarchical `TR`$\mu$MoEs ($R_3 = 512$) fine-tuning CLIP `ViT-B-32` on ImageNET1k.

| Hierarchy | Val acc | Weight tensor shape | Total # experts | # Params | # Params needed (w/ 1 hierarchy $\mu$MoE) | # Params needed (w/ regular MoE) |
|---|---|---|---|---|---|---|
| 1 | $74.66 \pm 0.09$ | $\mathcal{W} \in \mathbb{R}^{128 \times I \times O}$ | 128 | 3,723,264 | 3,723,264 | 98,432,000 |
| 2 | $74.72 \pm 0.08$ | $\mathcal{W} \in \mathbb{R}^{128 \times 2 \times I \times O}$ | 256 | 3,724,832 | 3,823,616 | 196,864,000 |
| 3 | $74.75 \pm 0.14$ | $\mathcal{W} \in \mathbb{R}^{128 \times 2 \times 2 \times I \times O}$ | 512 | 3,726,400 | 4,024,320 | 393,728,000 |
| 4 | $74.76 \pm 0.11$ | $\mathcal{W} \in \mathbb{R}^{128 \times 2 \times 2 \times 2 \times I \times O}$ | 1024 | 3,727,968 | 8,851,456 | 787,456,000 |
| 2 | $\mathbf{74.82 \pm 0.11}$ | $\mathcal{W} \in \mathbb{R}^{128 \times 4 \times I \times O}$ | 512 | 3,726,400 | 4,024,320 | 393,728,000 |
| 3 | $74.67 \pm 0.12$ | $\mathcal{W} \in \mathbb{R}^{128 \times 4 \times 4 \times I \times O}$ | 2048 | 3,729,536 | 5,228,544 | 1,574,912,000 |
| 4 | $74.73 \pm 0.11$ | $\mathcal{W} \in \mathbb{R}^{128 \times 4 \times 4 \times 4 \times I \times O}$ | 8192 | 3,732,672 | 10,045,440 | 6,299,648,000 |

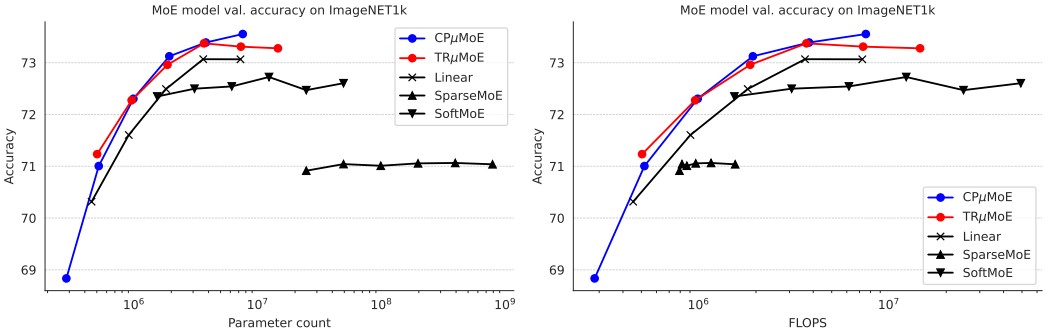

Figure 24: Results fine-tuning CLIP ViT-B-32 final layers only on ImageNET1k for 1 epoch. For $\mu$MoE layers, we increase parameter counts by varying the ranks for a fixed 64 experts. For dense ("Soft") and sparse MoEs, we increase the parameters through increased expert counts.

the $\mu$MoEs also perform very favorably against the alternative MoE models when fine-tuning CLIP on ImageNET1k.

## J  Fairness baselines & metric details

Here we present more details about the fairness comparisons and metrics used in the main paper.

**Metrics**

- **Equality of opportunity** requires the true positive rates for the sensitive attribute subpopulations to be equal, defined in Hardt et al. [76] as $P(\hat{Y} = 1 | A = 0, Y = 1) = P(\hat{Y} = 1 | A = 1, Y = 1)$ for sensitive attribute $A$, target attribute $Y$, and predictor $\hat{Y}$. In the first of our CelebA experiments we measure the absolute difference of the true positive rates between the 'blond female' and 'blond male' subpopulations for the 'blond hair' target attribute. For the second we measure the difference between that of the 'old female' and 'old male' subpopulations, taking the 'old' label as the true target attribute.

- **Standard deviation bias** computes the standard deviation of the accuracy for the different subpopulations [77]. Intuitively, a small STD bias indicates similar performance across groups.
- **Max-Min Fairness** quantifies the worst-case performance for the different demographic subpopulations [78], with $\max \min_{y \in \mathcal{Y}, a \in \mathcal{A}} P(\hat{Y} = y | A = a, Y = y)$. We compute this as the minimum of the test-set accuracy for the $4$ subpopulations in each experiment.

**Baselines**

- **Oversample** we oversample the low-support subpopulation to balance the number of input images that have the sensitive attribute for the value of the target attribute wherein bias occurs. For example, we oversample the 'blond males' to match the number of 'blond females' for the first experiment, and oversample the number of 'old females' to match the number of 'old males' for the second.
- **Blind thresholding** is implemented by unconditionally increasing/decreasing the logits in the target direction for all outputs. Concretely, the results in the main paper are achieved by setting $\lambda := 2.5$ and $\bar{\mathbf{a}}$ to a vector of ones in Equation (5) for all experiments. We find this value of $\lambda$ to give us the best results for the attribute-blind re-writing [76].
- **Adversarial debiasing** we observe in Table 2 the same poor performance for the adversarial debiasing technique as is reported in Wang et al. [90]. We hypothesize that the same issues face the technique in our experimental setup. In particular, even in the absence of discriminative information for the 'gender' label in the final representation, information about correlated attributes (e.g. wearing makeup) are likely still present. This makes it fundamentally challenging to apply fairness-through-unawareness techniques in the CelebA multi-class setting.

## K  Fairness: additional results

### K.1  Model re-writing

The full per-subpopulation test set accuracies are shown in Figure 25 for the two experiments in the main paper. The first rows show the accuracies before layer re-write, the second rows after re-write, and the third rows the absolute difference between the two. As can be seen in the 'before-after difference' final rows of Figure 25, the proposed expert-conditional re-write provides much more precision in changing only the computation for the target populations.

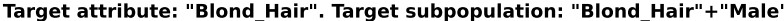

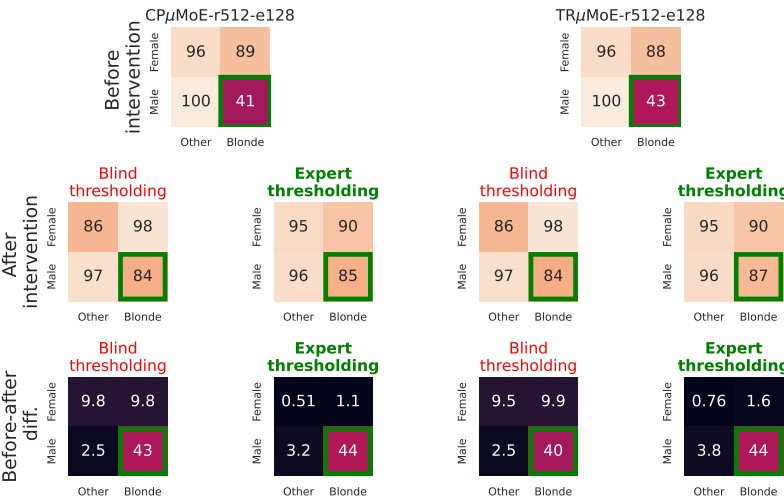

(a) 'Young blond' intervention for Blond hair attribute prediction head

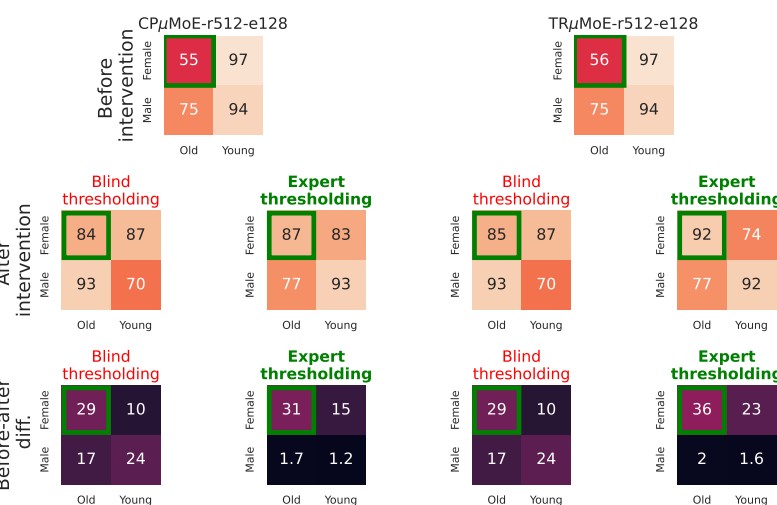

(b) 'Old female' intervention for age attribute prediction head

Figure 25: CelebA Subpopulation accuracies before (first rows) and after intervention (second rows), followed by their absolute difference (third rows). **Green rectangles** denote the target subpopulation for each experiment (subfigure).

