# OpenReview forum: "Multilinear Mixture of Experts: Scalable Expert Specialization through Factorization"
_NeurIPS.cc/2024/Conference — NeurIPS 2024 poster_

### Official Review · Reviewer_1sqG · 2024-07-13

**Soundness:** 4
**Presentation:** 3
**Contribution:** 4
**Rating:** 7
**Confidence:** 4

**Summary:**

In the paper, the author introduces $\mu$MoE layers, which factorize large weight tensors to facilitate implicit computation, thereby accelerating the computation process. By increasing the number of experts, the model enhances its specialization in vision tasks. Further, the author conducted experiments with the GPT2 and MLP-Mixer models, demonstrating that these models maintain high accuracy and exhibit increased specialization. Both qualitative and quantitative evaluations conducted by the author were commendably thorough and well-executed. The experiment is comprehensive.

**Strengths:**

1. **Innovative Metric:** The introduction of a quantitative metric for "class-level polysemanticity" is a novel approach to evaluating the specialization of experts within the model, providing a clear measure of the impact of increasing the number of experts.

2. **Comprehensive Evaluation:** The paper undertakes a broad range of experiments to validate the effectiveness of $\mu$MoE layers. The inclusion of both qualitative and quantitative analyses strengthens the argument for the utility of µMoE layers.

3. **Competitive Performance with Added Benefits:** The ability of $\mu$MoE layers to compete with larger models like GPT-2 and MLP-Mixer, coupled with the advantages of additional interpretability, is highlighted as a significant strength of the study.

**Weaknesses:**

1. **Limited Scope:** The research primarily focuses on vision models and specific datasets, which may limit the generalizability of the findings. Expanding the validation to more diverse models (LLaMA, gemma, and $e.t.c$) and datasets could provide a more comprehensive view of the $\mu$MoE layers' effectiveness.

2. **Lack of Robustness Analysis:** The paper would benefit from including experiments that assess the robustness and performance of µMoE layers on out-of-distribution data, offering insights into the model's reliability under varied conditions.

3. **Need for Implementation Details:** There is a notable gap in the disclosure of practical implementation details and computational requirements for µMoE layers. Addressing this could aid in the reproducibility and application of the findings.

**Questions:**

1. Could you specify the computational hardware and configuration utilized for these experiments?
2. Is it feasible to apply this methodology to large-scale MoE Language Models with a substantial number of experts, similar to the extensive parameter count found in Mixtral 8$\times$7B?

**Limitations:**

The authors adequately discuss the limitations and do not have the potential negative societal impact.

---

> ### Author Rebuttal · Authors · 2024-08-06
>
> We thank Reviewer 1sqG for their positive assessment of the paper; for praising the “innovative” quantitative metrics, the “comprehensive evaluation”, and “competitive performance with added benefits”. We address the stated weaknesses below:
>
>
> ## [W1] Limited model/dataset scope
>
> The reviewer is correct that we focus primarily on vision models and ImageNET1k. However, we highlight that our experiments also demonstrate the benefits of the layer on multi-modal vision-language models (CLIP and DINO in Figures 3, 14-17, and 21), and LLMs using the large and diverse Open WebText natural language dataset [1].
>
>
> ## [W2] Robustness analysis
>
> Our quantitative evaluation of vision models is only on the test set and does not include out-of-distribution (OOD) data, a limitation we have acknowledged in the conclusion.
>
> However, in our experiments on language models (as shown in Figures 5, 12, and 13), we use sequences generated by large language models (LLMs) as input instead of the training set data. Additionally, we conducted an extra experiment for Reviewer-qC51 (in the rebuttal PDF), where the input prompt is free-form text, demonstrating initial evidence of the experts' generalizability beyond the training and test data.
>
> We thank the reviewer for their suggestion to focus explicitly on robustness. We will accordingly include this explicitly in the limitations as a possible avenue for future work.
>
>
> ## [W3 + Q1] Need for implementation details
>
> We highlight that we include the full pseudocode of all muMoE layers in Appendix B, and an anonymous link in the paper's abstract links to our full model source code implementing the layers (please see `MuMoE.py` in particular).
>
> For full details about the computational requirements, hardware, and configurations used for the experiments, please see Table 6 in the appendix.
>
>
> ## [Q2] Large MoE models
>
> We do not anticipate any particular difficulties applying muMoEs to even larger networks.
>
> MuMoEs are indeed a uniquely attractive alternative at very large scales, unlocking (through their parameter efficiency) a route towards achieving fine-grained specialization that is not possible with Sparse/Soft MoEs due to their prohibitively large parameter counts.
>
> However, we unfortunately do not have access to the computational resources to verify this experimentally. As shown in Table 6, we have access to a maximum of 4 A100 GPUs, which limits us to pre-training only the smallest GPT-2 model with 124M parameters. We have added the following to the “limitations” section to acknowledge this:
>
>
>     Despite demonstrating the benefits of scalable expert specialization with muMoE layers for various architectures with different modalities, we have not conducted experiments on the largest scales of commercial LLMs. Future work should explore this where resources permit.
>
> ---
>
> - [1] Aaron Gokaslan and Vanya Cohen. Openwebtext corpus, 2019.

---

### Official Review · Reviewer_YTQ3 · 2024-07-19

**Soundness:** 2
**Presentation:** 4
**Contribution:** 1
**Rating:** 5
**Confidence:** 5

**Summary:**

The paper proposes the use of (factorized) Multilinear Mixture of Experts as an alternative to Sparse MoEs, which are prone to training issues due to the sparse top-k activation. Several factorization options are described and compared, which result in models of better accuracy in vision tasks, when matching the parameter count, compared to models using vanilla dense MLPs. The paper also includes experiments on a language task, where the proposed method is almost as good as the baseline.

**Strengths:**

- The proposed approach (using either the CP structure, or Tensor Ring structure, or both) achieves better accuracy than MLPs on image tasks, on two different architectures: MLP-Mixer S-16 and CLIP B-32, with models of similar parameter count.
- The discussion about different factorization strategies for the MLPs in the appendix is a delight. In fact, all the subsections in the appendix contain many useful details. I enjoyed reading several of them.
- The main paper contains a thorough and well written description of the proposed method, as well as the experiments. Any of the details and ablations that are missing from the main paper can be found in the appendix.

**Weaknesses:**

- The results on GPT-2 are close to the dense MLPs, but not better than them. This is problematic, since this architecture is the biggest one trained (with 124M params) and probably the one trained on the richest data source. This begs the question of whether the proposed approach can scale well to bigger backbone architectures and datasets.
- Appendix C.2 discusses why the low-rank assumption in MLPs is reasonable. In particular, a trained MLP-Mixer is taken and the rank of the matrices is reduced with truncated SVD. This shows that the low rank approximation is reasonable once the model is trained, but completely ignores the effect of this approximation _during training_, especially when the architecture size and training data scales.
- Most importantly: there is not quality comparison between the proposed approach and Sparse MoEs. The recipes for training Sparse MoEs have improved significantly, and they are present in most state-of-the-art language models, and multi-modal models (for instance, take a look at the [Mixtral report](https://arxiv.org/abs/2401.04088), [DeepSeekMoE](https://arxiv.org/abs/2401.06066), [Gemini 1.5 report](https://arxiv.org/abs/2403.05530), [Databrick's DBRX](https://www.databricks.com/blog/introducing-dbrx-new-state-art-open-llm) which is available on HuggingFace, and many others). In vision tasks, Soft MoEs in ["From Sparse to Soft Mixture of Experts"](https://arxiv.org/abs/2308.00951) have shown also much better results than vanilla dense transformers and Sparse MoEs. Thus, either of these should be considered as a stronger baseline than the use of vanilla MLPs.
- A discussion about the speed of the different methods is also missing from the main paper, only parameter count is considered, which may not be the only (or main) limiting factor.

-----------
During the rebuttal the authors satisfactorily addressed two of my weaknesses. They also pointed out that the method is not an alternative to Sparse MoEs, but to provide a better parametrization of standard (dense) models. I still believe that Sparse MoEs would be a better baseline than Dense MoEs (formerly referred to as Soft MoEs in the paper), as I mentioned in my comment to the author's rebuttal. I also find concerning the fact that when the method is applied on relatively large models, it provides little benefit, although I acknowledge that it may be more interpretable. Given this, I'm slightly increasing my initial rating from "Reject" to "Borderline accept".

**Questions:**

This is a suggestion, not a question: The term "Soft MoE" used to refer to the seminal work of Robert A. Jacobs et al. from 1991 can be confusing with the recent work ["From Sparse to Soft Mixture of Experts"](https://arxiv.org/abs/2308.00951) by Puigcerver et al, from last year. Since the term Soft is only used once in the 1991 work, I would suggest using "Dense MoE" to refer to it, which is also more opposed to "Sparse MoE".

**Limitations:**

No negative societal impacts are specific to this work, in my opinion.

---

> ### Author Rebuttal · Authors · 2024-08-06
>
> We thank Reviewer YTQ3 for engaging thoroughly with the paper. However, we respectfully argue below that the reviewer overlooks our stated goals and claims in the paper, and the benefits to scalable expert specialization the methodology brings that are otherwise infeasible:
>
> In particular, muMoE layers are a parameter-efficient method for scalable expert specialization. **MuMoEs aim is to provide interpretability and controllability benefits to dense layers without increasing the parameter or FLOP counts** of the original networks. We respectfully highlight that these goals and scope are outlined in the paper’s claims (e.g. [L17, L81, L353]), and that two reviewers praise this explicitly:
>
> - **Reviewer XreH**: `"specialization of experts is one of the most important goals in MoE".`
> - **Reviewer Su68:** `"µMoE models demonstrate a more parameter-efficient alternative to traditional MoEs, an important consideration for practical applications."`
>
> Concretely, as detailed from [L344], our experiments comparing model accuracy in Section 4.3 serve only to verify that muMoE layers do not trade-off accuracy relative to existing dense layers of both equivalent parameters and FLOP counts. **Sparse MoEs introduce significantly more parameters; thus, there is no possible fair comparison in this setting.**
>
> In the paper, we make no claims about competing with Sparse MoEs’ performance—only that our model form avoids Sparse MoEs’ non-differentiability by design. Rather than exploring conditional computation for increasing performance and network capacity, the paper instead has the orthogonal goal of facilitating more interpretable subcomputations (e.g. allowing downstream tasks/properties of manual bias correction, interpretability, or network editing/steering as shown in the rebuttal PDF), without introducing any additional parameters over MLPs.
>
> We would be grateful if the reviewer could re-assess the work with the paper’s stated goals and claims in mind, and to judge the method through the novelty it brings to scalable expert specialization at no additional computation cost--along with its downstream benefits shown in the paper.
>
>
> ## Sparse/Soft MoE comparison results
>
> **We emphasize the significance of muMoE’s parameter-efficient expert specialization:** even at the moderately large expert counts we explore in this paper, Sparse MoEs quickly approach a prohibitively large number of parameters. Even the GPT2-small model with a 2048 expert Sparse MoE has on the order of 100 billion MLP parameters--compared to just ~100 million parameters with CPmuMoEs. Meanwhile, the GPT2-XL model with 2048 expert Sparse MoE layers has on the order of a trillion MLP parameters alone. For even larger dense models whose base parameter counts are already in the billions, the large expert counts necessary for fine-grained specialization are infeasible with Sparse MoEs.
>
> We accordingly focus on an experimental setup in which it *is* possible to use large numbers of experts in both Sparse and Soft MoEs--without an infeasible number of parameters or FLOPs. In this setting, the computational requirements of all MoEs can be matched fairly. Please find in the global rebuttal PDF (at the top of openreview) a comparison of muMoEs to both Soft and Sparse MoEs for fine-tuning CLIP on ImageNET1k: **whilst it is not the goal of the layer, both variants of muMoEs provide Pareto-optimal improvements to the parameter count vs accuracy frontier.**
>
> Furthermore, as outlined above, the paper’s aim is not to increase accuracy or replace existing MoEs — but rather to replace parameter- and FLOPs-matched dense layers with interpretable subcomputations. This is why we compare dense layers as a baseline and not transformer-specific MoE layers such as the suggested [1].
>
> What’s more, we highlight that the MoE and routing strategy in “From Sparse to Soft Mixtures of Experts” [1] is tailored to transformer/token-based architectures, and the method and paper aim at improving transformer model performance. In contrast, the muMoE focuses instead on scalable expert specialization, and in **dense layers** more generally. As demonstrated in the paper, our focus is on facilitating applications such as last-layer fine-tuning models with MoEs (and subsequently removing model bias in Table 2). These tasks addressed in the paper are not possible with the token-specific methodology of [1].
>
>
> ----------
> - [1]: Puigcerver, Joan et al. “From Sparse to Soft Mixtures of Experts.” ICLR 2024.
>
>
> ## Low rankness at train-time
>
> The reviewer is correct that Appendix C.2 explores the minimal impact of applying truncated SVD on *pre-trained* weights. However, all the results in Section 4.3 of the main paper **do** validate the ability of the muMoE layer to retain performance with rank constraints **at train time**. As stated on [L769], this is our main evidence to support the claims of the paper.
>
>
> ## Benchmarking of speed & memory
>
> Please find (in the rebuttal PDF) a summary of latency and peak memory usage for various layers. We use the same number of experts for all MoE variants and set the rank of muMoEs to match the parameter count of a linear layer mapping from 768 to 1000 dimensions. Whilst muMoEs do not offer significant speed gains over alternative MoEs, they have an order of magnitude less peak memory usage. We will include a discussion of these results in the revised main paper with the additional page count.
>
> ## “Dense MoEs” suggestion
>
> We thank the reviewer for the helpful suggestion to use “Dense MoE” to refer to the baseline methodology. Not only does such a name accurately describe the properties of this model's forward pass, but it indeed helps distinguish it from the recent ICLR paper. We will gladly adopt this name in the paper.

---

> ### Author Response · Authors · 2024-08-12
>
> Dear Reviewer-YTQ3,
>
> We are grateful for the thorough and thoughtful review of our submission.
>
> The reviewer's primary concern is missing comparisons to Sparse MoE's capabilities. We kindly draw our attention to rebuttal, where we provide **new requested experiments showing that the proposed muMoE layer offers Pareto-optimal improvements to accuracy over Sparse and Soft MoE**. We provide further requested experiments to benchmark speed and peak memory usage and discuss all remaining concerns raised.
>
> Additionally, we provide **clarifications about the goal and claims of the paper as providing scalable expert specialization to dense layers without increasing either the FLOPs or parameter counts** (orthogonal to the aim of pushing capabilities through more parameters/FLOPs).
>
> If there are any other questions or concerns that we can address to enhance your support for our paper, we would greatly appreciate the opportunity to respond.
>
> Sincerely,
> Submission #3407 Authors.

---

> > ### Author Response · Authors · 2024-08-13
> >
> > Dear **Reviewer YTQ3**,
> >
> > We eagerly await your response to our rebuttal where we address your primary concerns including the requested comparisons to sparse and soft MoEs and the requested benchmarking for speed.
> >
> > We highlight that **Reviewer qC51** had a similar question about comparisons to Sparse MoEs in their initial review, and has since firmly increased their score after our rebuttal and experimental comparisons. In addition to this, **Reviewer XreH** also increased their score following our rebuttal and new experiments results.
> >
> > We very much appreciate the insights and attention to detail provided in your initial review, and we would be very grateful for the chance to respond to any remaining concerns or questions you have that would increase your support for the paper.
> >
> > Sincerely,
> >
> > Submission #3407 Authors.

---

> > > ### Comment · Reviewer_YTQ3 · 2024-08-13
> > >
> > > I appreciate very much the author's response to my initial review.
> > >
> > > I do understand better now the focus of the paper. I still believe believe that Sparse MoEs are a valid baseline: if Dense MoEs (or Soft MoEs as called in the initial version of the paper) are a valid baseline, which have a huge memory and runtime footprint, why not using instead Sparse MoEs which have the same memory footprint but far less runtime cost?
> > >
> > > My concern about the scalability of the approach (i.e. not so good results on GPT-2, where models are bigger) was not satisfactorily addressed in your rebuttal.
> > >
> > > Nevertheless, the other two weaknesses that I pointed out have been clarified. I also have to admit that, perhaps, I initially scored the paper too harshly, since I misjudged the focus of the paper. I apologise for any distress that this may have caused to the authors, and will increase my score accordingly.

---

> > > > ### Author Response · Authors · 2024-08-13
> > > > **Thanks to Reviewer YTQ3; responses and suggested additions to paper's "limitations"**
> > > >
> > > > Dear Reviewer YTQ3,
> > > >
> > > > Thank you kindly for your time and effort in engaging with our rebuttal.
> > > >
> > > > We are delighted to hear that we have addressed two of your primary concerns and that the goals of the paper have been well-clarified. We are grateful for the increase to the paper's score.
> > > >
> > > > Your comments have been invaluable in helping improve the clarity of the paper's aims and objectives in the revised paper. We will make it clear throughout the revised text that the layer does not have the goal of outperforming the *accuracy/loss* of either Soft or Sparse MoEs. Regardless, we will include all comparisons to both Soft and Sparse MoEs and supporting discussions from this thread in the camera-ready version.
> > > >
> > > > Regarding the final concern, the reviewer is absolutely correct to note that the original GPT-2 model indeed has a slightly lower validation loss (e.g. 0.01 below the TRMuMoE GPT-2 model), despite not providing any interpretable expert specialization of the muMoEs. Furthermore, whilst we don't anticipate any problems unique to muMoEs when scaling to even larger models (due to the layers' parameter-efficiency), we acknowledge that we have not validated this experimentally.
> > > >
> > > > Inspired by the reviewer's comments, we propose to include the following limitation to the revised paper to acknowledge this explicitly (jointly in response to **Reviewer 1sqG**):
> > > >
> > > > ```
> > > > Despite demonstrating the benefits of scalable expert specialization with muMoE layers for various architectures with different modalities, we have not conducted experiments on the largest scales of commercial LLMs. Furthermore, our experimental results demonstrated comparable accuracies of muMoE networks for models only up to parameter counts on the order of 100 million. Where resources permit, future work should explore the scalability of expert specialization and performance of muMoEs in even larger-scale LLMs.
> > > > ```
> > > >
> > > > We thank the reviewer once again for their careful and detailed assessment of the paper. We are more than happy to listen to any other suggestions they might have for the paper.
> > > >
> > > > Sincerely,
> > > >
> > > > Submission #3407 Authors.

---

### Official Review · Reviewer_XreH · 2024-07-23

**Soundness:** 2
**Presentation:** 2
**Contribution:** 2
**Rating:** 5
**Confidence:** 5

**Summary:**

This paper proposes the Multilinear Mixture of Experts (µMoE) layer for scalable expert specialization by performing an implicit computation on prohibitively large weight tensors entirely in factorized form. Both qualitative and quantitative results show that scaling µMoE layers when fine-tuning foundation models for vision tasks leads to specialized experts at the class level.

**Strengths:**

* The problem studied in this paper is well-motivated, specialization of experts is one of the most important goals in MoE.
* The formulated illustration of factorization is clear.
* Figures and tables are clear and easy to read.

**Weaknesses:**

* It would be better if more results on language models could be included.

**Questions:**

* The description of the dense µMoE layer in Section 3.1 is a little bit confusing to me, for example, what is the difference between it and the soft MoE layer?
* According to the methodology description, e.g. in Section 3.1.1, there is no “top-k” routing in the µMoE layer. Nevertheless, how are the “top-activating” experts selected in Figures. 4 & 5?

**Limitations:**

Yes.

---

> ### Author Rebuttal · Authors · 2024-08-06
>
> We thank **Reviewer-XreH** for praising the well-motivated problem studied in the paper and the clear presentation of the factorization methodology.
>
>
> ## More results on language models
>
> Whilst we do not have the computational resources to pre-train new additional large language models during the rebuttal, we do provide more LLM results on the GPT-2 language models already trained for **steering the LLM outputs using the experts,** inspired by the reviewer’s comments.
>
> In particular, we use a GPT-2 model trained with muMoE layers at each MLP layer, using 2048 experts (at every layer). By modifying the forward pass of the trained model—specifically, adding selected expert cluster center vectors to each token's input latent activation vector before applying the muMoE layer—we can consistently control the model to generate outputs aligned with specific themes. Illustrations of this approach, using 3 different experts, are included in the global rebuttal PDF.
>
> The selected experts guide the language model's outputs toward discussing topics such as climate change, police brutality, or foreign politics. We suggest that these findings further demonstrate the effectiveness of the muMoE layer in facilitating controllable generation of language model outputs—all whilst requiring no additional training.
>
>
> ## [Q1] muMoE vs Soft MoE
>
> Soft MoEs perform a dense forward pass taking a linear combination of all $N$ experts’ outputs simultaneously. With large expert counts, this is problematic due to both high parameter counts and restrictively high FLOP costs.
>
> As detailed in Section 3.1.2, the existing soft MoE is a special case of muMoE when (a) the weight tensors are explicitly stored without decomposition and (b) no implicit fast factorized computation is performed. The key technical contribution of the muMoE layer is to make a similar full dense forward pass (using *all* $N$ experts simultaneously) feasible at high expert counts. In contrast, muMoEs elegantly skirt both issues with Soft MoE: the former (a) through factorization, and the latter issue (b) through the derivations of the fast implicit forward passes.
>
>
> ## [Q2] How to find top-activating experts
>
> The reviewer is correct that we apply no top-k routing at training time, nor at test time. After the model is trained however, we can still rank inputs by their expert coefficients for the qualitative experiments. Concretely, we pass a dataset of inputs through the network and rank images/text-token inputs based on the magnitude of the corresponding expert coefficient (the strength with which each expert is activate). It is through this standard technique that the top-activating images are ranked and displayed in Figures 4 and 5.
>
> As shown in the “Expert coefficients color map” in the top-right of Figure 5, the color indicates the soft expert coefficient weight (on the interval [0,1]) with which this particular expert processes each text token.

---

> > ### Comment · Reviewer_XreH · 2024-08-10
> >
> > Thank you for the detailed response to the reviews and additional experimental results. After reading all the reviews and responses, I will raise my score to 5.

---

> ### Author Response · Authors · 2024-08-10
> **Thanks to Reviewer XreH**
>
> We're pleased to hear that the reviewer has raised their score following the additional experiments and rebuttal. We wish to thank Reviewer XreH for their time and efforts in engaging with our response.
>
> Are there any additional questions about our work we can answer that would further increase the reviewer's rating of the paper? Please do not hesitate to ask us any further questions you may have.
>
> Best,
>
> Submission #3407 Authors.

---

### Official Review · Reviewer_qC51 · 2024-07-26

**Soundness:** 2
**Presentation:** 2
**Contribution:** 2
**Rating:** 6
**Confidence:** 5

**Summary:**

The paper proposes a new way of computation in a mixture of expert (MoE) layer. Unlike SparseMoE, instead of using the top-K operator and then doing full computation only for that expert, this paper do not do topK, but does a linear combination of all experts, but each expert is now a low rank matrix, which saves compute/parameters. The authors do show some properties that performing MoE computation in such a way can help in interpretability of the experts, their task specialization and so on. This can further help in certain cases like to mitigate biases.

**Strengths:**

- The idea seems to be simple enough to be easily implemented in existing MoE setups.
- Anecdotal and some statistical results suggests that it does help in  mitigating bias in certain cases in a constrained setting.

**Weaknesses:**

- Figure 2 can be made much stronger by showing statistical differences between the two different cases -- like entropy of experts, overlap among experts, distribution, etc, and also comparing it with similar parameter and/or compute sized sparse/soft MoEs.

- Again, for Figure 3, there should be a way to compare it at least with sparse MoE.

- Comparison with Sparse and SoftMoE in vision and LLMs are missing -- keeping the same parameter size and/or compute fixed across algorithms, as well as how does performance scale with more parameters in the MoE part. These are essential, as increasing the capacity of the network by increasing parameter count, while preserving the same compute is one of the main motivations behind MoEs, other than the interpretability lens which the works focuses mostly on.

- Several ablations seems to be missing -- performance improvements when you increase number of experts, placement of the MoE layers, experiments on different model sizes, larger datasets, effect of different non-linearities in the gating function, etc.

**Questions:**

- As mentioned above, I think it is quite important to show how \muMoE perform compared to SoftMoE and SparseMoE.

- Making Figure 2 more statistically significant rather than anecdotes.

- More experiments needs to be conducted to showcase the efficacy of the interpretability of the experts and its use cases. And how are those benefits compared to Soft and SparseMoE.

Suggestions:
- I think the notation used in Eqn 1 is a bit difficult to follow, specifically with the colons used without brackets.

**Limitations:**

The authors have identified the limitations of the work.

---

> ### Author Rebuttal · Authors · 2024-08-06
>
> We thank **Reviewer-qC51** for their detailed review; for praising the ease with which the methodology can be implemented and the benefits of downstream applications in bias mitigation. We address the stated weaknesses below and draw attention to the existing ablation studies requested. Additional experiments requested to further show the benefits of the interpretability gained are also provided through steering LLMs’ outputs using the experts.
>
> ## Comparisons to Sparse/Soft MoE
>
> As the reviewer correctly states, our sole interest is the independent benefits of scalable expert specialization (these are the claims and paper's goals as stated in [L17, L81, L353]). **muMoEs are parameter-efficient layers for providing interpretability and controllability benefits without increasing the parameter or FLOPs count** of the original networks. We highlight that two reviewers praise this explicitly:
>
> - **Reviewer XreH**: `"specialization of experts is one of the most important goals in MoE".`
> - **Reviewer Su68:** `"µMoE models demonstrate a more parameter-efficient alternative to traditional MoEs, an important consideration for practical applications."`
>
> Consequently, we argue that the fair and relevant comparison to validate the paper’s claims is **with a fixed computational budget**: between the original networks with MLP layers and parameter- and FLOPs-matched muMoE layers. This serves to verify that we do not trade-off accuracy when introducing the many benefits of muMoE layers. We do not claim (nor desire) to *increase* parameter counts / capacity / performance. Sparse and Soft MoEs significantly increase the number of parameters and the number of FLOPs, respectively, making a fair comparison impossible.
>
> In contrast, Soft and Sparse MoEs mostly aim to increase *performance* through increased parameter counts and/or FLOPs. Crucially, fine-grained expert specialization in large models is impractical with both Soft MoEs and Sparse MoEs. For Soft MoEs, a restrictively high FLOP count is required. For Sparse MoEs, the parameter count alone is problematic: even the smallest GPT2 variant with 2048 expert Sparse MoE layers has on the order of 100 *billion* MLP parameters--compared to just ~100 *million* parameters with muMoEs. **For even larger dense models whose base parameter counts are already in the billions, large expert counts with Sparse MoE are infeasible.**
>
>
> ## Requested comparisons
>
> Inspired by the comments of the reviewer, we've included a new detailed comparison of muMoEs with both soft and sparse MoEs for CLIP fine-tuning (in the rebuttal PDF). This single-layer experimental setup makes it possible to use an otherwise prohibitively large number of experts in Sparse/Soft MoEs, all while maintaining a feasible number of parameters and FLOPs.
> Although optimizing the parameter count / FLOPs VS accuracy trade-off is not the aim of the paper, both muMoE variants achieve Pareto-optimal improvements in this aspect for last-layer fine-tuning of CLIP on ImageNET1k.
>
> ## Ablations
>
> We kindly draw the reviewer’s attention to the following figures where **the** **requested ablations** **can** **already** **be** **found in the submission**:
>
>
> - **Performance increase as a function of expert count**: please see Figure 21 of the appendix ([L933]) where performance increases as expert count is increased.
> - **Placement of muMoE layers**: please see Figures 3, 16, 17 for final and penultimate layer muMoEs configurations. We note that Section 4.3’s results are with networks with muMoEs at *every* layer, and results from *all* layers are shown in Figure 11 of the appendix.
> - **Different model sizes**: experiments are already performed throughout on 4 network architectures of various sizes (CLIP, DINO, GPT2, and MLP Mixers).
> - **Gating function**: An ablation study is already presented on [L888] between activation functions.
>
>
> ## Additional use cases of experts
>
> As requested, to further demonstrate the usefulness of the interpretability and experts, we show how in GPT2 models, the experts can be used to **steer the model’s output to particular themes of interest**.
>
> Concretely, we take a GPT-2 model trained with CPmuMoE layers at every MLP layer, each with 2048 experts. We find that intervening in the trained model’s forward pass (concretely: adding particular expert cluster center vectors to every token’s input latent activation vector before applying the muMoE layer), reliably steers the model to produce outputs related to specific themes. Examples are shown for 3 experts in the rebuttal PDF at the top of the page.
>
> Chosen experts steer the LLM outputs toward talking about themes such as climate change, police brutality, or programming. We suggest these results constitute further evidence of the usefulness of the muMoE layer in providing controllable generation of LLM outputs (without any additional training needed).
>
>
> ## Figure 2
>
> Our experiments are separated into qualitative studies (Section 4.1.1.) and quantitative studies (Sections 4.1.2 and 4.2). The sole purpose of Figure 2 is a qualitative study to visually motivate the benefits of increasing the expert count. Separate quantitative results are found throughout the latter two subsections. Furthermore, the distribution of expert assignments is already visualized in Figure 20 of the appendix.

---

> ### Author Response · Authors · 2024-08-12
>
> Dear Reviewer-qC51,
>
> Thank you for your thorough review of our submission.
>
> The reviewer's key issues were about comparisons to Soft/Sparse MoE and missing ablation studies. In our rebuttal, we highlight where **each of the requested ablations can be found** throughout the main paper and the appendix. Furthermore, we address the reviewer's primary issue by **including new comparisons to both Soft and Sparse MoEs, where the muMoE layer strictly outperforms the two existing alternatives**.
>
> If there are any other questions or concerns that we can address to further strengthen your support for our paper, we would be eager to respond.
>
> Sincerely,
> Submission #3407 Authors.

---

> > ### Comment · Reviewer_qC51 · 2024-08-12
> > **Clarifying questions**
> >
> > Thanks to the authors for the extensive answers to the comments. While it answered most of my concerns, I have a few clarifying questions.
> >
> > 1. Figure 1 (right) of rebuttal document, how do you get mutliple points for Linear, SparseMoE and SoftMoE? Is the dimension of MLP changed? or applied on multiple layers?
> > 2. In Figure 21 of Appendix, while there is a slight increase in performance (max diff of +0.3%) for TR\mu MoE, there is hardly any difference for CP\mu MoE. Is there any explanation behind it?
> > 3. While I understand the authors did experiments, visualizations with \mu MoE in different configuration - final layer, penultimate layer or all layers. Is there any conclusion that one can draw about the placement of expert layers from interpretability or performance and the trade-off if any?

---

> ### Author Response · Authors · 2024-08-13
> **Responses to additional questions**
>
> Thanks to Reviewer-qC51 for their response.
>
> **We are glad to hear that our rebuttal has answered most of your concerns.**
> We thank the reviewer for their further insights and chance to answer the remaining questions below:
>
> ## 1. Varying parameter counts of baselines
>
> As described in the Caption of Figure 1, we increase the Soft/Sparse MoEs' parameter counts by increasing the expert counts.
>
> We also use a linear transformation (to match the functional form of the baseline layers), but instead compose the weight matrix as a product of two tall matrices $\mathbf{W}_1$ and $\mathbf{W}_2$, whose number of rows are varied to increase the linear layer's parameter count. Concretely we compute:
> $(\mathbf{W}_1^\top\mathbf{W}_2)\mathbf{x}+\mathbf{b}$.
>
> We are thankful to the reviewer for the attentive study of our responses. We will include these details in the revised version.
>
> ## 2. CP vs TR
> Figure 21 of the appendix demonstrates that **parameter-matched** muMoEs *strictly outperform* linear layers in this experiment, for *all* possible choice of expert counts.
> Even with equal parameter counts, we see performance gains of up to 0.7% above the linear layer (which is non-trivial on ImageNET).
>
> However, to parameter-match the single linear layers, we must decrease the muMoE layer rank upon increasing the expert count.
> Crucially, as we see from Figure 7 of the appendix, TRMuMoEs are often much more parameter-efficient at high expert counts (a result we discuss in depth in Section C.1).
>
> As a consequence of this, **TRMuMoEs' resulting expert matrix ranks are increasingly larger than that of CPMuMoEs**, even when the layers are parameter-matched. For example, the parameter-matched layers with 512 experts in Figure 21 have an expert matrix rank of 165 for the CPMuMoE compared to a much larger 208 for the TRMuMoE.
>
> We attribute TRMuMoE's even greater performance gains over CPMuMoEs to the more favorable relationship between tensor rank and expert matrix rank (a larger weight matrix rank meaning the resulting layers' activations live in a larger dimensional subspace).
>
> We will include a table in the revised version comparing the matrix ranks to the expert counts in this experiment, and include this discussion in the revised supplementary material.
>
> ## 3. Implications of layer placement
>
> Our most noteworthy observation about the implications of layer placement is in vision models. As shown in Figure 4 and discussed in [L326], we find that MuMoEs at early layers exhibit specialism to more coarse-grained patterns (e.g. texture and colors), whilst later layers exhibit more fine-grained expert specialisms (to classes and objects). Please also observe this in Figure 11 of the appendix for both MuMoE variants. To our knowledge, there is no significant trade-off when varying placements of the layer in terms of training resources or final performance.
>
> ---
>
> We thank the reviewer again for the insightful discussion. Might they have any further questions we could answer to increase their score?
>
> Sincerely,
>
> Submission #3407 Authors.

---

> > ### Comment · Reviewer_qC51 · 2024-08-13
> >
> > Thanks to the authors for the answers.
> > I have increased my score to 6.
> > I strongly suggest the authors to include all the details/discussions presented here, to be included in the main paper.

---

> > > ### Author Response · Authors · 2024-08-13
> > > **Thanks to the reviewer**
> > >
> > > Dear Reviewer qC51,
> > >
> > > We thank you for the **strong support** and confidence in our paper.
> > >
> > > We will include the linear layer experimental setup details and discussions about the 2 further insightful questions asked by the reviewer (regarding CP vs TR, and the significance of muMoE layer placement) in the revised main paper. Furthermore, we are open to any additional suggestions the reviewer might have to improve the paper.
> > >
> > > Sincerely,
> > > Submission #3407 Authors.

---

### Official Review · Reviewer_Su68 · 2024-07-29

**Soundness:** 3
**Presentation:** 3
**Contribution:** 3
**Rating:** 4
**Confidence:** 4

**Summary:**

The paper presents an interesting and novel approach with the µMoE layer, offering a method for scalable expert specialization. Extensive experiments in multi-dimensional have been conducted to show the effectiveness of the proposed method. However, the marginal improvements in results, coupled with several other weaknesses, including experimental variability, novelty of factorization, clarity of metrics, and scalability, limit the overall impact of the work.

**Strengths:**

1. Introducing the Multilinear Mixture of Experts (µMoE) layer is novel and addresses the computational cost challenges associated with scaling the number of experts for fine-grained specialization. The mathematical formulation of µMoE layers is thorough, and the factorization approach is well-explained and technically sound.
2. Practicality: The proposed µMoE models demonstrate a more parameter-efficient alternative to traditional MoEs, an important consideration for practical applications.
3. The paper includes extensive qualitative and quantitative evidence to support the claims. Experiments on vision and language models illustrate the versatility and effectiveness of the proposed method.
4. This paper is well-written and easy to follow.

**Weaknesses:**

1. The idea of factorization is not new, and it seems that the authors simply apply factorization in the MoE context. The authors should specify and address any new challenges that arise when applying factorization in the MoE context.
2. Marginal Improvements and lack of multiple runs: The experimental results in Table 2 show only marginal improvements over existing methods, which might not be compelling enough to justify the additional complexity introduced by the µMoE layers. All experiments were only run once without reporting mean and standard deviation. It is recommended that the authors add results from at least three runs to provide a better understanding of the variability and reliability of their results.
3. The metrics used in the paper are not clearly explained. The authors should include a paragraph either in the experiments section or in the appendix to explain the metrics used.
2. The evaluation captures expert behavior only on the test set, not on out-of-distribution data, which is crucial for understanding the robustness and generalizability of the model.
4. While the paper qualitatively demonstrates interpretability, it lacks objective, quantitative measures of the interpretability gains compared to other methods.
8. It is unclear how the proposed methods perform across different model architectures and sizes. The authors should discuss whether the method scales well and provide evidence to support their claims.

More Comments:
1. The paper is generally well-written and clearly presents the proposed methodology and results. However, some sections, particularly those dealing with mathematical formulations, could benefit from additional explanations or visual aids to enhance comprehension.
2. Addressing the limitations and adding some discussions, such as out-of-distribution performance and quantitative measures for interpretability, would significantly strengthen the paper. Further research should also focus on demonstrating more significant performance gains and improving the ease of implementation to enhance the practical utility of the proposed method.

**Questions:**

N/A

**Limitations:**

Yes, the authors addressed limitations and the direction for future research.

---

> ### Author Rebuttal · Authors · 2024-08-06
>
> We thank **Reviewer-Su68** for their detailed assessment of the paper: for praising the “novel[ty]” of the layer and the “technically sound” methodology. Furthermore, we are glad the reviewer appreciates the “extensive qualitative and quantitative evidence to support the claims” of the paper.
>
> All stated weaknesses are addressed below, pointing out where the requested details can be found:
>
> ## [W1]: Factorization challenges
>
> Low-rank factorization is indeed a well-known technique in deep neural networks, frequently employed to reduce the number of parameters. However, as far as we are aware, our work is the first to introduce a novel technical link between hierarchical MoEs and multilinear models. Instead of utilizing factorization to simply reduce parameter counts, our work provides a way of designing trainable layers with prohibitively large numbers of experts in the first place. **This is only possible through the paper’s extra derivations and formulations of MuMoE layers’ fast forward passes** (beyond simple factorization).
>
> We have an in-depth discussion in Appendix C [L722] of how the factorization choice impacts MoEs. For example, we derive the theoretical relationship between tensor and subsequent expert matrix rank, and the relative strengths of different factorizations in this applied context. Furthermore, in Appendix C.2 we also discuss the impact of low-rankness in the specific context of the models studied in this paper. We believe this further detailed explanation and exploration of the methodology also addresses the reviewers’ second comment about further enhancing comprehension.
>
>
> ## [W2]: Marginal Improvements and lack of multiple runs
>
> As stated on [L293], the **results in Table 2 are already the mean over 10 runs**. We have added this to the caption to make it more clear. Furthermore, an improvement of (for example) 5% to the min-max fairness metric (for “age”) and half the equality of opportunity (for “blond hair”) are significant improvements over existing techniques.
>
> ## [W3]: Metrics not clearly explained
>
> **The metrics used in the paper are** **described in detail** **in Appendix I** ([L958]), as written on [L307] of the main paper. Furthermore, we describe in depth their interpretations in the context of the celebA bias problem studied, and the implementation details of the baselines. We will also explicitly reference this appendix discussing the metrics in Table 2’s caption.
>
> ## [W4]: Evaluation on OOD data for vision models
>
> Indeed, our quantitative evaluation of vision models is limited to the test set (and not OOD data), which we already noted as a limitation in the conclusion.
>
> Despite this, our experiments on language models in Figures 5, 12, and 13 use *generated* LLM sequences as their input (rather than that of the training set). Furthermore, we also perform an additional experiment on steering LLMs for **Reviewer-qC51**, where the input prompt is free-form text. This provides initial evidence of the generalizability of the experts outside of the training/test data.
>
> ## [W5]: Lacking quantitative measures of interpretability
>
> We kindly draw attention to both Section 4.1.2 and Section 4.2. In the former, a new quantitative metric capturing expert specialization is introduced. In the latter, interpretability is further measured quantitatively through its downstream ability to edit models compared to existing methods. We highlight that **Reviewer-1sqG** praises the “innovative metric", stating that it `"provide[s] a clear measure of the impact of increasing the number of experts"`. If the reviewer has any other valid metric in mind, we will be glad to consider it in our evaluation.
>
>
> ## [W6]: Different model architectures and sizes
>
> We kindly note that we already perform experiments across **four different architectures**: CLIP, DINO, MLP-Mixer, and GPT-2 (in Figure 3 and Table 4). These models are each of vastly different sizes and configurations (and operate on various modalities including language and vision), already demonstrating the wide applicability of the layer.
>
> That said, we have access to a maximum of 4 A100 GPUs, limiting the size of the models we can pre-train to GPT2-sized models. We have added a limitation to the paper’s conclusion to note explicitly how we have not tested the layer at the largest scales of commercial LLMs:
>
>
>     Despite demonstrating the benefits of muMoE layers for various architectures with different modalities, we have not conducted experiments on the largest scales of commercial LLMs. Future work should explore this where resources permit.
>
> For the camera ready, we will include additional experiments on a GPT-2 model trained from scratch with 2048 expert CPMuMoEs at each layer, to provide further evidence of the layer’s benefit at very large expert counts.
>
> ## “Ease of implementation” and visual illustration
>
> We would like to draw attention to Appendix B, which contains pseudocode implementations of all muMoE variants. Furthermore, we note that the `MuMoE.py` file in the anonymous code linked to in the abstract provides a full implementation of the layers in PyTorch.
>
> To aid with visual illustration, we plan to include an annotated variant of Figure 1 in the appendix that shows a step-by-step explanation of each stage of the computation of the forward pass for clarity.
>
> We thank the reviewer for their suggestion to improve the ease of implementation of the methods and to include additional visual aids to improve comprehension of the method.

---

> ### Author Response · Authors · 2024-08-12
>
> Dear Reviewer-Su68,
>
> Thank you for your time and effort in providing the review of our submission.
>
> In our rebuttal, we clarify the reviewer's key concerns: that **the paper's fairness results are indeed over repeated runs, and draw attention to where explanations of the metrics can be found**. We also provide additional clarifications and responses to all further questions raised.
>
> We would like to ask if the reviewer has any additional questions we are able to answer that would increase their support in the paper?
>
> Sincerely,
> Submission #3407 Authors.

---

> > ### Author Response · Authors · 2024-08-13
> > **Might Reviewer-Su68 have any remaining questions?**
> >
> > Dear Reviewer-Su68,
> >
> > We are keenly awaiting your feedback on our rebuttal, where we clarify your key concerns: that the fairness results are indeed based on repeated runs and highlight where the metric explanations are detailed.
> >
> > As the author-reviewer discussion is almost finished, we would appreciate an opportunity following our rebuttal to answer any additional questions you have that would increase your score of the paper.
> >
> > Sincerely,
> >
> > Submission #3407 Authors.

---

### Author Rebuttal · Authors · 2024-08-06

We are grateful to each of the 5 reviewers for their thorough comments; for their positive assessments of both the paper and the novelty of the layer in facilitating parameter-efficient, scalable expert specialization:


- **Reviewer Su68** praises the “thorough” mathematical formulation, “technically sound” factorization approach, and `"extensive qualitative and quantitative evidence to support the claims"`. They also highlight the importance of the parameter-efficiency of muMoEs: `"a more parameter-efficient alternative to traditional MoEs, an important consideration for practical applications".`
- **Reviewer qC51** highlights the ease with which the method can be implemented in existing MoE setups and improvements to mitigating bias.
- **Reviewer XreH** also praises the paper’s motivation (`"specialization of experts is one of the most important goals in MoE".`), and states that the technical formulations of the layer are clear.
- **Reviewer YTQ3** highlights the model performance over MLPs and the detailed appendix and technical details. They also summarize the main paper as thorough and well-written.
- **Reviewer 1sqG** praises the “innovative metric” introduced to quantify the expert specialization, the “comprehensive evaluation”, and the benefit of the layer as providing `"Competitive Performance with Added Benefits"`.

Our initial response addresses all of the stated weaknesses and questions. In the instances where reviewers have asked for experiments that already exist, we note explicitly in which sections they are located.

We highlight **three** new sets of experiments requested by various reviewers (which can be found in the rebuttal PDF):

1. **Comparisons of the muMoE layer to both Soft and Sparse MoEs** for CLIP fine-tuning: the muMoE offers Pareto-optimal improvements to accuracy over both MoE alternatives and dense layers.
2. **Specialized experts’ additional ability to steer the generation of GPT-2 muMoEs**: adding specific experts’ gating vectors to the activations reliably causes GPT-2 models with muMoEs to produce text of specific themes (such as about politics, or the weather).
3. Peak memory usage (PMU) and latency metrics comparisons between MoE layers, where muMoEs have an order of magnitude less PMU than Sparse MoEs.

We are happy to elaborate further on any remarks made in the rebuttal.

---

### Comment · Area_Chair_LvsV · 2024-08-13

Dear Reviewers

This is another reminder to engage with the authors in this phase of the rebuttal. The deadline to respond to authors is EOD Anywhere on Earth timezone today.

---

### Author Response · Authors · 2024-08-14
**Discussion summary**

Dear Area Chair(s) and Reviewers,

We thank the AC kindly for their effort in handling our submission, and for the help facilitating the discussion with the reviewers.
Throughout the discussions, reviewers praised the paper for the well-motivated approach to the important goal of scalable expert specialization (**Reviewer XreH**), its technical soundness and `"extensive qualitative and quantitative evidence to support the claims"` (**Reviewer Su68**), and the innovative new metric introduced to quantify expert specialization (**Reviewer 1sqG**).

**All three of the reviewers to respond to our rebuttals (**qC51**, **XreH**, and **YTQ3**) raised their score following our new experimental results**: **Reviewer qC51** increased their score to a 6 following the new comparisons to Soft and Sparse MoEs, where muMoEs demonstrated Pareto-optimal improvements to accuracy for CLIP fine-tuning. **Reviewer YTQ3** also increased their score after these experiments and additional clarifications about the paper's aims. We also performed new experiments on steering GPT-2 models for **Reviewer XreH**, who raised their support for the paper too.

Two reviewers are yet to respond to our rebuttals. However, **our rebuttals address all concerns of the remaining reviewers**, which we summarize below (and describe how the key concerns are addressed):

- **Reviewer-1sqG**: the reviewer already leaves a favorable score of 7, and we propose to acknowledge additional limitations in the camera-ready (regarding robustness to OOD data and experiments on very large models).
- **Reviewer-Su68**: we clarify in our rebuttal that **the fairness results are indeed already over multiple runs**, and highlight where detail of the metrics can be located in the appendix.

Given the lack of responses from the remaining reviewers, we kindly request that you consider our rebuttal's comprehensive resolution of the remaining concerns when making your decision about the paper.

We will incorporate all changes suggested by the reviewers and clarifications that emerged from the insightful technical discussions into our camera-ready version. We are grateful to the reviewers for helping improve the paper significantly.

Sincerely,

Submission #3407 Authors.

---

### Decision · Program_Chairs · 2024-09-25

**Decision:**

Accept (poster)

**Comment:**

The authors proposed a multilinear Mixture of Experts Layer. This method, unlike existing MoE approaches, scales to a large number of experts, and the authors show improved interpretability, specialisation and accuracy. The proposed method is differentiable as it is a linear combination of the different experts, and scalable by using low-rank factorisations of these experts.

During the rebuttal, the authors also clarified that although they propose an MoE architecture, the motivation is to improve the parameterisation of standard dense models, which is why the parameter-counts of their models are not as high as Soft- or Sparse-MoE baselines.

The authors also performed numerous experiments throughout the rebuttal which addressed most of the reviewers concerns. Authors should incorporate all of these changes into the final camera ready version of the paper.